# Significant formation of sulfate aerosols contributed by the heterogeneous drivers of dust surface

Tao Wang[1], Yangyang Liu[1], Hanyun Cheng[1], Zhenzhen Wang[1], Hongbo Fu[1], Jianmin Chen[1], Liwu Zhang[1,2*]

1 Shanghai Key Laboratory of Atmospheric Particle Pollution and Prevention, National Observations and Research Station for Wetland Ecosystems of the Yangtze Estuary, IRDR international Center of Excellence on Risk Interconnectivity and Governance on Weather, Department of Environmental Science & Engineering, Fudan University, Shanghai, 200433, Peoples' Republic of China

2 Shanghai Institute of Pollution Control and Ecological Security, Shanghai, 200092, Peoples' Republic of China

*Correspondence to:* Liwu Zhang (zhanglw@fudan.edu.cn)

**Abstract.** The importance of dust heterogeneous oxidation in the removal of atmospheric $SO_2$ and formation of sulfate aerosols is not adequately understood. In this study, the Fe, Ti, Al-bearing components, $Na^+$, $Cl^-$, $K^+$, and $Ca^{2+}$ of the dust surface were discovered to be closely associated with the heterogeneous formation of sulfate. Regression models were then developed to make a reliable prediction for the heterogeneous reactivity based on the particle chemical compositions. Further, the recognized gas-phase, aqueous-phase and heterogeneous oxidation routes were quantitatively assessed and kinetically compared by combining the laboratory work with modeling study. In the presence of 55 μg m$^{-3}$ airborne dust, heterogeneous oxidation accounts for approximately 28.6% of the secondary sulfate aerosols during nighttime, while the proportion decreases to 13.1% in the presence of solar irradiation. On the dust surface, heterogeneous drivers (e.g. transition metal constituents, water-soluble ions) are more efficient than surface adsorbed oxidants (e.g. $H_2O_2$, $NO_2$, $O_3$) in the conversion of $SO_2$, particularly during nighttime. Dust heterogeneous oxidation offers an opportunity to explain the missing sulfate source during severe haze pollution events, and its contribution proportion in the complex atmospheric environments could be even higher than the current calculation results. Overall, the dust surface drivers are responsible for the significant formation of sulfate aerosols and have profound impacts on the atmospheric sulfur cycling.

# 1    Introduction

As an important component of atmospheric particulate matters, sulfate exerts profound impacts on the Earth's climate system, air quality, and public health (Seinfeld and Pandis, 2016; Wang et al., 2021a). The rapid formation of sulfate was proven as largely responsible for London Fog and Beijing Haze (Wang et al., 2016; Cheng et al., 2016). Secondary sulfate aerosols originate predominately from the conversion of sulfur dioxide ($SO_2$) via the gas-phase oxidation in gaseous environments, aqueous-phase oxidation in liquid media, and heterogeneous oxidation on aerosol surfaces (Ravishankara, 1997; Mauldin III et al., 2012; Su et al., 2020; Liu et al., 2021a). In recent years, the newly found sulfate formation pathways were kinetically compared with the documented ones to evaluate the relative importance of them (Cheng et al., 2016; Gen et al., 2019; Liu et al., 2020a; Wang et al., 2020a). In addition, the reported oxidation channels were compared with each other by aerosol observations or modeling investigations (Berglen et al., 2004; Sarwar et al., 2013; He et al., 2018; Ye et al., 2018; Fan et al., 2020; Tao et al., 2020; Zheng et al., 2020; Song et al., 2021; Tilgner et al., 2021; Liu et al., 2021b; Gao et al., 2022; Wang et al., 2022a; Ye et al., 2022). In general, these studies emphasized the importance of certain newly discovered aqueous-phase process or compared the contributions from the documented gas- and aqueous-phase pathways. Nevertheless, heterogeneous reaction was scarcely involved in discussion, thus hindering the deeper understanding of the atmospheric relevance of aerosol surfaces.

Heterogeneous reaction alters the concentrations of gas-phase $SO_2$ and particle-phase sulfate, and its atmospheric influences were considered by observation and modeling works (Fairlie et al., 2010; Alexander et al., 2012; Chen et al., 2017; Wang et al., 2021b). As summarized by Table S1, when simulating the sulfate burst events, researchers observed the positive feedbacks after implementing heterogeneous mechanism into the WRF-Chem (Li et al., 2017), GEOS-Chem (Shao et al., 2019), CAMx (Huang et al., 2019) and CMAQ (Zhang et al., 2019a) models. However, these improved models highlighted the heterogeneous oxidation motivated by the surface adsorbed oxidants rather than the heterogeneous drivers of aerosol surface. To address the knowledge gap, the revised GEOS-Chem (Wang et al., 2014) and WRF-CMAQ (Zheng et al., 2015) models considered the heterogeneous oxidation driven by aerosol surfaces, and successfully reproduced the rapid sulfate formation. Xue et al. (2016) moved forward to develop an observation-based model and found that heterogeneous pathway contributed up to one third of the secondary sulfate in a typical haze-fog event. However, the "aerosol surface" mentioned in the previous works was not distinguished by its physical or chemical properties (Wang et al., 2014; Zheng et al., 2015; Xue et al., 2016), thereby making it difficult to compare the atmospheric importance of the diverse airborne surfaces. As discussed above, it is of great importance to investigate the heterogeneous drivers of one selected aerosol surface, and further evaluate the atmospheric significance of the relevant heterogeneous pathway.

The resurgence of sandstorms in North China makes the air pollution situation more complex than ever before. The concentration of $PM_{10}$ (particulate matter with an aerodynamic diameter of less than ten micrometers) in Beijing reached up to 3600 $\mu g\ m^{-3}$, largely beyond the standards of World Health Organization and Chinese government (Li et al., 2021a). As the

most abundant primary aerosol in the troposphere (Textor et al., 2006; Tang et al., 2016), dust particles could transport more than one full circuit around the globe within ~ 2 weeks (Uno et al., 2009) and concurrently participate into an array of atmospheric reactions. Heterogeneous reactions over dust surface consume and produce various trace gases, thereby affecting the dust property and tropospheric oxidation capacity (Tang et al., 2016). One of the most extensive concerns is that the numerous surface sites of windblown dust provide opportunities for a variety of atmospheric reactions to occur, e.g. oxidation of $SO_2$ and formation of sulfate (Usher et al., 2003). In the past decades, plenty of laboratory works have been performed to explore the heterogeneous behaviors of $SO_2$ on dust surfaces.

When discussing the heterogeneous oxidation on dust proxy, environmental factors like humidity, temperature, irradiance were frequently concerned. Adsorbed water not only helps to accumulate surface $S_{(IV)}$ species, but also competes with $SO_2$ for surface sites (Rubasinghege and Grassian, 2013). The reversible adsorption of $SO_2$ is believed to be exothermic (Clegg and Abbatt, 2001a). In contrast, there were positive temperature dependences observed for $CaCO_3$ below 250 K over the entire reaction (Wu et al., 2011) and for $Fe_2O_3$ within 284-318 K during the initial reaction stage (Wang et al., 2018a). Light irradiation normally accelerates the transformation of (bi)sulfites to (bi)sulfates (Li et al., 2010; Nanayakkara et al., 2012; Han et al., 2021), while iron oxides may undergo photoreactive dissolution and thus exhibit weaker reactivity under sunlight (Fu et al., 2009). The reactivity can be enhanced by $H_2O_2$ (Huang et al., 2016), or $O_3$ (Li et al., 2006; Li et al., 2007; Zhang et al., 2018), or $NO_x$ (Ma et al., 2008; Liu et al., 2012; He et al., 2014; Yu et al., 2018; Zhao et al., 2018; Yang et al., 2018a; Wang et al., 2020b), or $NH_3$ (Yang et al., 2016; Yang et al., 2018b; Yang et al., 2019), or $Cl_2$ (Huang et al., 2017). By contrast, organic compounds, like $CH_3CHO$ (Zhao et al., 2015), HCOOH (Wu et al., 2013), $CH_3COOH$ (Yang et al., 2020; Wang et al., 2022b) and $C_3H_6$ (Chu et al., 2019), were found to suppress the interactions due to the accumulation or production of particle-phase organic acids. Moreover, $CO_2$ hinders the heterogeneous oxidation under dark condition (Liu et al., 2020b), whereas presents positive impacts on sulfate formation under solar irradiation (Liu et al., 2022). The particle physical properties, including size (Baltrusaitis et al., 2010; Zhang et al., 2016), morphology (Li et al., 2019) and crystal structure (Yang et al., 2017), display varied impacts on heterogeneous reaction. When considering the particle chemical properties, $Fe_2O_3$ is more active than $Al_2O_3$ and $SiO_2$ in the heterogeneous uptake of $SO_2$ and formation of sulfate (Chughtai et al., 1993; Zhang et al., 2006; He et al., 2014). Furthermore, the presence of moderate nitrate (Kong et al., 2014; Du et al., 2019), or $Al_2O_3$ (Wang et al., 2018b), or surfactant (Zhanzakova et al., 2019), or oxalate (Li et al., 2021b), or (bi)carbonate (Liu et al., 2022) on dust surfaces could favor the heterogeneous kinetics under specific conditions. Relative to the environmental factors, the heterogeneous effects relevant to particle property are still under debate.

The usage of simple mineral oxide as a substitution for natural dust may be problematic as such approach could undermine the atmospheric importance of more complex mineralogy. Authentic dusts were utilized in laboratory works (Table S2). Some studies concerned single samples like Saharan dust (Ullerstam et al., 2002; Ullerstam et al., 2003; Adams et al., 2005; Harris et al., 2012), Arizona test dust (Park and Jang, 2016; Zhang et al., 2019a; Zhang et al., 2019b), China loess (Usher et al., 2002) and Asian dust (Ma et al., 2012). The comparison of diverse samples has been attracting increasing attention. Zhou et al. (2014) observed the positive temperature effect on Xinjiang sierozem, in contrast to the negative temperature dependence for Inner

Mongolia desert dust. Huang et al. (2015) discovered the accelerated oxidation of $SO_2$ by $H_2O_2$, and attributed the different moisture effects to the dusts' varying components. Maters et al. (2017) explored the heterogeneous uptake on volcanic ash and glass samples, and related the reactivity differences to the varying abundances of surface basic and reducing sites. Park et al. (2017) compared the heterogeneous reactions on Gobi desert dust and Arizona test dust, and linked the sulfate formation to the quantity of semi-conductive metals. Wang et al. (2019) discovered the enhanced uptake of $SO_2$ on clay minerals after the simulated cloud processing, and explained this evolution by the modification of iron speciation. Urupina et al. (2019) discussed the kinetics of diverse volcanic dusts, and further experimentally proved that neither one selected pristine oxide nor a mixture of them can adequately typify the behavior of natural dust (Urupina et al., 2021). Recently, based on the measurement method for sulfite and sulfate (Urupina et al., 2020), they determined the associations between sulfate production and dust chemical properties like (Fe+Al)/Si and Na abundance (Urupina et al., 2022). The aforementioned works broaden the horizons of the heterogeneous drivers on dust surface. Up to now, the dominate dust surface drivers remain controversial due to the limited statistical linkages between the chemical composition of dust and the production rate of sulfate.

While the atmospheric relevance of the oxidation of $SO_2$ on dust has been widely recognized (Wu et al., 2020; Xu et al., 2020), the contribution of dust heterogeneous oxidation to secondary sulfate aerosols has not been quantitatively determined. By means of the improved WRF-CMAQ model, Wang et al. (2012) attributed a 27% decrement of $SO_2$ concentration and a 12% increment of sulfate concentration to the heterogeneous processes during an Asian dust storm. Moreover, Tian et al. (2021) recently simulated the heterogeneous formation of dust sulfate by the revised GEOS-Chem model and found that, during the dust episodes in North China, up to 30% of the secondary sulfate resulted from the heterogeneous processes on dust surface. However, the photocatalytic reactivity of dust was not considered by the advanced models. Yu et al. (2017) developed the atmospheric mineral aerosol reaction (AMAR) model based on the laboratory works, and suggested that the heterogeneous photocatalysis of mineral dust surface contributed more than half of the secondary sulfate. However, the heterogeneous reactivity was measured by quantifying all the adsorbed $SO_2$ rather than calculating the yield of particle-phase sulfate. Moreover, limited gas- and aqueous-phase pathways were included in these models, and the heterogeneous reactivities of surface adsorbed oxidants and dust surface drivers have not been distinguished yet. Therefore, it is highly desirable to comprehensively compare the dust heterogeneous pathways with other documented sulfate formation pathways.

Hereby, upon understanding the driving factors and driving force of the airborne dust surface, this work compared dust heterogeneous pathways with the gas- and aqueous-phase ones with respect to the formation rate of sulfate and atmospheric lifetime of $SO_2$. In order to characterize the sensitivity of heterogeneous reaction to dust loading, the scenarios with different dust concentrations were also considered. The joint influences of ionic strength and aerosol liquid water content on the aqueous-phase oxidation of $SO_2$ were further discussed to prove the significance of heterogeneous kinetics under diverse atmospheric conditions. The recently reported microdroplet interfacial oxidations were additionally compared with the dust-related heterogeneous pathways to emphasize the atmospheric relevance of dust surface drivers. This study attempted to verify the significant formation of sulfate aerosols contributed by the heterogeneous drivers of dust surface.

## 2 Methods

### 2.1 Methodology overview

This study attempted to investigate whether the heterogeneous oxidation of $SO_2$ on dust surface, particularly that induced by the dust surface drivers, makes great impacts on the loss of gaseous $SO_2$ and formation of sulfate aerosols (Text S1, Fig. S1). The gas- and aqueous-phase pathways were assessed by the documented methodologies and parameterizations (Cheng et al., 2016; Seinfeld and Pandis, 2016; Shao et al., 2019; Song et al., 2021), as briefly introduced by Sect. 2.2 (more details in Supporting Information). The heterogeneous conversion of $SO_2$ comprises dust-mediated and dust-driven modes, emphasizing the oxidants co-adsorbed with $SO_2$ and the drivers of dust surface, respectively. In dust-mediated mode, dust surface functions as a reaction medium that supports the interaction between adsorbed oxidants and $SO_2$. In dust-driven mode, the oxidation of adsorbed $SO_2$ is initiated by the active components of dust surface. The former mode was assessed by the particle properties based on the reported methodology, while the later mode was quantitatively characterized by the laboratory works.

### 2.2 Gas- and aqueous-phase oxidation pathways

The gas-phase oxidation of $SO_2$ is initiated by hydroxyl radical (OH), stabilized Criegee intermediates (CIs) and nitrate radical ($NO_3$). The former two oxidants promote the sulfate formation during daytime, while the latter one works mainly during nighttime. More details can be found in Text S2.

The aqueous-phase sulfate formation is pH-dependent and can be quantified based on the published documents and the references cited therein (Cheng et al., 2016; Su et al., 2020; Liu et al., 2021a). Herein, aqueous-phase pathway refers to the liquid $SO_2$ conversion by the transition-metal ion-catalyzed oxygen (TMI-$O_2$), ozone ($O_3$), hydrogen peroxide ($H_2O_2$), nitrogen oxide ($NO_2$), methyl hydroperoxide ($CH_3OOH$), peroxyacetic acid ($CH_3COOOH$), hypochlorous acid (HOCl), hypobromous acid (HOBr), dissolved nitrous acid (HONO), photosensitization (T*) and nitrate photolysis ($P_{NO_3^-}$). The processes relevant to T* and $P_{NO_3^-}$ are only considered for the daytime scenario. More details can be found in Text S3 and Tables S3-S4.

The oxidant concentration data were derived from the atmospheric observation campaigns performed in Beijing, North China. The measurements conduced in warm seasons were considered in priority to correspond the experimental temperature. The relevant temperature range is 293-303 K in representative of warm season. Considering the relatively high irradiance used in the laboratory experiments, the oxidant parameters for daytime discussion were selected from the observations performed at noon time. The influences of dust loading on the oxidant concentrations, nitrate photolysis kinetics and TMI abundances were considered to reflect the linkages between different reactions. More details can be found in Text S4 and Table S5.

### 2.3 Heterogeneous oxidation-particle characterizations

Five airborne clay minerals, including Nontronite (NAu), Chlorite (CCa), Montmorillonite (SWy), Kaolin (KGa) and Illite

(IMt), were obtained from the Source Clay Minerals Repository (Purdue University, West Lafayette, Indiana, USA). The purchased clays were sent for the following measurements.

The clay minerals were analyzed by X-ray fluorescence spectrometer (Axios Advanced, PANalytical, Netherlands) for element distributions (Table S6). The Brunauer-Emmett-Teller (BET) specific surface areas ($S_{BET}$) of NAu, CCa, SWy, KGa and IMt were measured by a Quantachrome Nova 1200 BET apparatus to be 19.76, 5.67, 22.64, 18.77 and 20.05 $m^2$ $g^{-1}$, respectively.

Particles were suspended in Milli-Q water (18.2 MΩ.cm at 25 °C) before the size measurement (ViewSizer$^{TM}$ 3000, MANTA Instruments, USA). The particle diameters range mostly from 50 to 1000 nm and are averaged to be 399 nm for NAu, 272 nm for CCa, 438 nm for SWy, 396 nm for KGa, and 366 nm for IMt (Fig. S2).

Prepared particles were ultrasonically extracted in Milli-Q water, followed by the filtration through a polytetrafluoroethylene membrane filter. The obtained solution was analyzed by an ion chromatography (883 Basic, Metrohm, Switzerland) for the concentrations of anions ($HCOO^-$, $Cl^-$, $NO_3^-$, $SO_4^{2-}$) and cations ($Na^+$. $NH_4^+$, $K^+$, $Mg^{2+}$, $Ca^{2+}$) (Fig. S3) by the reported methods (Wang et al., 2020c). The water-soluble ions account for 4.6‰, 0.1‰, 8.5‰, 0.3‰ and 0.8‰ of the mass contents of NAu, CCa, SWy, KGa and IMt, respectively.

A mixture, denoted as natural dust hereafter, was prepared by mechanically mixing the studied clay minerals by their atmospheric abundances. Because the clays in Kaolinite group (NAu, KGa), Montmorillonite group (SWy), Illite group (IMt) and Chlorite group (CCa) occupy respectively 6.6%, 4.0%, 53.8% and 4.3% mass fractions of the airborne dust (Usher et al., 2003), the prepared natural dust sample comprises 8.8 wt% of NAu, 8.8 wt% of KGa, 5.3 wt% of CCa, 71.4 wt% of IMt, and 5.7 wt% of CCa.

## 2.4 Heterogeneous oxidation-DRIFTS measurements

The *in-situ* DRIFTS (diffuse reflectance infrared Fourier transform spectroscopy) spectra were collected using a FTIR spectrometer (Tracer-100, Shimadzu, Japan) equipped with a mercury-cadmium-telluride detector cooled by liquid nitrogen. A gas supply system was constructed by linking the experimental units through Teflon tubes. Mass flow controllers (D07-19, Severstar, China) adjusted the reactant gases to the expected reactant concentration and relative humidity (RH). Gas cylinders: high-pure air (79% $N_2$ and 21% $O_2$), $SO_2$ (2.46 × $10^{15}$ molecules $cm^{-3}$ diluted by $N_2$).

Before each experiment, a ceramic cup holding particles was placed into the reaction chamber. The particles were treated in a stream of dry air (300 ml $min^{-1}$) for 30 min to minimize the surface water and impurities (Wang et al., 2018a, 2018c). After the pretreatment, the sample was exposed to humidified air (RH = 50%) to reach moisture saturation, followed by the collection of background spectrum and then the introduction of reactant gas for 240 min. The $SO_2$ concentration was 3.69 × $10^{13}$ molecules $cm^{-3}$ in a total flow rate of 100 ml $min^{-1}$.

The simulated solar irradiation with an actinic flux of 6.51 × $10^{15}$ photons $cm^{-2}$ $s^{-1}$ was provided by an Xenon lamp (TCX-250, Ceaulight, China). The reaction temperature (296.8 K) was controlled by a heater attached to a recirculating cooling water system and determined by a calibration curve introduced previously (Wang et al., 2018a). The kinetic parameters involved in

the gas- and aqueous-phase oxidation pathways (listed by Sect. 2.2) were corrected by the experimental temperature as much as possible. All exposures were performed in triplicates. The experimental setups are displayed by Fig. S4, and the recorded spectra are shown in Fig. S5 and S6.

## 2.5  Heterogeneous oxidation-acidity and kinetics

The recorded spectra were analyzed by referring to the previous literatures and the references cited therein (Persson and Vgren, 190    1996; Peak et al., 1999; Goodman et al., 2001; Zhang et al., 2006; Wu et al., 2011; Liu et al., 2012; Nanayakkara et al., 2012; Nanayakkara et al., 2014; Huang et al., 2016; Ma et al., 2017; Yang et al., 2017). Totally six sulfur-containing species can be identified: hydrated $SO_2$ ($SO_2 \cdot H_2O$), bisulfite, sulfite, solvated sulfate, coordinated sulfate, bisulfate, and the assignments are summarized by Table S7. The overlapping bands were further analyzed by Gaussian/Lorentzian deconvolution to obtain the product distributions (Fig. S7). The consistent deconvolution procedure was performed for the infrared spectra derived from 195    the repeated experiments.

Because part of the measured ions exist in the surface coordinated forms or crystalline states, the particle acidity (pH) may not be accurately characterized by the typical proxy methods (Hennigan et al., 2015). Herein, the ionization equilibrium of dissolved $SO_2$ in the water layers of particle surface is considered to be associated with particle acidity. As reported, the dissolved $SO_2$ would transform from $SO_2 \cdot H_2O$ to $HSO_3^-$, and then to $SO_3^{2-}$ as the medium evolves from the extremely acidic 200    to the nearly alkaline (Haynes, 2014; Zhang et al., 2015). The relative abundance of $SO_2 \cdot H_2O$ and $SO_3^{2-}$, as assumed to be equivalent to the relative integral area of their characteristic peaks, is utilized to calculate the particle acidity (more details in Text S3-2 of Supporting Information). The relative abundance of $HSO_4^-$ and $SO_4^{2-}$, which can be used to calculate the acidity (Rindelaub et al., 2016; Ault, 2020), was not considered by the present study because the characteristic signals of bisulfate cannot be observed in some of the recorded spectra. As noted by Keene et al. (2004) and Hennigan et al. (2015), estimating 205    the pH of airborne aerosol by ion balance may be difficult because it is unable to distinguish between free and undissociated $H^+$ (e.g. protons associated with $HSO_4^-$ and $HSO_3^-$). Herein, the relative abundance of $S_{(IV)}$ species is derived from the different infrared absorption signals relevant to atomic sulfur. Thus, the current method is recommended for the pH determination of humidified gas-solid interface.

Heterogeneous kinetics can be assessed by the reactive uptake coefficient ($\gamma$) by assuming a first-order loss of $SO_2$. The $\gamma$ 210    for dust-driven heterogeneous reaction can be calculated by:

$$\gamma = \frac{d[SO_4^{2-}]/dt}{Z} \tag{1}$$

$$Z = \frac{1}{4} \times A_S \times [SO_2] \times v_{SO_2} \tag{2}$$

$$v_{SO_2} = \sqrt{\frac{8RT}{\pi M_{SO_2}}} \tag{3}$$

where $d[SO_4^{2-}]/dt$ is the rate of sulfate production on dust surface (ion s[-1]), $A_s$ is the reactive surface area (m[2]), $[SO_2]$ is the

experimental concentration of $SO_2$ (molecules m$^{-3}$), $v_{SO_2}$ is the molecular velocity of $SO_2$ (m s$^{-1}$), R is gas constant (J mol$^{-1}$ K$^{-1}$), T is the experimental temperature (K), $M_{SO_2}$ is the molecular weight of $SO_2$ (kg mol$^{-1}$). Because the infrared intensity is proportional to the amount of surface product, $[SO_4^{2-}]$ can be translated by the integral area of the sulfate characteristic peaks:

$$[SO_4^{2-}] = f \times \text{(integral area)} \tag{4}$$

where f is the conversion factor and represents the number of $SO_4^{2-}$ corresponding to per unit integral area. The sulfate production rate of dust surface can be translated by the calibration curves by mixing weighed $Na_2SO_4$ with the target particle sample to a set of concentrations (Martin et al., 1987; Li et al., 2006; Wu et al., 2011; Wang et al., 2020d). The conversion factors are calculated to be $1.32 \times 10^{18}$, $4.62 \times 10^{17}$, $6.97 \times 10^{17}$, $8.20 \times 10^{17}$, $9.44 \times 10^{17}$, and $9.25 \times 10^{17}$ for NAu, CCa, SWy, KGa, IMt, and natural dust, respectively. Because $Na_2SO_4$ was thoroughly mixed with the particles, $S_{BET}$ was used to calculate $\gamma$.

All samples except SWy presented steady sulfate production potentials over the entire experiment, while the products on SWy increased by reaction time in the beginning and then gradually remained unchanged (Fig. S8). Accordingly, the process on SWy was assessed by the experimental data of the first 30 min of reaction, and the kinetics of other samples were calculated by the spectra recorded throughout the experiment.

Besides the dust surface drivers, the heterogeneous conversion of $SO_2$ can be additionally initiated by the gaseous oxidants ($O_3$, $H_2O_2$, $NO_2$, HOCl, HOBr, $CH_3OOH$, $CH_3COOOH$, HONO) co-adsorbed with $SO_2$. Considering that the adsorption of oxidant onto dust surfaces would produce gas- and particle-phase species (Usher et al., 2003; Tang et al., 2017), the dust-mediated heterogeneous oxidation can be assumed to primarily proceed in the surface water layers. Such assumption can be experimentally proved as the acceleration of $SO_2$ oxidation induced by the co-oxidant (e.g. $H_2O_2$, $NO_2$, $O_3$) becomes more significant under higher humidity (Huang et al., 2015; Park et al., 2017; Zhang et al., 2018). Based on the literatures (Hanson et al., 1994; Jacob, 2000; Seinfeld and Pandis, 2016; Shao et al., 2019), the $\gamma$ for dust-mediated heterogeneous reaction can be calculated by:

$$\gamma = \left[\frac{1}{\alpha} + \frac{v}{4H^*RT\sqrt{D_a k_{chem}}} \times \frac{1}{f_r}\right]^{-1} \tag{5}$$

$$k_{chem} = \frac{R_a}{\sum S_{(IV)}} \tag{6}$$

$$f_r = \coth\frac{r_p}{l} - \frac{l}{r_p} \tag{7}$$

$$l = \sqrt{\frac{D_a}{k_{chem}}} \tag{8}$$

where $k_{chem}$ is the pseudo first-order reaction rate constant of the studied $S_{(IV)}$ specie(s) (s$^{-1}$) that produce in the rate of $R_a$ (M s$^{-1}$), $f_r$ is the diffusive correction term comparing the radius of natural dust $r_p$ (m) with the diffuse-reactive length l (m), $\alpha$ is the mass accommodation coefficient of $SO_2$ (dimensionless), $D_a$ is the aqueous-phase diffusion coefficient of $SO_2$ ($1.78 \times 10^{-5}$ m$^2$

s$^{-1}$ at 296.8 K) (Himmelblau, 1964; Haynes, 2014), H$^*$ is the effective Henry's law constant for the studied S$_{(IV)}$ specie(s) (M atm$^{-1}$), and is jointly determined by the gas-liquid equilibrium of SO$_2$ and the ionization equilibriums of dissolved sulfur species, and R is gas constant (L atm mol$^{-1}$ K$^{-1}$). The terms on the right-hand side (RHS) of Eq. (5) show the two contributions to the overall resistance to dust-mediated heterogeneous uptake: mass accommodation at surface, diffusion and reaction in surface liquid water layers. The S$_{(IV)}$ species include SO$_2$·H$_2$O, HSO$_3^-$ and SO$_3^{2-}$.

The sulfate formation rate in the atmosphere can be calculated by the following equations (Jacob, 2000; Li et al., 2020a).

$$\frac{d[SO_4^{2-}]}{dt} = \left(\frac{r_p}{D_g} + \frac{4}{v\gamma}\right)^{-1} S_p[SO_2] \tag{9}$$

$$S_P = C \times F \times S_{BET} \tag{10}$$

where $d[SO_4^{2-}]/dt$ is the atmospheric sulfate formation rate (µg m$^{-3}$ h$^{-1}$), $r_p$ is the particle radius (m), $D_g$ is the gas-phase diffusion coefficient of SO$_2$ (m$^2$ h$^{-1}$), $v$ is the molecular velocity of SO$_2$ (m s$^{-1}$), $\gamma$ is the reactive uptake coefficient (dimensionless), $S_p$ is the particle surface area density (m$^2$ m$^{-3}$), [SO$_2$] is the atmospheric SO$_2$ concentration (µg m$^{-3}$), C is the dust concentration of 55 µg m$^{-3}$ in representative of the common atmospheric condition of China (Zhang et al., 2012), F is the mass fraction of clay mineral in the natural dust community (%), $S_{BET}$ is the BET specific surface area (m$^2$ g$^{-1}$).

## 2.6  Atmospheric lifetime of SO$_2$

For the gas- and aqueous-phase pathways, the lifetime of SO$_2$ ($\tau$) can be calculated by Eq. (11) (Jacob, 2000).

$$\tau = \frac{1}{k_{chem}} \tag{11}$$

where $k_{chem}$ is the assumed first-order reaction rate constant of the studied S$_{(IV)}$ specie(s) (s$^{-1}$).

For the heterogeneous pathways, the $\tau$ can be calculated by Eq. (12) (Clegg and Abbatt, 2001b).

$$\tau = \frac{4}{\gamma v S_p} \tag{12}$$

where $\gamma$ is the reactive uptake coefficient (dimensionless), $v$ is the molecular velocity of SO$_2$ (m s$^{-1}$), $S_p$ is the particle surface area density (m$^2$ m$^{-3}$), as described by Eq. (10).

The atmospheric lifetime caused by the multiple pathways ($\tau_{total}$) can be estimated by Eq. (13) (Seinfeld and Pandis, 2016).

$$\tau_{total} = \left(\sum \frac{1}{\tau}\right)^{-1} \tag{13}$$

# 3    Results and discussion

## 3.1    Driving factors of dust surface

Correlation analysis is performed to identify the dust surface drivers (Fig. 1), based on which the reaction mechanism of dust
heterogeneous pathway can be better understood (Scheme 1).

The mineral drivers are investigated at first (Fig. 1a). Under dark condition, the sulfate production rate correlates positively
with Fe, while presents negative dependence against Al. As documented, airborne sulfate was highly associated with Fe-rich
dusts, and the heterogeneous reaction of $SO_2$ can be regarded as a possible explanation (Sullivan et al., 2007). Typical Fe-
bearing minerals accelerate the oxidation of $S_{(IV)}$ species by either the surface active oxygen ($O^-$) derived from the adsorbed
$O_2$ in oxygen vacancy or the iron redox cycling initiated by $Fe^{3+}$ from surface acidic media, or both (Baltrusaitis et al., 2007;
Fu et al., 2007; Yang et al., 2016). Conversely, $Al_2O_3$ presents weaker heterogeneous reactivity than $Fe_2O_3$ (Zhang et al., 2006;
Yang et al., 2018b; Xu et al., 2021), and may hinder heterogeneous reaction by blocking the active sites of other mineral
constituents (Wang et al., 2018b). Generally, Fe-bearing component plays a crucial role in the heterogeneous reaction of $SO_2$,
while the presence of Al-bearing component directly weakens the dust's reactivity or indirectly decreases the proportion of
other active mineral constituents.

The sulfate yield enhanced by solar irradiation associates positively with the abundance of Ti or Al. Transition metal oxides
in dust act as photocatalyst that yields electron-hole pairs, followed by the formation of reactive oxygen species (ROS) such
as hydroxyl radical ($^\cdot OH$), superoxide ($O_2^{\cdot-}$), hydroperoxyl radical ($HO_2^\cdot$) and dissociated active oxygen species ($O^*$) (Chen et
al., 2012; Abou-Ghanem et al., 2020; Wang et al., 2020c; Sakata et al., 2021). Apart from the site-blocking effect under dark
condition, irradiated Al-bearing constituents additionally facilitate the sulfate formation. Physically, Al-bearing components
disperse other efficient mineralogical constituents in case of agglomeration (Darif et al., 2016). Chemically, sunlight may alter
the electronic configuration of $\alpha$-$Al_2O_3$, which presented photoactivity in the reported heterogeneous process (Guan et al.,
2014). Generally, the kinetic discrepancy between dark reaction and photoreaction relates primarily to the existence of Ti- and
Al-bearing minerals.

No correlation can be observed between the abundance of element and the sulfate production rate of photoreaction. The dust
with higher proportion of elemental Ti was reported to exhibit greater reactivity toward $SO_2$ (Park et al., 2017), or $NO_2$ (Ndour
et al., 2009), or $O_3$ (Abou-Ghanem et al., 2020) in the photochemical processes. However, these results were derived from
qualitative comparisons rather than the quantitative analysis performed herein. Analogous to the $Al_2O_3$ discussed above, semi-
conductive metal exhibits dual behaviors in dark and light reactions. For instance, the iron oxide in dark reaction owns great
reactivity toward $SO_2$, whereas that under solar irradiation may present impaired heterogeneous kinetics due to dissolution (Fu
et al., 2009). In addition, the heterogeneous reaction on the surface of $TiO_2$ or Ti-bearing mineral is thermodynamically favored
under dark condition but depends largely on photocatalysis under solar irradiation (Chen et al., 2012). Due to the complexity
of photoreaction that involves both dark and light reaction mechanisms, the sulfate formation rate of photoreaction may not be

directly linked to particle chemical compositions.

The driving effects of water-soluble ions are further studied (Fig. 1b). Herein, $Na^+$ and $Cl^-$ are observed to present positive impacts on the sulfate formation under dark condition. The presence of halite (NaCl) has positive implications for the dust's hydroscopic growth (Wang et al., 2014; Tang et al., 2019), and the moderate surface water provides an efficient medium for the adsorption of gas-phase $SO_2$ and the ionization of dissolved sulfur species (Rubasinghege and Grassian, 2013). Moreover, there are negative associations between the photoinduced sulfate enhancement and the abundances of $Na^+$, $K^+$, $Ca^{2+}$. These

cations can be hydrolyzed by adsorbed water to produce $H^+$, thereby elevating the particle acidity. In the surface aqueous medium, increased acidity retards the hydrolysis and dissociation of $SO_2$ (Park et al., 2008; Huang et al., 2015), and inhibits the production of OH by irradiated dust surface (Zheng et al., 1997; Yang et al., 2008; Liu et al., 2017a). Urupina et al. (2022) reported the positive correlations between secondary dust sulfate and the amount of elemental Na. The results here provide additional insights into the heterogeneous effects of the water-soluble constituents on dust surface.

Apart from the dust surface drivers and inhibitors, sulfate radical ($SO_4^{\bullet-}$) may also participate into the heterogeneous event. In the absence of solar irradiation, laboratory studies discovered the uncatalyzed $SO_2$ autoxidation by $SO_4^{\bullet-}$ at the acidic droplet interface (Hung and Hoffmann, 2015; Hung et al., 2018; Chen et al., 2022). With an appropriate humidity and sufficient saline components, there could be an air-liquid interface occurring over the solid dust surface via water uptake (Gaston et al., 2017; Tang et al., 2019; Wu et al., 2020), then producing $SO_4^{\bullet-}$. Moreover, on the surface of iron oxide, the $^{\bullet}OH$ derived from Fenton

reaction reacts with $SO_4^{2-}$ to form $SO_4^{\bullet-}$ (Kim et al., 2019; Li et al., 2020b). The presence of solar irradiation would accelerate the formation of $SO_4^{\bullet-}$ due to the abundant $^{\bullet}OH$ produced via photocatalysis (Antoniou et al., 2018). Generally, the dust surface with higher proportion of saline components and transition metals may cause the more significant oxidation by $SO_4^{\bullet-}$, especially under solar irradiation with higher ambient humidity and particle acidity.

    Based on the correlation analysis, regression analysis can be further performed to predict the surface kinetics by the chemical

compositions of dust. Referring to the publication of Zhang et al. (2020), we derived an exponential parameterization for the production rate of particle-phase sulfate:

$$SF_{dark} = M[A]^a[B]^b[C]^c[D]^d[E]^e[F]^f + N \tag{14}$$

where $SF_{dark}$ is the yield of $SO_4^{2-}$ on the particle sample of per unit mass within per unit time (ions $g^{-1}$ $s^{-1}$); [A], [B], [C], [D], [E] and [F] are the mass fractions of element and ion (%); M and N are the constant parameters (dimensionless).

After the statistical procedures by SPSS (version 22.0), we obtained the regression equation for dark reaction:

$$SF_{dark}(\times 10^{15}) = 22.858\ [Al]^{-0.001}[Fe]^{0.111}[Cl^-]^{0.001}[Na^+]^{0.001} - 13.404 \tag{15}$$

    Analogously, the discrepancy between the sulfate production rates under dark and illuminated conditions, denoted as $R_{light/dark}$, can be described by:

$$R_{light/dark} = SF_{light}/SF_{dark} = 14.539\ [Al]^{0.073}[Ti]^{0.167}[Cl^-]^{-0.001}[Na^+]^{-0.001}[K^+]^{-0.001}[Mg^{2+}]^{-0.001} - 3.186 \tag{16}$$

Then,

$$SF_{light} = SF_{dark} \times R_{light/dark} \tag{17}$$

Because the abundance variation of water-soluble ion presents significantly less influence on the prediction result relative to the elemental transition metal, the complete regression models can be simplified by merely considering the mineral element abundances:

$$SF_{dark}(\times\ 10^{15}) = 30.880\ [Fe]^{0.077} - 21.679 \tag{18}$$

$$R_{light/dark} = 15.581[Al]^{0.062}[Ti]^{0.141} - 4.277 \tag{19}$$

$$SF_{light} = SF_{dark}\ \times\ R_{light/dark} \tag{20}$$

The complete and simplified regression models both accurately simulate the experimental data with all linear slopes approaching 1.0 and all $R^2$ values larger than 0.995 (Fig. 2). The simplified regression model can be considered as the preferred recommendation due to the fewer parameters needed and its greater performance in the photoreaction prediction. Shang et al. (2010) found that the heterogeneous sulfate production on pristine $TiO_2$ (Degussa P25) can be accelerated by 8.4 times by the presence of ultraviolet light (365 nm, 350 μW cm$^{-2}$). Such photoinduced enhancement is comparable with the prediction result (10.2 times) derived from the simplified regression model developed by this study. Additionally, because the chemical composition of bulk sample may not fully explain the sulfate formation over dust surface, further study to discuss the model uncertainty is warranted.

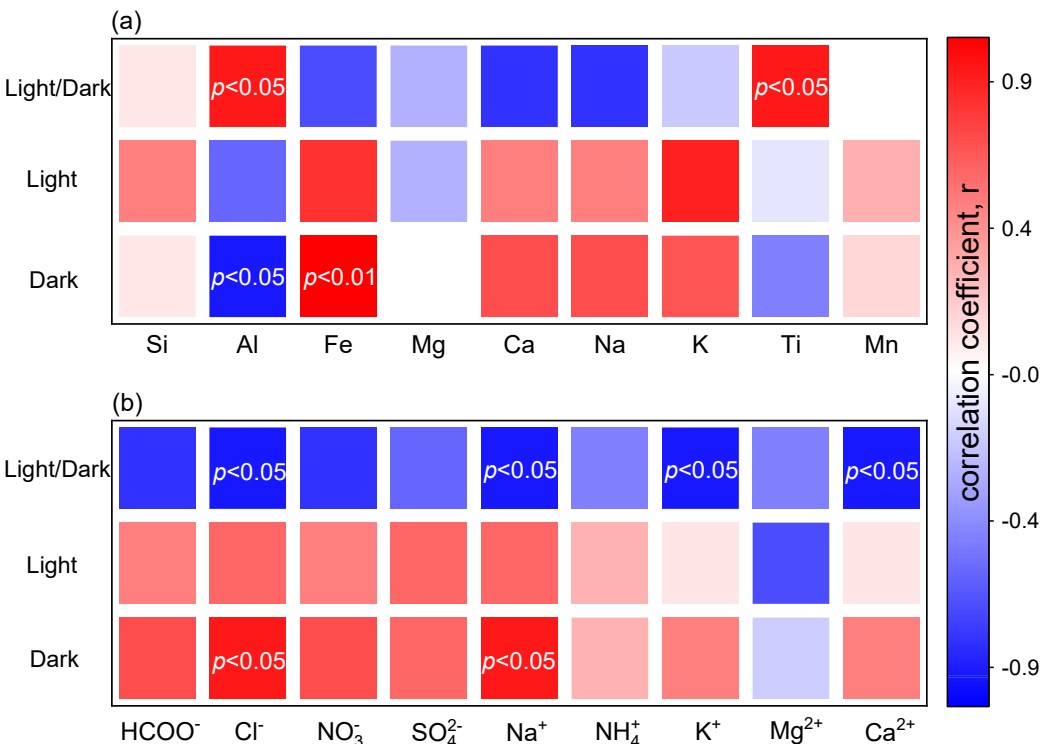

**Figure 1.** Correlation analysis on the sulfate production rate of dust surface.

Spearman correlation coefficient for the relationship between the rate of sulfate production and the abundance of **(a)** mineral element or **(b)** water-soluble ion. The significant correlations with $p$ values lower than 0.05 or 0.01 are highlighted by the labels in white.


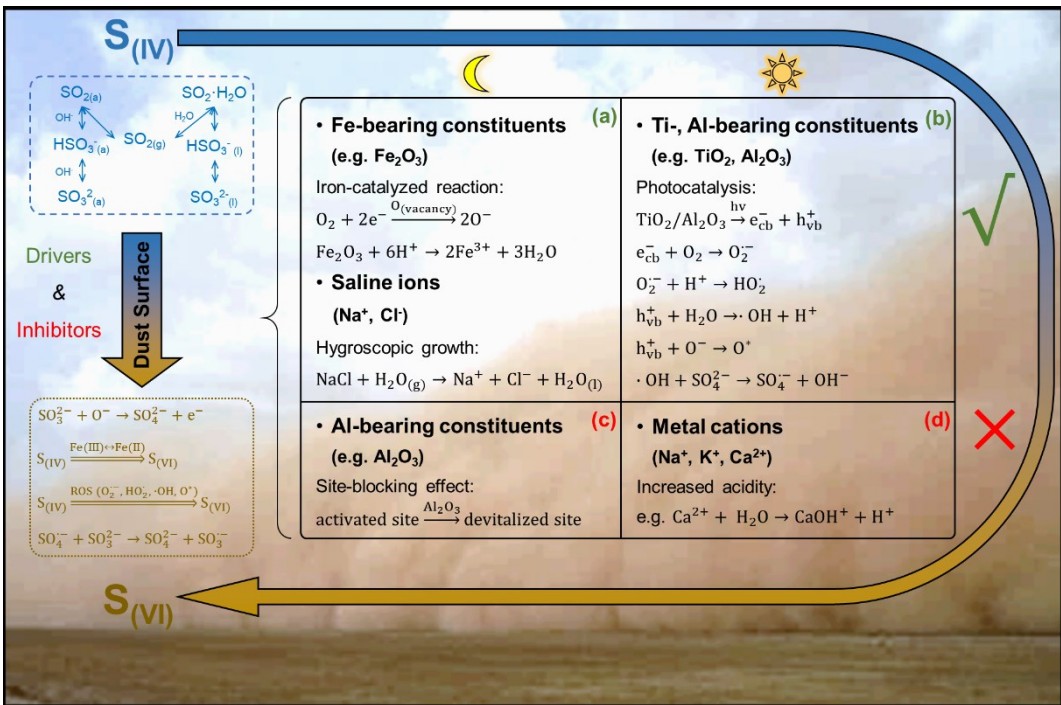

**Scheme 1.** Mechanism illustration on the dust-driven heterogeneous conversion of SO₂.

The heterogeneous reaction of airborne SO₂ on natural dust surface includes the initial adsorption of gas-phase SO₂ and the later conversion of $S_{(IV)}$ species to $S_{(VI)}$ products. The formed $S_{(IV)}$ species include both adsorbed and dissolved SO₂, bisulfite and sulfite (dashed box in blue). The generation of $S_{(VI)}$ products can be attributed to the surface active oxygen (O⁻), Fe(III)-Fe(II) redox cycling, reactive oxygen species (ROS), and sulfate radical (SO₄•⁻) (dotted box in deep yellow). Heterogeneous oxidation is largely influenced by the dust surface drivers and inhibitors, which promote and hinder the heterogeneous procedures, respectively (solid box in black). Dust surface drivers are determined as **(a)** the Fe-bearing constituents and saline ions (Na⁺, Cl⁻) under dark condition, and **(b)** the Ti- and Al-bearing constituents under solar irradiation. Dust surface inhibitors are identified as **(c)** the Al-bearing constituents under dark condition, and **(d)** the metal cations (Na⁺, K⁺, Ca²⁺) under solar irradiation.

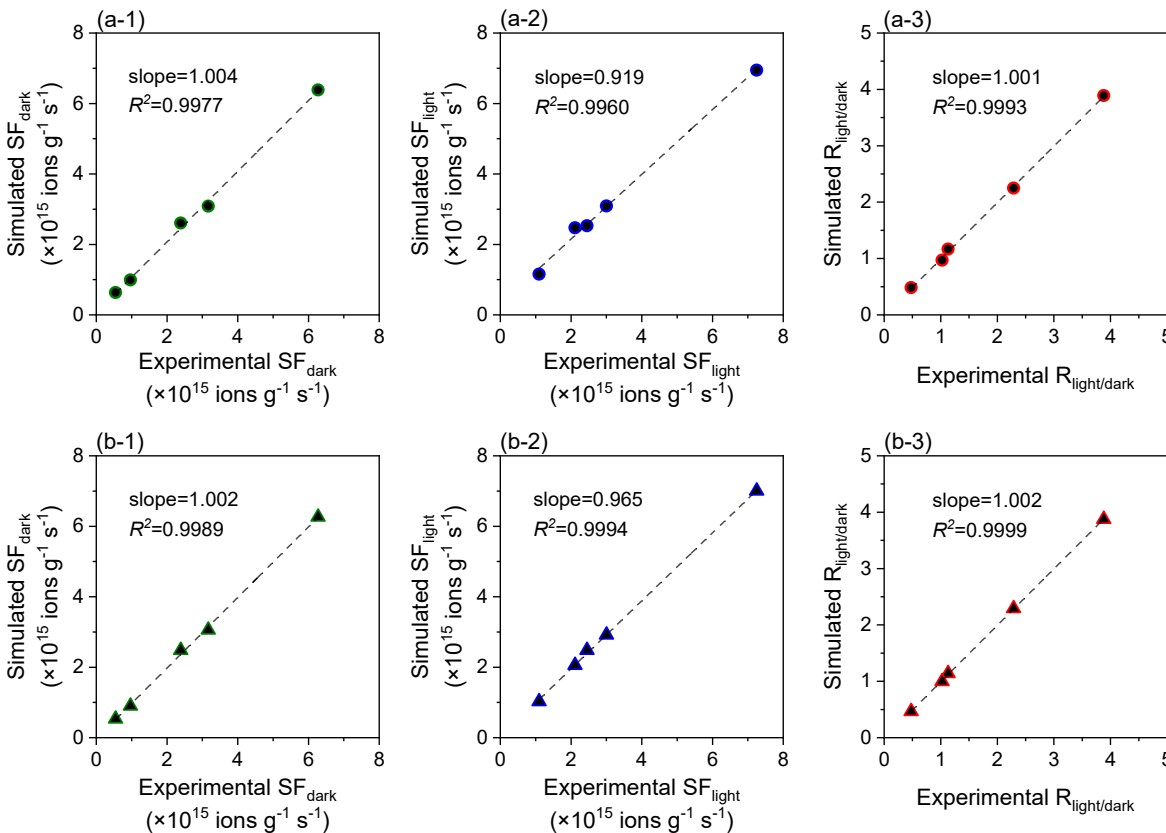

**Figure 2.** Regression analysis on the sulfate production rate of dust surface.

Linear relationships between the experimental $SF_{dark}$, $SF_{light}$, $R_{light/dark}$ and those simulated by the **(a)** complete and **(b)** simplified regression models. The unit of SF (ions $g^{-1}$ $s^{-1}$) indicates the number of $SO_4^{2-}$ formed on the particle sample of per unit mass within per unit time, and $R_{light/dark}$ indicates the ratio of SF values in presence and absence of simulated solar irradiation.

## 3.2 Driving force of dust surface

The driving force of dust surface can be characterized by reactive uptake coefficient and particle acidity (Fig. 3, Table S8). Under dark condition, the $\gamma$ values are highest for SWy and NAu, followed by IMt and CCa, with that of KGa being the lowest. The presence of simulated solar irradiation causes a different rank: IMt>SWy>CCa>NAu>KGa. The $\gamma$ values of CCa, SWy, KGa, IMt under dark condition are respectively 1.52, 1.01, 2.94, 2.30 times greater than those for photoreactions, reflecting the distinct photocatalytic performances of the clay minerals. Conversely, NAu presents the decreased heterogeneous uptake capacity when exposed to solar irradiation due to its rich abundance of Fe, whose oxides may occur photoreductive dissolution in acidic media (Fu et al., 2010; Shi et al., 2012). Comparing the studies performed by the same experimental approach and assessment procedures, these clay minerals are more efficient than the previously concerned mineral dust proxies, including $CaCO_3$ (Li et al., 2006; Wu et al., 2011; Zhang et al., 2018), $Al_2O_3$ (Liu et al., 2017b), $TiO_2$ (He and Zhang, 2019) and manganese oxides (Wang et al., 2020d), in the heterogeneous production of sulfate under the parallel conditions.

The $\gamma$ values for the natural dust-driven heterogeneous reactions are calculated to be $6.08 \times 10^{-6}$ under dark condition and $1.14 \times 10^{-5}$ under illuminated condition (Fig. 3a). Within the data set of authentic dust, the $\gamma$ for dark reaction is lower than the reported uptake coefficients of China loess ($3.0 \times 10^{-5}$), Inner Mongolia desert dust ($2.41 \times 10^{-5}$), Xinjiang sierozem ($8.34 \times 10^{-5}$), Saharan dust ($6.6 \times 10^{-5}$), Asian mineral dust ($2.54 \times 10^{-5}$), Tengger desert dust ($4.48 \times 10^{-5}$) and ATD ($1.92 \times 10^{-5}$) under the similar experimental conditions (Usher et al., 2002; Adams et al., 2005; Zhou et al., 2014; Huang et al., 2015). It should be noted that, the previous studies measured net uptake coefficient that quantifies all the heterogeneously adsorbed $SO_2$, and some of them assessed the oxidations accelerated by surface adsorbed oxidants (e.g. $NO_2$, $O_3$, $H_2O_2$). In order to quantify the driving force of dust surface in the contribution of particle-phase sulfate, $S_{(IV)}$ species were not involved in the current kinetics calculation. Quantitatively, the dust surface drivers are responsible for the atmospheric sulfate formation rates of 0.195 and 0.365 $\mu g\ m^{-3}\ h^{-1}$ during nighttime and daytime, respectively.

Particle acidity is further calculated to discuss the driving force of dust surface for sulfate formation. After the $SO_2$ exposure, CCa is the most acidic, followed by IMt, KGa and NAu, leaving SWy being more neutral. Because no significant correlation can be found between the $\gamma$ for dust-driven sulfate formation and the pH of reacted clay mineral, the absolute acidity level depends largely on the basic nature of dust. The lowest pH assigned to CCa can be explained by its highest content of elemental Mg relative to the other clays (see Table S6). The Mg-bearing constituents dissolve to be $Mg^{2+}$ that can be further hydrolyzed by water to produce $H^+$, thus accelerating the acidification of particle (Park et al., 2008; Huang et al., 2015). For one clay mineral, its different acidities after dark and light reactions reflect the distinct heterogeneous kinetics. All the studied clays, with the exception of NAu, become more acidic after the photoreaction than the dark reaction, which can be explained by the photoinduced $SO_2$ adsorption and sulfate production on these samples. The opposite situation of NAu coincides to the decreased heterogeneous kinetics over its surface by the presence of solar irradiation. Generally, the natural dust presents the particle acidity (pH) of 4.18 after dark reaction and 4.41 after photoreaction (Fig. 3b). Such results locate within the acidity ranges of dust seeds (3.0-7.0) (Ault, 2020; Pye et al., 2020) and haze aerosols (3.5-4.8) (Ding et al., 2019; Song et al., 2019),

suggesting that the dust-driven heterogeneous pathway could affect aerosol acidity by a certain extent.

Figure 4 presents the pH-dependent $\gamma$ for natural dust-mediated heterogeneous pathway, along with the experimental data of natural dust-driven heterogeneous pathway. The dust-mediated $\gamma$ is orders of magnitude lower than the $\alpha$ of $SO_2$ (~ 0.14 under the experimental temperature of 296.8 K). By definition, $\alpha$ is the probability that a $SO_2$ molecule striking a liquid surface finally enters into the liquid phase, whereas $\gamma$ is the sulfate formation rate normalized by the total surface collision rate of $SO_2$ (Jacob, 2000; Davidovits et al., 2006). That is, $\gamma$ involves all the uptake processes, including the mass accommodation at surface (Seinfeld and Pandis, 2016). Moreover, unlike the pH-independent $\alpha$, $\gamma$ varies with pH, somewhat coinciding to the evolution of dissolved sulfur species that can be quantitatively described by the H* of $SO_2$. The $\gamma$ values of $H_2O_2$, $CH_3OOH$ and $CH_3COOOH$ are pH-independent under specific conditions, in relation to the acid-catalyzed rate-limiting steps relevant to these peroxides (Lind et al., 1987; Liu et al., 2021a). Generally, the dust-mediated heterogeneous $\gamma$ is largely determined by the diffusion and reaction processes in the water layers of dust surface, as characterized by the second item on the RHS of Eq. (5).

The dust-mediated heterogeneous sulfate formation is primarily contributed by the surface adsorbed $H_2O_2$ below the nocturnal pH of 5.50 or the diurnal pH of 5.27. When the acidity exceeds the thresholds, the dust-mediated pathway would be kinetically dominated by $NO_2$ and $O_3$ during nighttime, or HOBr and HOCl during daytime. Comparing the $\gamma$ values with the same acidity, the dust-driven pathway appears to be more efficient than the dust-mediated one, thus having the possibility to account for more secondary sulfate aerosols. In the following sections, the oxidation of $SO_2$ mediated or driven by the natural dust would be set as the typical dust-mediated and dust-driven heterogeneous reactions to compare with the widely documented gas- and aqueous-phase pathways.

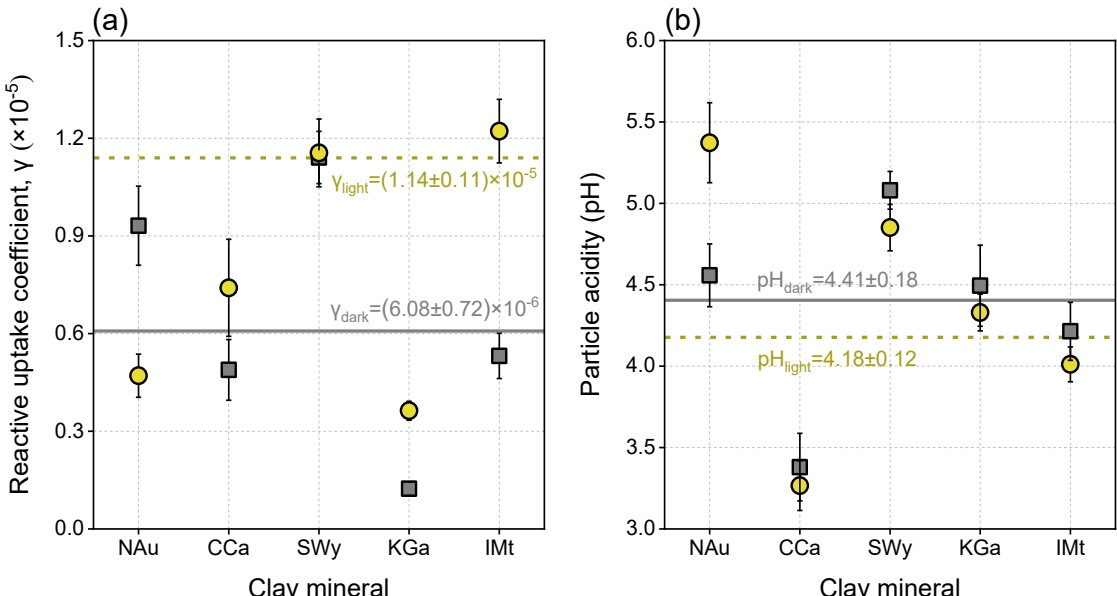

**Figure 3.** Analysis results of the *in-situ* infrared spectra recorded for the heterogeneous reaction of $SO_2$ on clay minerals and natural dust. **(a)** Reactive uptake coefficients ($\gamma$) for the heterogeneous formation of sulfate. **(b)** Particle acidity (pH) of the reacted particle samples. The dark (grey square) and light (yellow circle) conditions were both considered. Dots represent the results of clay minerals, and those of natural dust are showed by the lines. All error bars represent 1 SD.

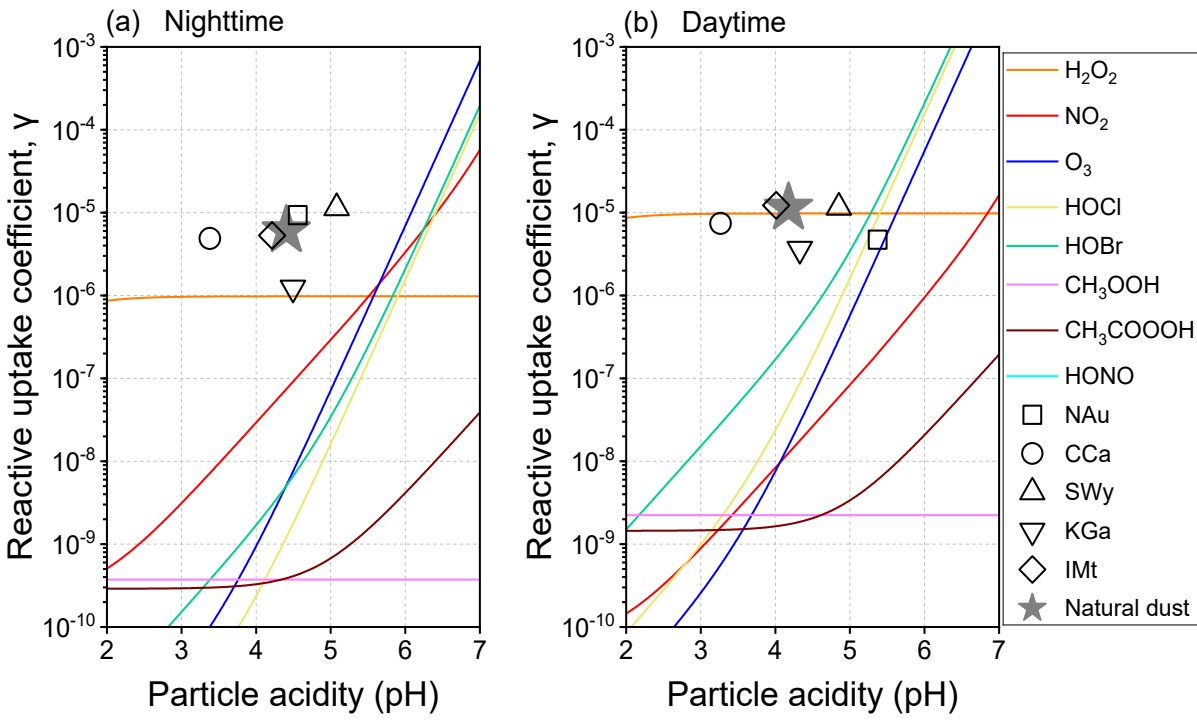

**Figure 4.** Particle acidity-dependent reactive uptake coefficients (γ) for the dust-mediated and dust-driven heterogeneous pathways. The dust-mediated pathway can be induced by the surface oxidants co-adsorbed with $SO_2$, including hydrogen peroxide ($H_2O_2$), nitrogen oxide ($NO_2$), ozone ($O_3$), hypochlorous acid (HOCl), hypobromous acid (HOBr), methyl hydroperoxide ($CH_3OOH$), peroxyacetic acid ($CH_3COOOH$) and dissolved nitrous acid (HONO). The dust-driven pathway can be induced by the heterogeneous drivers (transition-metal-bearing components and water-soluble ions) on natural dust and clay minerals [Nontronite (NAu), Chlorite (CCa), Montmorillonite (SWy), Kaolin (KGa), Illite (IMt)].

### 3.3 Comparison of atmospheric oxidation pathways

Figure 5 compares the sulfate formation rates of diverse atmospheric oxidation pathways by a newly developed comparison model. Based on the parameterization scheme in model and the experimental results discussed above, the total sulfate formation rates are summed to be 0.795 $\mu g\ m^{-3}\ h^{-1}$ during nighttime and 5.179 $\mu g\ m^{-3}\ h^{-1}$ during daytime, under the acidity of natural dust (Fig. 5a and e). The higher sulfate flux during daytime can be explained by the photo-increased oxidation channels, oxidant concentrations, heterogeneous reactivity, and the elevated particle acidity that facilitates the TMI catalysis. It is worthwhile to

mention that, the nocturnal sulfate concentration was reported to approach or exceed the subsequent diurnal level, as explained by the higher nocturnal humidity facilitating the liquid oxidations or the lower boundary layer at night causing the adverse diffusion conditions (Liu et al., 2017c; Tutsak and Koçak, 2019; Li et al., 2020c). Meteorological factors like humidity are not considered by the current model, which emphasizes the comparison of diverse pathways through kinetic regime. The estimated sulfate fluxes are generally lower than some published data derived from the same concentration of $SO_2$ (40 ppb) because the

gas- and aqueous-phase parameters here are corrected by the experimental temperature (296.8 K) rather than the much lower level (271 K) used previously (Cheng et al., 2016; Gen et al., 2019; Liu et al., 2020a; Su et al., 2020; Wang et al., 2020a; Liu et al., 2021a). Considering that some kinetic parameters were experimentally determined under room temperature with the lack of temperature dependence, uncertainties may exist in the discussion on cold environments. The current sulfate formation is assessed near room temperature that is pertinent to the nature of sandstorm occurring during late spring and early summer

in East Asia (Wu et al., 2020; Ren et al., 2021).

During nighttime, the gas-phase, aqueous-phase and heterogeneous pathways explain 31.6, 39.8 and 28.6% of secondary sulfate, respectively (Fig. 5b). The diurnal contribution proportions of gas-phase (45.5%) and aqueous-phase pathways (41.4%) exceed their nocturnal levels, thus lowering the heterogeneous proportion to 13.1% during daytime (Fig. 5f). Although the heterogeneous sulfate yield during nighttime is lower than that during daytime (see Sect. 3.2), nocturnal heterogeneous process

accounts for the higher proportion of secondary sulfate. The diurnal sulfate formation rates of gas-phase, aqueous-phase and heterogeneous pathways are respectively 9.4, 6.8 and 3.0 times greater than the nocturnal levels, indicating that the oxidations in gaseous and liquid media could be more kinetically susceptible to the occurrence of sunlight than those relevant to the humidified gas-solid interface. In gas-phase oxidation, OH is the predominant oxidant, followed by CIs, while $NO_3$ contributes little. In aqueous-phase oxidation, $TMI-O_2$ and $H_2O_2$ play crucial roles in the conversion of $SO_2$, coinciding to the reported

results (Fan et al., 2020; Song et al., 2021). Besides, there are lower contributions from the $NO_2$ during nighttime and the HOBr, T*, $P_{NO_3^-}$ during daytime. The heterogeneously formed sulfate is mostly ascribed to the dust-driven pathway rather than the dust-mediated one. In comparison, the sulfate contribution proportions of the studied dust heterogeneous oxidation are comparable with those obtained by the OBM model (30.6% for nighttime and 19.4% for daytime) (Xue et al., 2016), and the revised GEOS-Chem model (20-30%) (Tian et al., 2021) and WRF-CMAQ model (up to 12%) (Wang et al., 2012). While

the uptake coefficients used in the previous studies are generally greater than the present experimental results, more pathways are implemented in the current comparison model, thereby causing the parallel comparison results. By contrast, the AMAR

model highlighted the dust's photocatalytic surface that contributes remarkable secondary sulfate (>50%) under the constraint simulation conditions (Yu et al., 2017).

In dust-mediated heterogeneous oxidation, $H_2O_2$ is the most efficient oxidant, followed by the nocturnal $NO_2$ and diurnal hypohalous acids (HOBr, HOCl) responsible for less secondary sulfate (Fig. 5c, g). The dust-driven heterogeneous sulfate formation is mainly attributed to IMt that owns the largest proportion in dust community, followed by NAu and SWy with relatively significant contributions by heterogeneous sulfate formation (Fig. 5d and h). The nocturnal and diurnal sulfate fluxes of natural dust-driven heterogeneous pathway are respectively 5.8 and 1.2 times greater than those of the natural dust-mediated one. Hence, the heterogeneous sulfate formation is primarily ascribed to the dust surface drivers rather than the surface adsorbed oxidants, as experimentally proved by the laboratory studies concerning the heterogeneous $SO_2$ oxidation on authentic dusts accelerated by the presence of $NO_2$ (Park et al., 2017), or $O_3$ (Park et al., 2017), or $H_2O_2$ (Huang et al., 2015). The kinetic discrepancy between the dust-mediated and dust-driven heterogeneous pathways is more significant during nighttime than daytime. In the estimation, $H_2O_2$ is the predominant dust-mediated oxidant and its concentration becomes lower under weaker sunlight. The dust-mediated contribution during nighttime is thus relatively small by the presence of the relatively low $H_2O_2$ concentration. In general, when investigating the heterogeneous process on dust particles, particle variables could be more important than gas variables in elucidating the reaction characteristics and atmospheric implications.

Figure 6 summaries the particle acidity-dependent sulfate formation rates of gas-phase, aqueous-phase and heterogeneous pathways. Relative to the pH-independent gas-phase process, the aqueous-phase oxidation becomes relatively productive under the extremely acidic and near-neutral situations, while fails to support the rapid sulfate formation within the pH range of 4-5 (minimums at 4.85 for nighttime and 4.55 for daytime), which overlaps the acidity range of the weak dust-mediated heterogeneous pathway. The significant sulfate formation in acidic medium is primarily contributed by the TMI-$O_2$ pathway where the concentrations of TMIs and $H^+$ keep increasing as the aerosol water becomes more acidic. When pH>5.0, the sulfate formation rate increases with elevated pH, in step with the evolution of the concentration of dissolved sulfur species. Noticeably, the pH range of 4-5 involves the acidity of the aged natural dust and overlaps that of the common haze aerosols, implicating that the dust surface drivers have profound impacts on the secondary sulfate burst in highly polluted environment (Ding et al., 2019; Song et al., 2019). The important role of dust in sulfate formation was confirmed by atmospheric observation research. For example, secondary sulfate was observed to accumulate on the dust-dominant super-micron particles collected in the North China Plain, and the mass fraction of coarse-mode sulfate dramatically increased during the evolutionary stages of haze episode (Xu et al., 2020). Within the entire pH range, the aqueous-phase pathway is more active than the dust-mediated heterogeneous one in contributing to secondary sulfate.

Figure 7 compares the lifetimes of $SO_2$ influenced by the diverse atmospheric oxidation pathways. Calculations of lifetimes can be useful in estimating how long the $SO_2$ is likely to remain airborne before it is removed from the atmosphere (Seinfeld and Pandis, 2016). Theoretically, the lifetime of $SO_2$ determined by heterogeneous reaction is negatively correlated with $\gamma$ and $S_p$, and $S_p$ is positively associated with the concentration and $S_{BET}$ of dust, as respectively described by Eq. (12) and (10). As a result, the dust with greater heterogeneous reactivity, or higher atmospheric loading, or larger $S_{BET}$ is prone to cause the

shorter lifetime. During nighttime, IMt causes the shortest $SO_2$ lifetime (46.90 days) due to its highest concentration relative to the other clay minerals. The relatively large heterogeneous uptake capacity of NAu and SWy link to the second and third shortest $SO_2$ lifetimes (221.30 and 260.12 days, respectively). On the other hand, the weakest heterogeneous reactivity of KGa leads to the second longest lifetime (1758.65 days), and the longest result caused by CCa (2256.25 days) can be interpreted by its lowest $S_{BET}$ that causes the lowest Sp. The presence of solar irradiation alters the lifetime ranking: IMt (20.39 days) < SWy (256.96 days) < NAu (437.85 days) < KGa (597.54 days) < CCa (1488.46 days), as influenced by the different photoactivities of the clay minerals. The heterogeneous drivers of natural dust surface are comparable with TMI-$O_2$ in the loss of $SO_2$, and both of them function as the most important lifespan influencers in the absence of sunlight (Fig. S9a). During daytime, the natural dust-driven pathway is only next to the oxidations induced by OH, TMI-$O_2$ and $H_2O_2$ (Fig. S9b). The comparison results of lifetime agree well with those of sulfate formation rate (Fig. 5), illustrating that the heterogeneous drivers of dust surface are somewhat responsible for altering the concentrations of gas-phase and particulate sulfur species.

Gas- and aqueous-phase pathways serve as the most significant influencers of the diurnal lifespan of $SO_2$, followed by dust-driven heterogeneous pathway, while during nighttime these three present closer impacts (Fig. S10). The atmospheric lifetimes of $SO_2$ induced by the considered atmospheric oxidation pathways are calculated to be 6.17 days during nighttime and 0.99 days during daytime. Neglecting dust heterogeneous pathways may lengthen the $SO_2$ lifespans to 10.27 and 1.12 days in the absence and presence of solar irradiation, respectively. Analogously, scientists obtained the shorter lifetime of $SO_2$ from climate model after considering the heterogeneous reaction occurring on volcanic ash particles (Zhu et al., 2020). Clay minerals are more concentrated in the troposphere than volcanic ash and thus have more significant impacts on the removal of atmospheric $SO_2$.

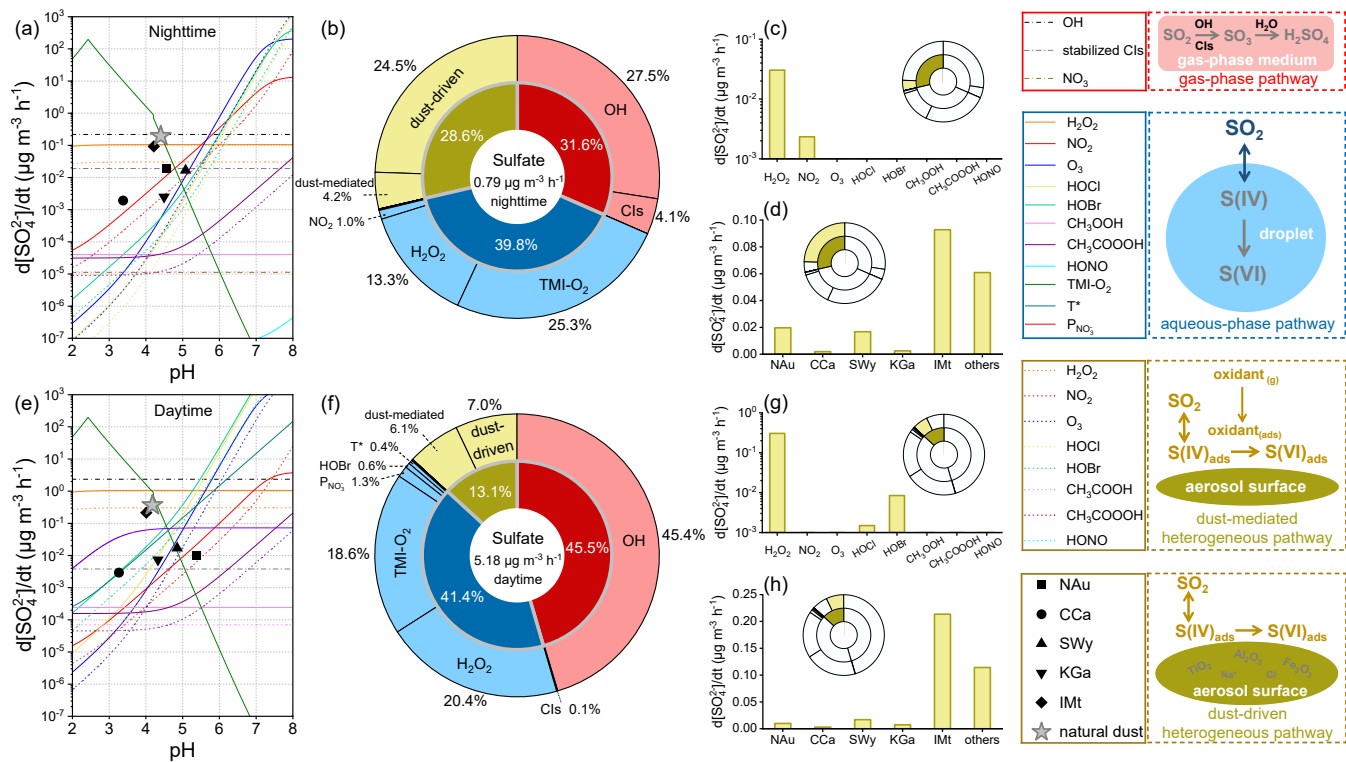

**Figure 5.** Contributions of diverse atmospheric oxidation pathways to secondary sulfate aerosols.

Gas-phase oxidation can be induced by hydroxyl radical (OH), stabilized Criegee intermediates (CIs), as well as the nitrate radical (NO₃) only for nighttime. Aqueous-phase oxidation can be induced by hydrogen peroxide (H₂O₂), nitrogen oxide (NO₂), ozone (O₃), hypochlorous acid (HOCl), hypobromous acid (HOBr), methyl hydroperoxide (CH₃OOH), peroxyacetic acid (CH₃COOOH), dissolved nitrous acid

(HONO), transition-metal ion-catalyzed oxygen (TMI-O₂), and the photosensitization (T*) and nitrate photolysis (P$_{NO_3^-}$) only for daytime. Dust-mediated heterogeneous oxidation can be initiated by the surface oxidants (H₂O₂, NO₂, O₃, HOCl, HOBr, CH₃OOH, CH₃COOOH, HONO) co-adsorbed with SO₂. Dust-driven heterogeneous oxidation can be ascribed to the heterogeneous drivers (transition-metal-bearing components and water-soluble ions) on the surfaces of natural dust and clay minerals [Nontronite (NAu), Chlorite (CCa), Montmorillonite (SWy), Kaolin (KGa), Illite (IMt)]. The **(a-d)** nighttime and **(e-h)** daytime conditions were distinguished by the different parameterizations.

**(a, e)** Particle acidity-dependent sulfate formation rates of the diverse atmospheric oxidation pathways. The elements' colors and shapes are characterized by the legends in solid boxes. **(b, f)** Quantified sulfate contribution proportions of the studied gas-phase (red), aqueous-phase (blue), and heterogeneous (yellow) reaction pathways. Sulfate formation rates of the dust-mediated and dust-driven pathways during **(c-d)** nighttime and **(g-h)** daytime. The effects of ionic strength on the aqueous-phase oxidation were not taken into account. The dust concentration was set to be 55 µg m⁻³, in representative of the common atmospheric condition of North China (Zhang et al., 2012). The panels in dashed

boxes right to the legends illustrate the primary physical-chemical processes of atmospheric sulfate formation. More parameterization and methodology details can be found in the Texts S2-S4 of Supporting Information and Sect. 2.1-2.5 of the main content.

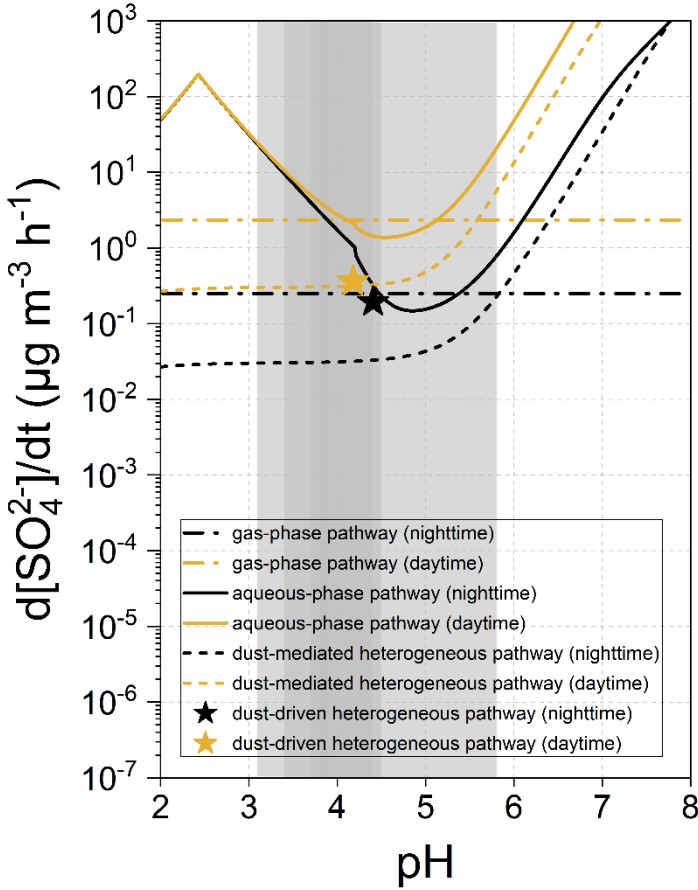

**Figure 6.** Formation rate of sulfate attributed to the gas-phase, aqueous-phase and dust heterogeneous pathways as a function of particle acidity (pH).

Grey areas indicate the pH ranges of the polluted particulate matters, with darker ones being more common (Ding et al., 2019).

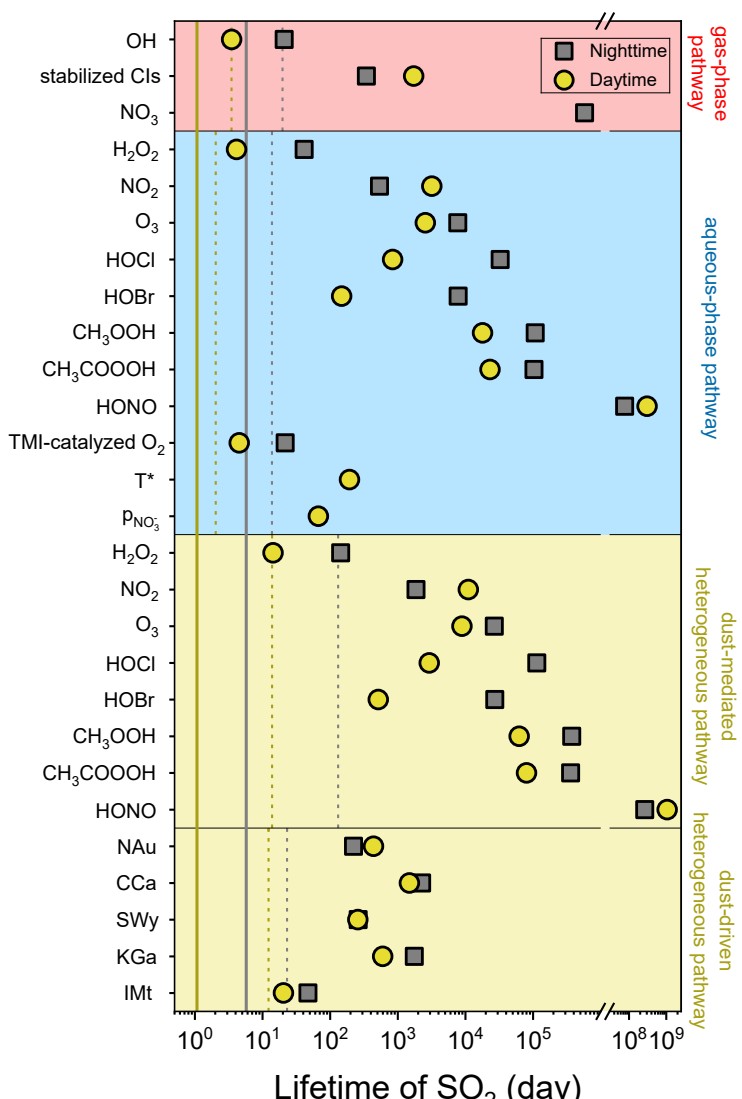

**Figure 7.** Atmospheric lifetimes of $SO_2$ induced by the diverse sulfate formation pathways.

Both nighttime (grey square) and daytime (yellow circle) conditions were considered. The lifetimes of $SO_2$ caused by gas-phase, aqueous-phase and dust heterogeneous pathways are displayed by the dashed lines. The atmospheric lifetimes of $SO_2$ induced by all the studied oxidation pathways are presented by the solid lines. The effects of ionic strength on the aqueous-phase $SO_2$ oxidation were not taken into account. The dust concentration was set to be 55 µg m$^{-3}$ to reflect the common atmospheric condition of North China. More methodology details can be found in Sect. 2.6 of the main content.

## 3.4  Sensitivity analysis

The aforementioned calculations set the concentration of natural dust to be 55 μg m$^{-3}$. In contrast to the common atmospheric loading, the burst of sandstorm was normally accompanied by the quickly elevated dust concentration up to thousands of μg m$^{-3}$ (Li et al., 2021a; Yin et al., 2021; Filonchyk, 2022). It would be meaningful to estimate the heterogeneous contributions in the dust-rich environments. Theoretically, dust concentration within the ranges of 72-770 μg m$^{-3}$ during nighttime and 24-260 μg m$^{-3}$ during daytime could cause the extra sulfate formation of 0.3-3.0 μg m$^{-3}$ h$^{-1}$ (Fig. 8a), in line with the acknowledged range of missing sulfate formation rate (Cheng et al., 2016; Liu et al., 2020a), which was found to be positively correlated with PM concentration as well (Cheng et al., 2016). Therefore, the heterogeneous reaction of SO$_2$ on dust surface is a considerable sulfate formation pathway and may evolve into the missing sulfate source in the atmosphere.

The occurrence of sandstorm, particularly during nighttime, aggravates the sulfate pollution in coarse aerosol mode (Fig. 8b). For instance, the dust concentration of 200 μg m$^{-3}$, which approaches the PM$_{10}$ level in North China on March 2021 (Yin et al., 2021), could heterogeneously explain 44.9% of the secondary sulfate during nighttime, as well as 29.6% during daytime. It is worthwhile to note that, the heterogeneous contribution proportion is susceptible to the evolution of relatively low dust concentration, further increase of dust loading will not significantly elevate the heterogeneous proportion, resulting in a plateau (nighttime) or decrease (daytime) under severe dust pollution. In fact, the increased dust concentration facilitates the aqueous-phase TMI-O$_2$ pathway and dust-mediated and dust-driven heterogeneous reactions, whereas presents negative impacts on the others by the removal of gaseous oxidants over dust surface. The unsusceptible response to dust concentration, as shown by Fig. 8b, is related to the increased sulfate contributions from TMI-O$_2$. Unlike the TMI-derived oxidation and dust-driven pathway receiving constant positive feedbacks from the increased dust concentration, the dust-mediated pathway is somewhat affected by the decreased oxidants adsorbed on dust surface. Since the importance of dust-mediated pathway is more significant during daytime than nighttime, there is a negative correlation between the high dust concentration and heterogeneous contribution proportion in the presence of solar irradiation.

The increased dust concentration, in fact, not only facilitates the heterogeneous process by providing more reactive surfaces, but also affects the gas- and aqueous-phase reactions by altering the atmospheric abundances of reactive species, as explained below. The evolution of dust pollution from slight to heavy conditions would cause the loss of various gaseous oxidants by heterogeneous uptake, and therefore the gas- and aqueous-phase sulfate fluxes, except that induced by TMI-O$_2$, decrease against dust concentration (Bian and Zender, 2003; Tang et al., 2017). Furthermore, the dissolution of mineral constituents produces TMIs in aerosol liquid media (Alexander et al., 2009; Shao et al., 2019), and the irradiated mineral dust was reported to emit gaseous ROS by surface photocatalysis (Dupart et al., 2012; Chen et al., 2021). Herein, the studied oxidation pathways are distinguished by their contribution proportions (Fig. 8c and d). The contribution proportions of OH and H$_2$O$_2$ decrease against dust loading, whereas the importance of TMI-derived oxidation and dust-driven pathway become more significant as the dust concentrates. The contribution proportion assigned to dust-mediated process increases with dust concentration at first and then decreases. While the increased dust loading provides more physical space for the dust-mediated reactions, the

simultaneously decreased gas-phase oxidants restrain the accumulation of particle-phase oxidants.

Atmospheric lifetime of $SO_2$ is also affected by the concentration of dust. As shown by Fig. 9a, the lifespan of airborne $SO_2$ during nighttime is higher than that during daytime, and both of them decrease against dust concentration. Analogous to the heterogeneous contribution proportion (Fig. 8b), the lifespan of $SO_2$ is more susceptible to the variation of dust concentration in clean and slightly polluted environments than that under heavily polluted conditions. The mild dust pollution, especially its level variations, should be paid more attention. The heterogeneous loss of $SO_2$ by dust surface was normally evaluated against the gas-phase loss by OH (Ullerstam et al., 2003; Adams et al., 2005; Li et al., 2006; Huang et al., 2015; Ma et al., 2018). Current estimation indicates that the airborne dust with the concentration of 45 $\mu g\ m^{-3}$ during nighttime, or 91 $\mu g\ m^{-3}$ during daytime, can be regarded as comparable with OH in controlling the removal of $SO_2$ (Fig. 9b). Such dust concentrations are sometimes common in the troposphere, especially during the dust storm periods (Li et al., 2021a; Yin et al., 2021; Filonchyk, 2022) or near the dust source regions (Ke et al., 2022). Therefore, the heterogeneous loss of $SO_2$ by airborne dust surface may have a similar magnitude as the main gas-phase loss process and can be taken as an important sink for $SO_2$.

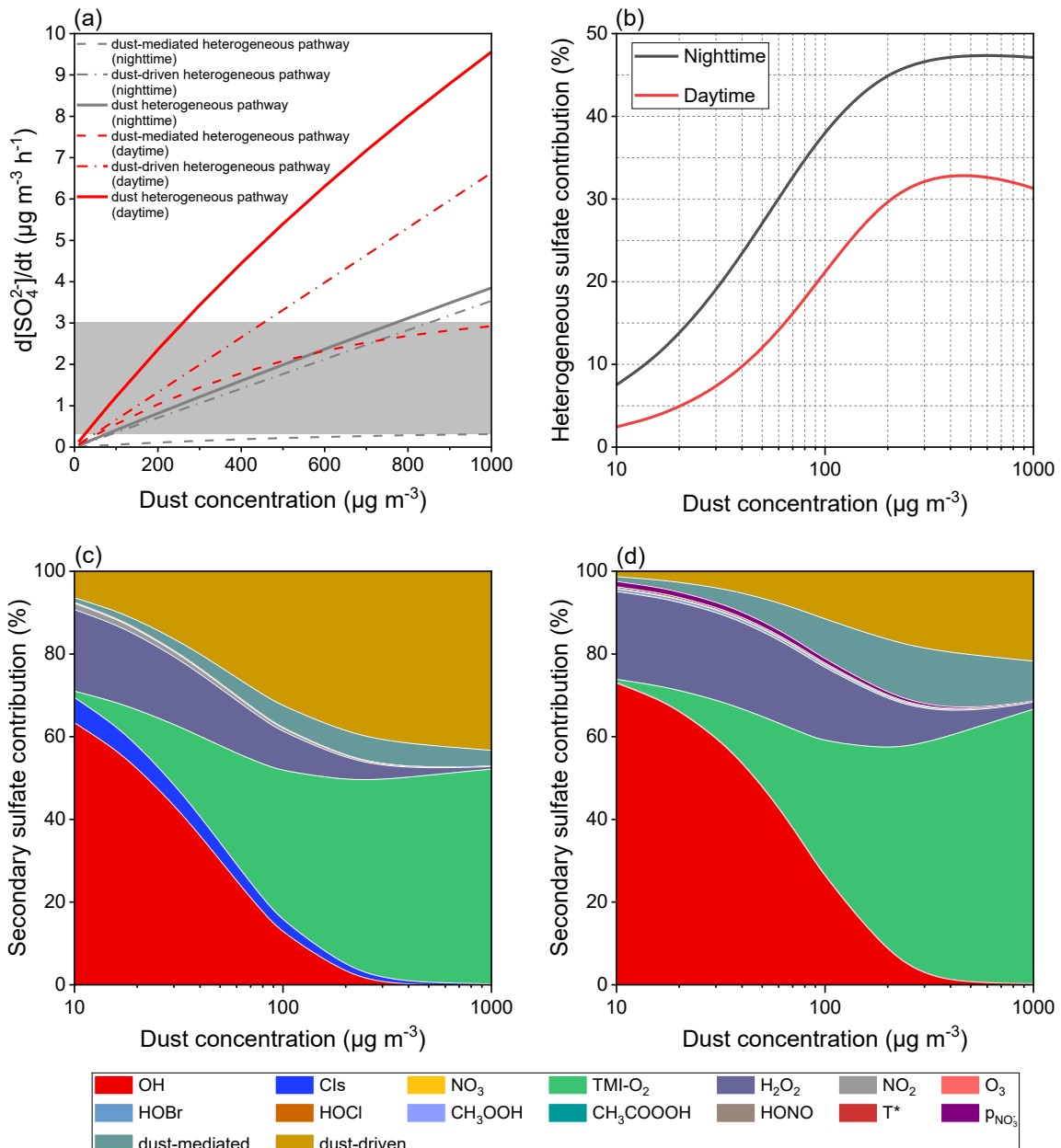

**Figure 8.** Sensitivity tests of sulfate formation rate to dust concentration.

**(a)** Sulfate formation rate of diverse dust-related heterogeneous pathways as a function of dust concentration. The grey area suggests the missing sulfate formation rate ranging from 0.3 to 3 μg m$^{-3}$ h$^{-1}$ as a reference (Cheng et al., 2016; Liu et al., 2021a). **(b)** Heterogeneous sulfate proportion varying with dust concentration. Secondary sulfate contributions attributed to the studied oxidation pathways during **(c)** nighttime and **(d)** daytime.

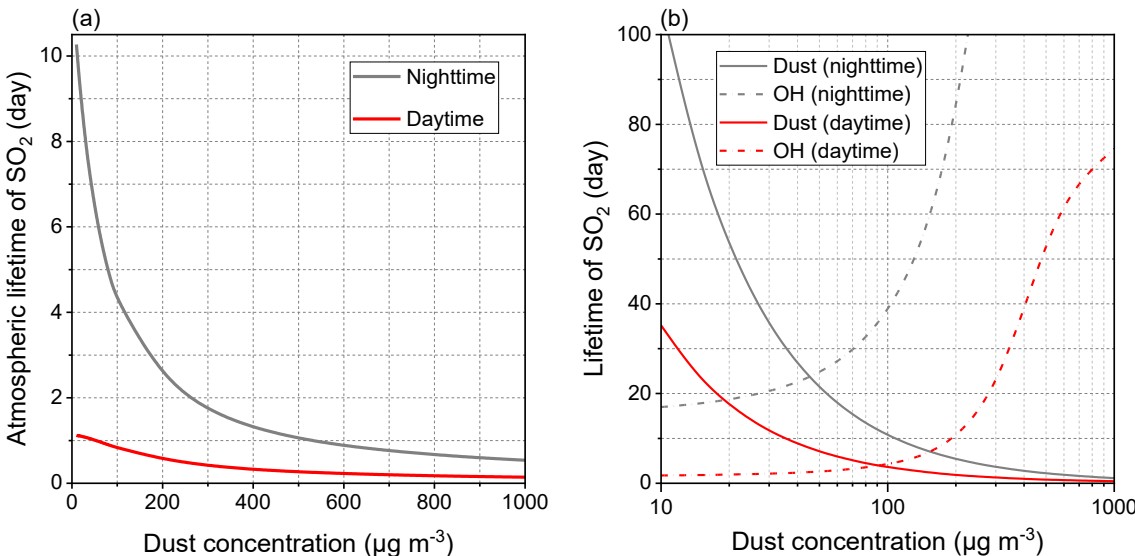

595

**Figure 9.** Sensitivity tests of SO₂ lifetime to dust concentration.

**(a)** Atmospheric lifetime of SO₂ induced by all the studied pathways varying with dust concentration. **(b)** Comparison between the SO₂ lifetimes ascribed to dust heterogeneous pathway and OH-initiated gas-phase oxidation.

## 3.5 Uncertainty analysis

The contribution proportion of dust heterogeneous oxidation could be over- or underestimated if considering the uncertainty factors. Herein, the joint impacts of ionic strength (I) and aerosol liquid water content (ALWC) on the aqueous-phase oxidation and the kinetic comparison between microdroplet interfacial oxidation and dust-driven heterogeneous oxidation are discussed to further understand the atmospheric relevance of the heterogeneous drivers on dust surface under the complex atmospheric conditions.

The aqueous-phase oxidation of $SO_2$ by $H_2O_2$, $O_3$, $NO_2$ and TMI-$O_2$ was quantified under different I-ALWC settings. At first, the sulfate formation rate was calculated as a function of ionic strength under the studied ALWC of 300 μg m$^{-3}$ (Fig. S11). TMI-$O_2$ and $H_2O_2$ dominate the liquid oxidation, while the impacts of $NO_2$ and $O_3$ only slightly peak at ~ 1.0 M during nighttime. Specifically, during nighttime, TMI-$O_2$ dominates the sulfate formation under the relatively low ionic strength, while the contribution of $H_2O_2$ exceeds that of TMI-$O_2$ over the ionic strength of 0.028 M. During daytime, $H_2O_2$ is the predominate oxidant within the studied ionic strength range. Relative to the ionic strength-free calculations, the aqueous oxidation could be weakened by the ionic strength lower than 17.8 M during nighttime or 14.3 M during daytime. Such values can be taken as criteria to distinguish the over- or underestimation of liquid kinetics under the ALWC of 300 μg cm$^{-3}$, which was widely used to characterize the haze events in North China (Cheng et al., 2016).

The joint influences of I and ALWC were further considered (Fig. 10). At each ionic strength, sulfate formation rate associates positively with ALWC. At each ALWC, the increase of ionic strength hinders the aqueous oxidation at first and then facilitates this process, as a consequence resulting in a threshold line distinguishing the negative or positive effects of ionic strength, as presented by Fig. 10a and b. Furthermore, the I-ALWC relationships of California, USA (Stelson and Seinfeld, 1981); Beijing, China (Song et al., 2021); Mexico City, Mexico (Volkamer et al., 2007; Hennigan et al., 2015); and the nine cities of Germany (Scheinhardt et al., 2013) were found to locate left to the thresholds, indicating the negative effect of ionic strength under the investigated scenarios (Fig. 10c). Calculated by the reported I-ALWC relationships, the dust-mediated pathway contributes 4.3-20.1% of the secondary sulfate during nighttime and 6.8-22.0% during daytime, and the dust-driven pathway accounts for respectively 29.1-41.6% and 9.9-12.4% of the sulfate formation in the absence and presence of sunlight (Fig. 10d). Therefore, the heterogeneous contribution proportions in the complex atmospheric environments can be even higher than those estimated by this study.

Besides the humidified gas-solid interface, gas-liquid interface of microdroplet is another type of medium that supports the heterogeneous formation of sulfate. The oxidation of $SO_2$ was found to proceed at the interfacial layer of a droplet with higher kinetics than the bulk process (Jayne and Davidovits, 1990). Recently, the interfacial roles of $O_2$ (Hung and Hoffmann, 2015; Hung et al., 2018; Chen et al., 2022), $NO_2$ (Liu and Abbatt, 2021; Yu, 2021) and $Mn^{2+}$ (Zhang et al., 2021; Wang et al., 2021a; Wang et al., 2022c) have been quantitatively described. Figure 11 compares dust-driven heterogeneous pathway with the documented gas-liquid interfacial oxidations. As reported by Hung et al. (2015, 2018), acidic droplet interface favors the noncatalyzed oxidation by the presence of sufficient $SO_3^{\cdot-}$ and $SO_4^{\cdot-}$ when pH<4.0, which can be further explained by the

structure differences of water at the interface and in the bulk phase. Recently, Chen et al. (2022) reevaluated the sulfate formation by interfacial $O_2$ within the pH range of 3.5-4.5 and discovered the positive dependence toward ionic strength. The oxidation of $SO_2$ by interfacial $NO_2$ displays the similar pH dependence as that occurs in bulk solution. The secondary sulfate contributed by interfacial $Mn^{2+}$ is associated positively with particle acidity. Generally, the $O_2$ at acidic interface dominates the oxidation when pH<4.0, whereas at pH>4.0 the interfacial oxidation is primarily controlled by $Mn^{2+}$. Comparing the sulfate formation rates under the same aerosol acidity, the dust surface drivers generally present greater reactivity than the interfacial $NO_2$ and $O_2$. Under the ionic strength of 40 M, the reactivity of interfacial $O_2$ can be regarded as equivalent to that of the dust surface drivers during nighttime. Interfacial $Mn^{2+}$ exceeds the dust surface drivers through kinetic regime. For instance, the $Mn^{2+}$-catalyzed oxidation characterized by Wang et al. (2021a) is 6.5 or 1.7 times more efficient than the dust-driven pathway in sulfate formation during nighttime or daytime, respectively.

The dust-driven heterogeneous pathway here was investigated by the infrared technique focusing on the bulk particle sample rather than air-suspended particles. Up to now, the micro-scale effects of dust particle have not been systematically studied. Suspended ATD was concerned by a smog chamber research and its heterogeneous reactivity toward $SO_2$ was characterized by the net uptake coefficient of $1.71 \times 10^{-6}$ under dark condition (Park and Jang, 2016). Such kinetic constant is approximately one order of magnitude higher than those obtained from the film-based flow tube tests ($\sim 1.75 \times 10^{-7}$) under the parallel experimental conditions (Zhang et al., 2019b). Therefore, in the atmosphere, dust surface drivers may kinetically approach the microdroplet interface in the removal of $SO_2$ and formation of sulfate.

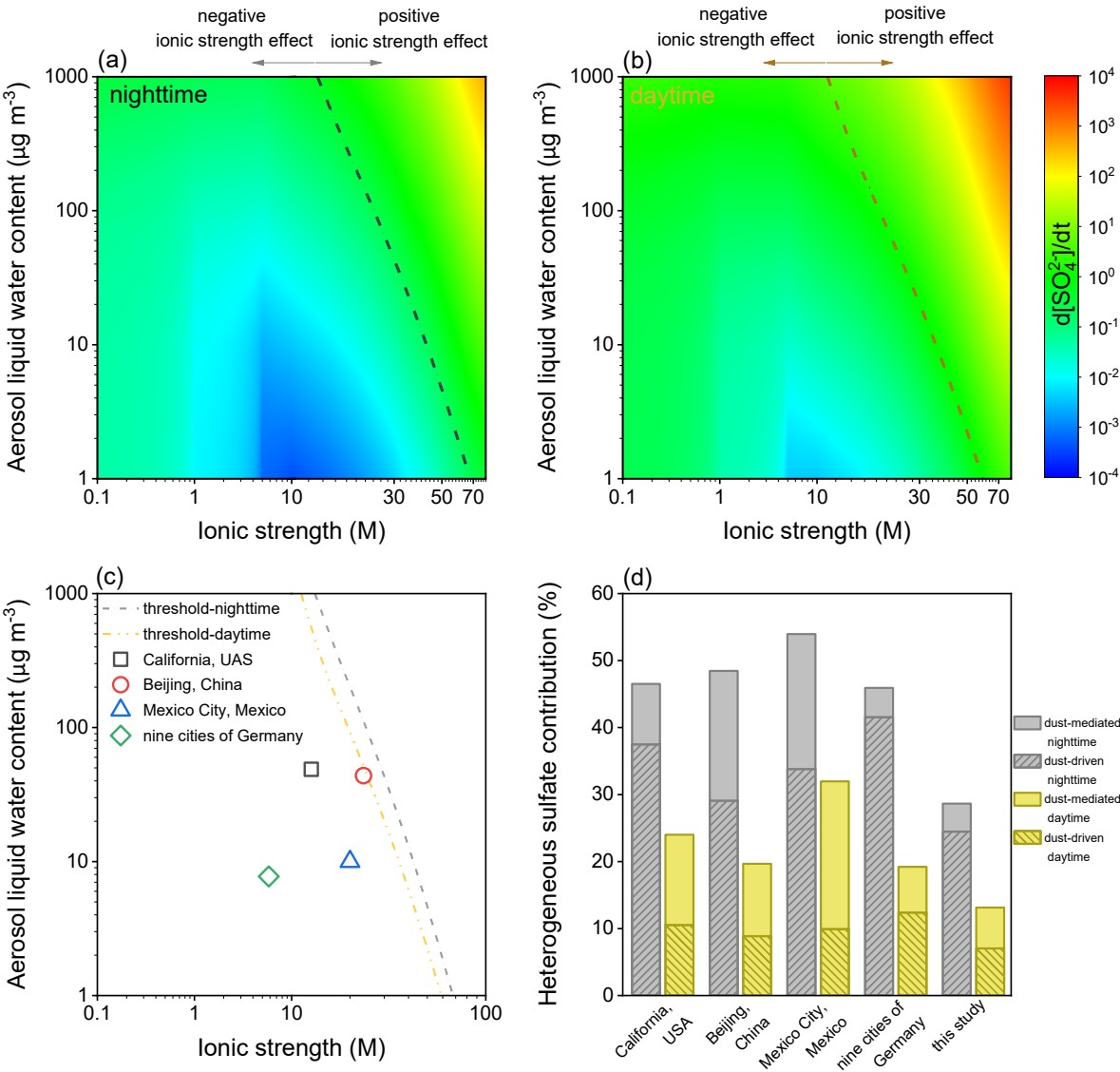

**Figure 10.** Joint influences of ionic strength (I) and aerosol liquid water content (ALWC) on the aqueous-phase oxidation of $SO_2$. Aqueous-phase sulfate formation rate varying with I and ALWC during **(a)** nighttime and **(b)** daytime. The dash-dotted lines indicate the thresholds distinguishing the negative and positive effects of ionic strength. **(c)** Reported I-ALWC relationships versus the nocturnal and diurnal thresholds. The field-observed date were collected from the measurements performed in California, USA (Stelson and Seinfeld, 1981); Beijing, China (Song et al., 2021); Mexico City, Mexico (Volkamer et al., 2007; Hennigan et al., 2015); and the nine cities of Germany (Scheinhardt et al., 2013). **(d)** Heterogeneous contribution proportions calculated by the reported I-ALWC relationships and parameterization of this study (ionic strength-free settings and an ALWC of 300 $\mu g\ m^{-3}$ for aqueous-phase oxidation; dust concentration of 55 $\mu g\ m^{-3}$ for heterogeneous oxidation).

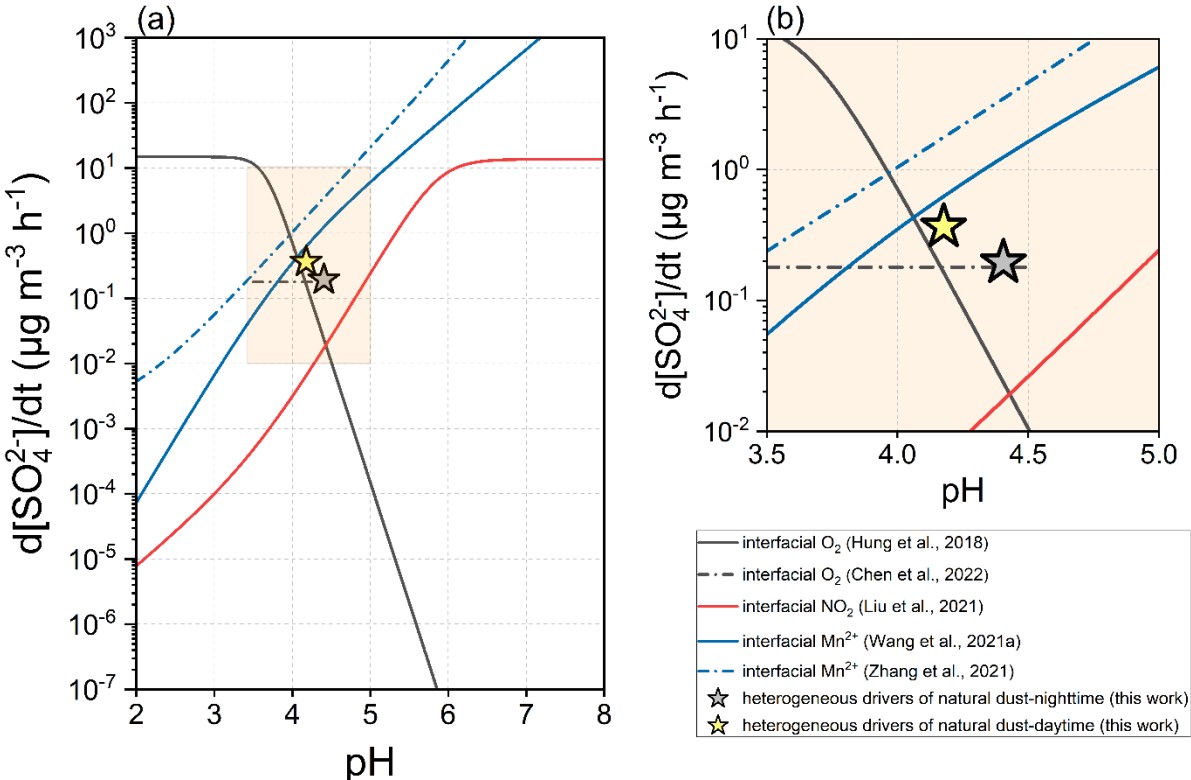

**Figure 11.** Kinetic comparison between microdroplet interfacial oxidation and dust-driven heterogeneous pathway. The interfacial oxidation of $SO_2$ can be induced by the $O_2$, or $NO_2$, or $Mn^{2+}$ at aerosol particle interfaces. The $O_2$-dominated oxidation can be assessed by the method concluded by Hung et al. (2018). Chen et al. (2022) refreshed the assessment method relevant to interfacial $O_2$, and the result here was obtained under the ionic strength of 40 M within the pH range of 3.5-4.5. The $NO_2$-dominated oxidation was assessed based on the work of Liu and Abbatt (2021). The $Mn^{2+}$-dominated oxidation can be assessed by the methods from Wang et al. (2021a) and Zhang et al. (2021). More parameterization and methodology details can be found in the Text S5 of Supporting Information.

## 4 Conclusions and implications

This study attempted to deeply understand the importance of heterogeneous oxidation, particularly that induced by dust surface drivers, in the loss of airborne $SO_2$ and formation of sulfate aerosols. Laboratory research was performed at first to investigate the heterogeneous driving factors and driving force. Based on the correlation and regression analysis, transition metal elements, particularly Fe for dark condition, and Al and Ti for photoreaction, were determined to dominate the heterogeneous oxidation, while water-soluble ions present minor influences. A series of empirical equations were developed to kinetically predict the dust-driven process. The Al, Fe, Ti-bearing mineralogical components, as well as their mixtures, are thus recommended as appropriate proxies for the following laboratory research. The dust-driven heterogeneous γ values were calculated to be 6.08 × 10⁻⁶ for dark reaction and 1.14 × 10⁻⁵ for photoreaction, corresponding respectively to the atmospheric sulfate formation rates of 0.195 and 0.365 μg m⁻³ h⁻¹ in the presence of 55 μg m⁻³ natural dust.

A comparison model was further developed to reveal the atmospheric relevance of the heterogeneous drivers on dust surface. Dust heterogeneous oxidation is suggested to explain 28.6% of the secondary sulfate aerosols during nighttime and 13.1% during daytime, and the dust surface drivers act as the dominate contributors. Furthermore, dust heterogeneous oxidation affects the atmospheric lifetime of $SO_2$. The increased dust concentration may aggravate the secondary sulfate pollution and has the potential to explain the acknowledged missing sulfate formation rate (0.3-3.0 μg m⁻³ h⁻¹). The joint effects of I and ALWC on liquid kinetics influence the importance of heterogeneous pathway. The heterogeneous contribution proportions estimated by the reported I-ALWC relationships are generally greater than those calculated by the parameterization in current comparison model. Additionally, dust heterogeneous pathway in the atmosphere is believed to be kinetically comparable with droplet interfacial oxidation.

Overall, this study suggests that the implementation of heterogeneous processes into atmospheric models shall vastly improve the agreement between the modeled and observed sulfate concentrations. Dust heterogeneous pathway should be treated as a significant contributor of secondary sulfate. More necessarily, dust surface drivers are needed to be viewed as an important research focus. Zheng et al. (2015) revised the CMAQ model by adding heterogeneous mechanism and observed the accurate simulation run upon determining the uptake coefficient to be 2.0 × 10⁻⁵, which is higher than the γ values of dust heterogeneous pathway herein (7.04 × 10⁻⁶ during nighttime and 1.55 × 10⁻⁵ during daytime). Hence, other solid aerosols, such as sea salts (Laskin et al., 2003; Rossi, 2003) and carbonaceous particles (He and He, 2020; Zhang et al., 2020), would better be taken into the following heterogeneous discussions together with the dust-related processes we studied.

Meteorological factors impact the atmospheric relevance of dust heterogeneous pathway. Taking humidity as an example, on dust surface, the heterogeneous reaction of $SO_2$ is humidity-dependent, and the exact dependence varies with the type of dust and the condition of reaction (Huang et al., 2015; Park et al., 2017; Urupina et al., 2022). The uptake of gas-phase oxidants over dust surface is also influenced by humidity (Kumar et al., 2014). For aerosol droplet, increased humidity elevates its liquid volume and radius (Wu et al., 2018; Ding et al., 2019). Aerosol liquid water serves as an efficient medium for multiphase

reactions (Wu et al., 2018; Yue et al., 2019), whereas radius is negatively associated with the sulfate formation at droplet interface (Hung et al., 2018; Wang et al., 2021a; Chen et al., 2022). Furthermore, droplet acidity decreases as the liquid volume increases, thereby increasing the ionization of dissolved sulfur species, followed by the increased sulfate formation rate (Yue et al., 2019; Jin et al., 2020; Gao et al., 2022). Overall, how meteorological factors influence the relative importance of gas-phase, aqueous-phase and heterogeneous pathways warrants further research.

Heterogeneous laboratory results were scarcely discussed together with other $SO_2$ conversion routes. This study attempted to set an example for kinetically comparing the heterogeneous oxidation characterized by laboratory work with the documented gas- and aqueous-phase data. Relative to the three-dimensional numerical models, the developed comparison model involves more oxidation pathways. Since the aqueous-phase oxidation here is relevant to aerosol droplet rather than cloud/fog droplet, the current parameterization could be more appropriate for simulating the fine particulate matters that were frequently collected by atmospheric observations and compared with modeling data. Furthermore, the heterogeneous oxidation was classified into dust-mediated and dust-driven modes to better distinguish the key surface impactors. Therefore, this comparison model has advantages over the traditional atmospheric chemistry models, and is recommended for the following heterogeneous laboratory research to systematically compare the experimental data with the acknowledged gas-phase/aqueous-phase/heterogeneous oxidation pathways.

This work broadens the application of infrared technique in atmospheric laboratory research. Apart from the uptake coefficient calculations normally concerned, this study moved forward to bridge the relationship between particle acidity and sulfate formation rate by analyzing the shape and intensity of infrared spectrum, and further compared heterogeneous oxidation with other atmospheric pathways. This research not only provides a promising methodology for the future heterogeneous research utilizing the classic *in-situ* DRIFTS approach, but also helps to take good advantages of the infrared technique in the laboratory studies in relation to atmospheric heterogeneous oxidation.

***Data availability.*** A dataset for this paper can be accessed at https://data.mendeley.com/datasets/hyvdz7khs6/1.

***Author Contributions.*** T.W. designed the experiments and processed the data, and wrote the paper together with Y.L; H.C. and Z.W. measured the physical and chemical properties of the particle samples; H.F., J.C., and L.Z. provided guidance in the data analysis and paper writing.

***Competing interests.*** The authors declare no competing financial interest.

***Financial support.*** The research was financially supported by National Natural Science Foundation of China (22176036, 21976030, 22006020, 42205099), Natural Science Foundation of Shanghai (19ZR1471200) and China Postdoctoral Science Foundation (2021M700792).

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
