# Peer review of "Significant formation of sulfate aerosols contributed by the heterogeneous drivers of dust surface"

_Atmospheric Chemistry and Physics, 2022_

## Author Comment (AC1)

This paper discusses the correlation of atmospheric mineral dust, and heterogeneous chemistry with the lifetime of $SO_2$ and would give a closer prediction of the formation of sulfate aerosols. Therefore this will be a good addition to our current knowledge in this field of atmospheric chemistry. I have a few comments and suggestions.

1. Abstract: I think we shouldn't say accurately predict - This is because it is still a prediction. Maybe with near accuracy, some rewording would be good.

Answer:

Thanks for your comments.

We have rewritten the sentence in the revised manuscript (line 13-14).

*"Regression models were then developed to make a reliable prediction for the heterogeneous reactivity based on the particle chemical compositions."*

2. Authors discuss at several points that the differences of dust surfaces can affect their heterogeneous reactivities. (Under driving factors of dust surface, line 265). Another addition to this section: $TiO_2$ and other titanium-bearing minerals such as ilmenite showed night-time thermal reactions that involve redox cycling with nitrates, as well as pH dependency.

Answer:

We have reconsidered the different heterogeneous performances of Ti-bearing constituent under the dark and light conditions. The following explanation has been added into the revised manuscript (line 279-281).

*"In addition, the heterogeneous reaction on the surface of $TiO_2$ or Ti-bearing mineral is thermodynamically favored during nighttime but largely dependent on photocatalysis under solar irradiation (Chen et al., 2012)."*

3. Figure 2: why do some minerals have lower sulfates in the daytime while the others have lower sulfates in the nighttime? Can this be explained with the change of particle pH? What is the pH of each sample before the reaction? Fig 2b shows only the pH after reactions.

Answer:

All of the studied clay minerals, with the exception of NAu, present greater sulfate production during daytime than nighttime, which can be explained by their photocatalytic performances (mainly attributed to the Al- and Ti-bearing components as discussed by Section 3.1). The opposite situation of NAu can be explained by its rich abundance of iron oxides that would undergo photoreductive dissolution in acidic media. We have emphasized this point in the

manuscript (line 331-335).

*"The diurnal uptake coefficients of CCa, SWy, KGa, IMt are respectively 1.52, 1.01, 2.94, 2.30 times greater than the corresponding nocturnal ones, reflecting the distinct photocatalytic performances of the clay minerals. Conversely, NAu presents the decreased heterogeneous uptake capacity under light irradiation than dark condition due to its rich abundance of Fe, whose oxide species may occur photoreductive dissolution in acidic media (Fu et al., 2010; Shi et al., 2012)."*

In addition, the association between the uptake coefficient for sulfate formation and the corresponding particle acidity has been discussed in the revised manuscript (line 355-359).

*"For one clay mineral, its different acidities after dark and light reactions correspond to the distinct heterogeneous kinetics. All of the studied clays, with the exception of NAu, become more acidic after the photoreaction than the dark reaction, which can be explained by the photoinduced $SO_2$ adsorption and sulfate production on these samples. The opposite situation of NAu coincides to the decreased heterogeneous kinetics over its surface by the presence of solar irradiation."*

The current methodology cannot estimate the acidity of the unreacted sample. The acidity of the reacted sample was derived from the relative abundance of $SO_2 \cdot H_2O$ and $SO_3^{2-}$, as assumed to be equivalent to the relative integral areas of their characteristic peaks. By contrast, the particle sample before reaction was scanned by infrared as a background spectrum, and therefore no infrared signals can be obtained for the unreacted particles.

4. Is chemistries a real word?

Answer:

As a real word, it was widely utilized in the research papers of this field (e.g. doi.org/10.1021/cr500501m; doi.org/10.1021/cr5003485).

5. Figure 3: why does the reactive uptake coefficient drastically increases after particle acidity reached ~4.5/ 5?

Answer:

Figure 3 displays the particle-acidity-dependent reactive uptake coefficients ($\gamma$) for the dust-mediated heterogeneous chemistry. We have discussed the dramatically increased $\gamma$ when pH exceeds 4.5 or 5.0, in the revised manuscript (line 367-369).

*"The $\gamma$ of $NO_2$, $O_3$, HOCl, HOBr and $CH_3COOOH$ presents positive dependence toward pH, in accordance with the evolution of the effective Henry's law constant for the studied sulfur species, as theoretically illustrated by Eq. (5)."*

6. For daytime chemistry, did you consider the possibility of the formation of sulfate radicals on surfaces with high Ti content?

Answer:

We have considered the impact of sulfate radical on the heterogeneous formation of sulfate. Necessary discussions are added into the revised manuscript (line 266-267).

*"Sulfate radical ($SO_4^{\cdot-}$) is generated by the presence of abundant $\cdot OH$ and participates into the oxidation events (Antoniou et al., 2018; Kim et al., 2019; Li et al., 2020b)."*

7. Figure 5: Why the kinetics has increased drastically after pH 5? Within the statistical error, do you see any difference between the aq phase and heterogeneous chemistry?

Answer:

We have explained the pH dependences of the aqueous-phase and dust-mediated heterogeneous kinetics in the revised manuscript (line 449-452).

*"In more detail, the significant sulfate formation in the acidic environment is primarily contributed by the TMI-$O_2$ pathway where the concentrations of TMIs and $H^+$ keep increasing as the aerosol water becomes more acidic. When pH>5.0, the formation rate of sulfate increases with the elevated pH, in step with the evolution of the concentration of dissolved sulfur species."*

Moreover, we have explained the kinetic difference between the aqueous-phase and dust-mediated chemistries. The addition to the revised manuscript (line 452-453) is shown as follows.

*"Within the entire pH range, the aqueous-phase chemistry is more efficient than the dust-mediated heterogeneous chemistry in contributing to secondary sulfate."*

8. Figure 7: Can you better label the y-axes to explain what % contribution? Panel b: why does the heterogeneous sulfate contribution for a daytime peak at a certain dust concentration and lowers, whereas it reaches a plateau at night time?

Answer:

We have re-labelled the y-axes of Fig. 7 as "Secondary sulfate contribution (%)".

In the revised manuscript (line 518-525), we have additionally explained the different responses to the concentration of dust, as shown follows.

*"The increased dust concentration facilitates the aqueous-phase TMI-$O_2$ pathway and the dust-mediated and dust-*

*driven heterogeneous reactions, whereas presents negative impacts on the others by the removal of gaseous oxidants over dust surface. The unsusceptible response to dust concentration, as shown by Fig. 7b, can be related to the increased sulfate contributions from the TMI-O$_2$ pathway. Unlike the TMI-O$_2$ pathway and dust-driven chemistry receiving constant positive feedbacks from the increased dust concentration, the dust-mediated chemistry is somewhat affected by the decreased oxidants adsorbed on dust surface. Since the importance of dust-mediated chemistry is more significant during daytime than nighttime, there is a negative correlation between the high dust concentration and heterogeneous contribution proportion under solar irradiation."*

9. Why is more SSA formed during nighttime, but more sulfate is formed during the daytime?

Answer:

We have noticed that the observed sulfate concentration during nighttime may be comparable to or even exceed that measured during daytime, in contrast to the calculated results derived from this work. Such phenomenon can be explained by the different meteorological conditions during nighttime and daytime in the real atmosphere. In the revised manuscript (line 396-401), we have discussed these uncertainties and emphasized the current comparisons through kinetic regime. The added discussion is shown as follows.

*"It is worthwhile to mention that, in the real atmosphere, the observed sulfate concentration during nighttime may be comparable to or exceed that during daytime, which can be explained by the higher nocturnal humidity facilitating the liquid oxidations or the lower boundary layer causing the adverse diffusion conditions (Liu et al., 2017c; Tutsak and Koçak, 2019; Li et al., 2020c). The relevant meteorological factors were not considered by this comparison model, and the current results emphasized the different sulfate formation potentials through kinetic regime.*

---

## Author Comment (AC2)

I have reviewed the manuscript "Significant formation of sulfate aerosols contributed by the heterogeneous drivers of dust surface" by Wang et al. Authors investigated the different metals and ions of the dust surface are associated with the heterogeneous formation of sulfate and used models to predict the heterogeneous reactivity. This study found that a significant amount of secondary sulfate aerosol was formed during nighttime and the heterogeneous drivers on the dust surface are more efficient to convert $SO_2$ to secondary sulfate. Given the importance of atmospheric secondary sulfate formation and heterogeneous chemistry of aerosols, this work is both significant and appropriate for this journal. However, there are some issues in this work that needs to be addressed before it can be published, as detailed below.

Introduction: authors should add some broad impacts of secondary sulfate aerosol as the motivation, such as the climate and health.

Answer:

Thanks for your comments.

We have discussed the relevant information at the start of Introduction section (line 25-27).

*"As an important component of atmospheric particulate matters, sulfate exerts profound impacts on the Earth's climate system, air quality, and public health (Seinfeld and Pandis, 2016; Wang et al., 2021a). The rapid formation of sulfate was proven as largely responsible for London Fog and Beijing Haze (Wang et al., 2016; Cheng et al., 2016)."*

Line 45: authors need to add relevant references "… mentioned in the previous works" and be more specific.

Answer:

We have rewritten the sentences to make the declaration more specific (line 47-49), as shown follows.

*"However, the "aerosol surface" mentioned in the previous works was not distinguished by its physical or chemical properties (Wang et al., 2014; Zheng et al., 2015; Xue et al., 2016), thereby making it difficult to compare the atmospheric importance of the diverse airborne surfaces."*

The following references are added.

Wang, Y., Zhang, Q., Jiang, J., Zhou, W., Wang, B., He, K., Duan, F., Zhang, Q., Philip, S., and Xie, Y.: Enhanced sulfate formation during China's severe winter haze episode in January 2013 missing from current models, J. Geophys. Res., 119, 10425-10440, http://doi.org/10.1002/2013jd021426, 2014.

Zheng, B., Zhang, Q., Zhang, Y., He, K. B., Wang, K., Zheng, G. J., Duan, F. K., Ma, Y. L., and Kimoto, T.: Heterogeneous chemistry: a mechanism missing in current models to explain secondary inorganic aerosol formation

during the January 2013 haze episode in North China, Atmos. Chem. Phys., 15, 2031-2049, http://doi.org/10.5194/acp-15-2031-2015, 2015.

Xue, J., Yuan, Z., Griffith, S. M., Yu, X., Lau, A. K. H., and Yu, J. Z.: Sulfate formation enhanced by a cocktail of high NOx, SO2, particulate matter, and droplet pH during haze-fog events in megacities in China: an observation-based modeling investigation, Environ. Sci. Technol., 50, 7325-7334, http://doi.org/10.1021/acs.est.6b00768, 2016.

Method Line 118: Can authors briefly describe the "documented methodologies" and add references.

Answer:

In the revised manuscript (line 122-124), we have briefly described the utilized methodologies and parameterizations.

*"The gas- and aqueous-phase pathways were assessed by the documented methodologies and parameterizations (Cheng et al., 2016; Seinfeld and Pandis, 2016; Shao et al., 2019; Song et al., 2021), as briefly introduced by Section 2.2 (more details in Supporting Information)."*

The following references are added.

Cheng, Y., Zheng, G., Wei, C., Mu, Q., Zheng, B., Wang, Z., Gao, M., Zhang, Q., He, K., Carmichael, G., Pöschl, U., and Su, H.: Reactive nitrogen chemistry in aerosol water as a source of sulfate during haze events in China, Sci. Adv., 2, e1601530, http://doi.org/10.1126/sciadv.1601530, 2016.

Seinfeld, J. H., and Pandis, S. N.: Atmospheric Chemistry and Physics, From Air Pollution to Climate Change, 3rd Edition, Wiley, New Jersey, USA, 2016.

Shao, J., Chen, Q., Wang, Y., Lu, X., He, P., Sun, Y., Shah, V., Martin, R. V., Philip, S., Song, S., Zhao, Y., Xie, Z., Zhang, L., and Alexander, B.: Heterogeneous sulfate aerosol formation mechanisms during wintertime Chinese haze events: air quality model assessment using observations of sulfate oxygen isotopes in Beijing, Atmos. Chem. Phys., 19, 6107-6123, http://doi.org/10.5194/acp-19-6107-2019, 2019.

Song, H., Lu, K., Ye, C., Dong, H., Li, S., Chen, S., Wu, Z., Zheng, M., Zeng, L., Hu, M., and Zhang, Y.: A comprehensive observation-based multiphase chemical model analysis of sulfur dioxide oxidations in both summer and winter, Atmos. Chem. Phys., 21, 13713-13727, http://doi.org/10.5194/acp-21-13713-2021, 2021.

Line 339-340: Can author provide more explanation about why the different clay minerals can affect the significant change of particle acidity?

Answer:

Firstly, the absolute acidity level varies with the type of clay mineral. The relevant discussions in the revised manuscript (line 350-355) is shown as follows.

*"After the exposure to $SO_2$, CCa is the most acidic, followed by IMt and KGa, leaving SWy and NAu being more neutral. Because no significant correlations can be found between γ and pH, the absolute acidity level is largely dependent on the basic nature of dust. The lowest pH assigned to CCa can be explained by its highest content of elemental Mg relative to the other clays (see Table S6). The Mg-bearing constituents dissolve to be $Mg^{2+}$ that can be further hydrolyzed by water to produce $H^+$, thus accelerating the acidification of particle (Park et al., 2008; Huang et al., 2015)."*

In addition, we discussed the different particle acidities after the dark and light reactions (line 355-359).

*"For one clay mineral, its different acidities after dark and light reactions reflect the distinct heterogeneous kinetics. All of the studied clays, with the exception of NAu, become more acidic after the photoreaction than the dark reaction, which can be explained by the photoinduced $SO_2$ adsorption and sulfate production on these samples. The opposite situation of NAu coincides to the decreased heterogeneous kinetics over its surface by the presence of solar irradiation."*

Line 343-343: It is very interesting to see the difference in particle acidity between dark reaction and photoreaction, are they statistically different? Can author discuss why the CCa has the lowest pH among all the different types of clay minerals?

Answer:

For a certain clay mineral, the particle acidity after dark reaction is statistically different from that after photoreaction, as reflected by the Mean±SD data presented in Fig.2b. We have added more details in the data processing section (line 187-188).

*"The consistent deconvolution procedure was performed for the infrared spectra derived from the repeated experiments."*

We have provided an explanation for the lowest pH of CCa. The added text in the revised manuscript (line 352-355) is shown as follows.

*"The lowest pH assigned to CCa can be explained by its highest content of elemental Mg relative to the other clays (see Table S6). The Mg-bearing constituents dissolve to be $Mg^{2+}$ that can be further hydrolyzed by water to produce $H^+$, thus accelerating the acidification of particle (Park et al., 2008; Huang et al., 2015)."*

Line 373-374: Earlier authors assert that the sulfate formation is more during nighttime and less during the daytime, but this statement is opposite and confusing. Can the author clarify this?

Answer:

In the real atmosphere, the sulfate concentration during nighttime could be comparable to or even exceed that during daytime, as normally explained by the different meteorological conditions. We have explained the relevant details in the revised manuscript (line 396-401) to make our calculation results more reasonable.

*"It is worthwhile to mention that, in the real atmosphere, the observed sulfate concentration during nighttime may be comparable to or exceed that during daytime, which can be explained by the higher nocturnal humidity facilitating the liquid oxidations or the lower boundary layer causing the adverse diffusion conditions (Liu et al., 2017c; Tutsak and Koçak, 2019; Li et al., 2020c). The relevant meteorological factors were not considered by this comparison model, and the current results emphasized the different sulfate formation potentials through kinetic regime."*

Line 380-390: In addition to the temperature dependence, did the authors consider the change of relative humidity during the day and night can also contribute to the heterogeneous chemistry reaction with sulfate?

Answer:

We attempted to carry out the heterogeneous laboratory experiments under the higher relative humidity (RH) relative to the current setting. However, the presence of surface adsorbed water makes the Gaussian/Lorentzian deconvolution difficult to be performed, as reflected by the large uncertainties in determining the relative abundance of $SO_2 \cdot H_2O$ and $SO_3^{2-}$. Accordingly, the RH of 50% was utilized for the experiments. In the revised manuscript (line 654-656), we have emphasized the importance of various meteorological factors when discussing the atmospheric relevance of heterogeneous chemistry, as shown follows.

*"In addition, the influence of meteorological factors, like temperature, humidity and irradiance, on the atmospheric relevance of heterogeneous chemistry warrants further research. "*

Line 391-393: This sentence "Relative to … heterogeneous chemistries by …" does not make sense, please rephrase/refine the sentence.

Answer:

We have rewritten the sentences to make the relevant content more readable (line 417-420).

*"The sulfate formation rates of gas-phase, aqueous-phase and heterogeneous chemistries under solar irradiation are respectively 9.4, 6.8 and 3.0 times greater than the corresponding nocturnal results, indicating that the oxidations in gaseous and liquid media could be more kinetically susceptible to the occurrence of sunlight than those relevant to the gas-solid interface."*

Figure 5: Can author provide some discussion about the trend shown in Figure 5? Why would it show a dip in sulfate formation rate at pH 5 and more sulfate at higher pH?

Answer:

The dip in sulfate formation rate around the pH of 5.0 is attributed to the significant sulfate formations under the extremely acidic and near-alkaline conditions. We have discussed the pH-dependent kinetics of the aqueous-phase and dust-mediated heterogeneous chemistries (line 449-452), as shown follows.

*"In more detail, the significant sulfate formation in the acidic environment is primarily contributed by the TMI-$O_2$ pathway where the concentrations of TMIs and $H^+$ keep increasing as the aerosol water becomes more acidic. When pH>5.0, the formation rate of sulfate increases with the elevated pH, in step with the evolution of the concentration of dissolved sulfur species."*

Figure 10 Line 558-559: Can author explain why the $O_2$ affects $SO_2$ oxidation over pH <4 and why is it starting to decrease after pH 4?

Answer:

We have explained the pH-dependent kinetics of the interfacial $O_2$ at acidic aerosol interface (line 592-594).

*"As reported by Hung et al. (2015, 2018), acidic droplet interface favors the noncatalyzed oxidation by the presence of sufficient $SO_3^{\bullet-}$ and $SO_4^{\bullet-}$ when pH<4.0, which can be further explained by the structure differences of water at the interface versus the characteristic water structure in bulk-phase water."*

**Comments in the quick report**

Can author provide more information about the particle acidity calculation and standard calibration? Rindelaub et al developed the direct pH measurement for single-particle using the ratio of bisulfate and sulfate peak.

Answer:

The calculation of particle acidity by the relative abundance of bisulfate and sulfate may not be applied for this study because the characteristic bisulfate signals cannot be determined in some of the recorded infrared spectra. We have discussed the report from Rindelaub et al. (2016) and provided more information on the calculation of particle acidity (line 195-197).

*"The relative abundance of $SO_2 \cdot H_2O$ and $SO_3^{2-}$, as assumed to be equivalent to the relative integral area of their*

*characteristic peaks, is utilized to calculate the particle acidity (more details in Text S3-2 of Supporting Information). The relative abundance of HSO$_4^-$ and SO$_4^{2-}$, which can be used to calculate the aerosol acidity (Rindelaub et al., 2016; Ault, 2020), was not considered by the present study because the characteristic signals of bisulfate cannot be observed in some of the recorded infrared spectra."*

Line 336: Author mentioned that the more sulfate formation during the daytime than the nighttime, but later author suggests more secondary sulfate aerosol formation during nighttime than daytime, can author clarify this?

Answer:

As discussed in Section 3.2, the dust surface drivers are responsible for the atmospheric sulfate formation rate of 0.195 µg m$^{-3}$ h$^{-1}$ during nighttime and 0.365 µg m$^{-3}$ h$^{-1}$ during daytime. As discussed in Section 3.3, the dust heterogeneous chemistry is responsible for 28.6% and 13.1% of the secondary sulfate during nighttime and daytime, respectively. We have clarified the different dependences of absolute sulfate yield and relative sulfate contribution in the revised manuscript (line 411-413).

*"In other words, although the heterogeneous sulfate production during nighttime is quantitatively lower than that during daytime (see Section 3.2), the nocturnal heterogeneous chemistry accounts for the higher proportion of secondary sulfate."*

In figure 5, it is interesting to see the formation rate of sulfate increase after pH5, can author provide some discussion about this trend and the increasing sulfate formation with increasing pH?

Answer:

In the revised manuscript (line 450-452), we have explained the positive pH dependence when pH>5.0.

*"When pH>5.0, the formation rate of sulfate increases with the elevated pH, in step with the evolution of the concentration of dissolved sulfur species."*

---

## Editor Decision (ED1)

Atmospheric
Chemistry
and Physics

**Heterogeneity and chemical reactivity of the remote troposphere defined by aircraft measurements**

**Hao Guo[1], Clare M. Flynn[2], Michael J. Prather[1], Sarah A. Strode[3], Stephen D. Steenrod[3], Louisa Emmons[4], Forrest Lacey[4,5], Jean-Francois Lamarque[4], Arlene M. Fiore[6], Gus Correa[6], Lee T. Murray[7], Glenn M. Wolfe[3,8], Jason M. St. Clair[3,8], Michelle Kim[9], John Crounse[10], Glenn Diskin[10], Joshua DiGangi[10], Bruce C. Daube[11,12], Roisin Commane[11,12], Kathryn McKain[13,14], Jeff Peischl[14,15], Thomas B. Ryerson[13,15], Chelsea Thompson[13], Thomas F. Hanisco[3], Donald Blake[16], Nicola J. Blake[16], Eric C. Apel[4], Rebecca S. Hornbrook[4], James W. Elkins[14], Eric J. Hintsa[13,14], Fred L. Moore[13,14], and Steven Wofsy[11]**

[1]Department of Earth System Science, University of California, Irvine, CA 92697, USA
[2]Department of Meteorology, Stockholm University, Stockholm 106 91, Sweden
[3]Atmospheric Chemistry and Dynamics Laboratory, NASA Goddard Space Flight Center, Greenbelt, MD 20771, USA
[4]Atmospheric Chemistry Observations and Modeling Laboratory, National Center for Atmospheric Research, Boulder, CO 80301, USA
[5]Department of Mechanical Engineering, University of Colorado, Boulder, CO 80309, USA
[6]Department of Earth and Environmental Sciences and Lamont-Doherty Earth Observatory, Columbia University, Palisades, NY 10964, USA
[7]Department of Earth and Environmental Sciences, University of Rochester, Rochester, NY 14611, USA
[8]Joint Center for Earth Systems Technology, University of Maryland, Baltimore County, Baltimore, MD 21228, USA
[9]Department of Geological and Planetary Sciences, California Institute of Technology, Pasadena, CA 91125, USA
[10]Atmospheric Composition, NASA Langley Research Center, Hampton, VA 23666, USA
[11]John A. Paulson School of Engineering and Applied Sciences, Harvard University, Cambridge, MA 02138, USA
[12]Department of Earth and Planetary Sciences, Harvard University, Cambridge, MA 02138, USA
[13]Cooperative Institute for Research in Environmental Sciences, University of Colorado, Boulder, CO 80309, USA
[14]Global Monitoring Division, Earth System Research Laboratory, NOAA, Boulder, CO 80305, USA
[15]Chemical Sciences Division, National Oceanic and Atmospheric Administration Earth System Research Laboratory, Boulder, CO 80305, USA
[16]Department of Chemistry, University of California, Irvine, CA 92697, USA

**Correspondence:** Hao Guo (haog2@uci.edu) and Michael J. Prather (mprather@uci.edu)

Received: 13 May 2021 – Discussion started: 19 May 2021
Revised: 20 August 2021 – Accepted: 24 August 2021 – Published: 16 September 2021

**Abstract.** The NASA Atmospheric Tomography (ATom) mission built a photochemical climatology of air parcels based on in situ measurements with the NASA DC-8 aircraft along objectively planned profiling transects through the middle of the Pacific and Atlantic oceans. In this paper we present and analyze a data set of 10 s (2 km) merged and gap-filled observations of the key reactive species driving the chemical budgets of $O_3$ and $CH_4$ ($O_3$, $CH_4$, CO, $H_2O$, HCHO, $H_2O_2$, $CH_3OOH$, $C_2H_6$, higher alkanes, alkenes, aromatics, $NO_x$, $HNO_3$, $HNO_4$, peroxyacetyl nitrate, other organic nitrates), consisting of 146 494 distinct air parcels from ATom deployments 1 through 4. Six models calculated the $O_3$ and $CH_4$ photochemical tendencies from this modeling data stream for ATom 1. We find that 80 %–90 % of the total reactivity lies in the top 50 % of the parcels and 25 %–35 % in the top 10 %, supporting previous model-only studies that tropospheric chemistry is driven by a fraction of all the air. In other words, accurate simulation of the least reactive 50 % of the troposphere is unimportant for global budgets. Surprisingly, the probability densities of species and reactivities averaged on a model scale (100 km) differ only slightly from the 2 km ATom data, indicating that much of the heterogeneity in tropospheric chemistry can be captured with current global chemistry models. Comparing the ATom reactivities over the tropical oceans with climatological statistics from six global chemistry models, we find excellent agreement with the loss of $O_3$ and $CH_4$ but sharp disagreement with production of $O_3$. The models sharply underestimate $O_3$ production below 4 km in both Pacific and Atlantic basins, and this can be traced to lower $NO_x$ levels than observed. Attaching photochemical reactivities to measurements of chemical species allows for a richer, yet more constrained-to-what-matters, set of metrics for model evaluation.

**1 Prologue**

This paper is based on the methods and results of papers that established an approach for analyzing aircraft measurements, specifically the NASA Atmospheric Tomography Mission (ATom), with global chemistry models. Here we present a brief overview of those papers to help the reader understand the basis for this paper. The first ATom modeling paper ("Global atmospheric chemistry – which air matters", Prather et al., 2017, hence P2017) gathered six global models, both chemistry–transport models (CTMs) and chemistry–climate models (CCMs). The models reported a single-day snapshot for mid-August (the time of the first ATom deployment, ATom-1), and these included all species relevant for tropospheric chemistry and the 24 h reactivities. We limited our study to three reactivities (Rs) controlling methane ($CH_4$) and tropospheric ozone ($O_3$) using specific reaction rates to define the loss of $CH_4$ and the production and loss of $O_3$ in parts per billion (ppb) per day. The critical photolysis rates ($J$ values) are also reported as 24 h averages.

$$\text{L-CH4}: CH_4 + OH \rightarrow CH_3 + H_2O \tag{1}$$

$$\text{P-O3}: HO_2 + NO \rightarrow NO_2 + RO \tag{2a}$$
$$RO_2 + NO \rightarrow NO_2 + RO, \tag{2b}$$
$$\text{where } NO_2 + h\nu \rightarrow NO + O \text{ and } O + O_2 \rightarrow O_3 \tag{2c}$$
$$O_2 + h\nu \rightarrow O + O (x2) \tag{2d}$$

$$\text{L-O3}: O_3 + OH \rightarrow O_2 + HO_2 \tag{3a}$$
$$O_3 + HO_2 \rightarrow HO + O_2 + O_2 \tag{3b}$$
$$O(^1D) + H_2O \rightarrow OH + OH \tag{3c}$$

$$\text{J-O1D}: O_3 + h\nu \rightarrow O(^1D) + O_2 \tag{4}$$
$$\text{J-NO2}: NO_2 + h\nu \rightarrow NO + O \tag{5}$$

Models also reported the change in $O_3$ over 24 h, and these match the P-O3 minus L-O3 values over the Pacific basin (a focus of this study). The models showed a wide range in the three Rs average profiles across latitudes over the Pacific basin, as well as 2D probability densities (PDs) for key species such as $NO_x$ ($NO + NO_2$) versus HOOH. A large part of the model differences was attributed to the large differences found in chemical composition. We found that single transects from a model through the tropical Pacific at different longitudes produced nearly identical 2D PDs, but these PDs were distinctly different across models. This result supported the premise that the ATom PDs would provide a useful metric for global chemistry models.

In P2017, we established a method for running the chemistry modules in the CTMs and CCMs with an imposed chemical composition from aircraft data: the ATom run, or "A run". In the A run, the chemistry of each grid cell does not interact with its neighbors or with externally imposed emission sources. Effectively the CTM/CCM is initialized and run for 24 h without transport, scavenging or emissions. Aerosol chemistry is also turned off in the A runs. This method allows each parcel to evolve in response to the daily cycle of photolysis in each model and be assigned a 24 h integrated reactivity. The instantaneous reaction rates at the time an air parcel is measured (e.g., near sunset at the end of a flight) do not reflect that parcel's overall contribution to the $CH_4$ or $O_3$ budget; a full diel cycle is needed. The A run assumption that parcels do not mix with neighboring air masses is an approximation, and thus for each model we compared the A runs using the model's restart data with a parallel standard 24 h simulation (including transport, scavenging, and emissions). Because the standard grid-cell air moves and mixes, we compared averages over a large region (e.g., tropical Pacific). We find some average biases of order $\pm 10\%$ but general agreement. The largest systematic biases in the A runs are caused by buildup of HOOH (no scavenging) and decay of $NO_x$ (no sources). The A runs are relatively easy to code for most CTM/CCMs and allow each model's chemistry module, including photolysis package, to run normally. The A runs do not distinguish between CTMs and CCMs, except that each model will generate/prescribe its own cloud fields and photolysis rates. Our goal is to create a robust understanding of the chemical statistics including the reactivities with which to test and evaluate the free-running CCMs, and thus we do not try to model the specific period of the ATom deployments. Others may use the ATom data with hindcast CTMs to test forecast models, but here we want to build a chemical climatology.

The first hard test of the A runs came with the second ATom modeling paper ("How well can global chemistry models calculate the reactivity of short-lived greenhouse gases in the remote troposphere, knowing the chemical composition", Prather et al., 2018, hence P2018). The UCI CTM simulated an aircraft-like data set of 14 880 air parcels along the International Date Line from a separate high-resolution

(0.5°) model. Each parcel is defined by the following core species: $H_2O$, $O_3$, $NO_x$, $HNO_3$, $HNO_4$, PAN (peroxyacetyl nitrate), $CH_3NO_3$, HOOH, $CH_3OOH$, HCHO, $CH_3CHO$ (acetaldehyde), $C_3H_6O$ (acetone), CO, $CH_4$, $C_2H_6$, alkanes ($C_3H_8$ and higher), $C_2H_4$, aromatics (benzene, toluene, xylene) and $C_5H_8$ (isoprene), plus temperature. Short-lived radicals (e.g., OH, $HO_2$, $CH_3OO$) were initialized at small concentrations and quickly reached daytime values determined by the core species. The six CTM/CCMs overwrote the chemical composition of a restart file, placing each pseudo-observation in a unique grid cell according to its latitude, longitude, and pressure. If another parcel is already in that cell, then it is shifted east–west or north–south to a neighboring model cell. For coarse-resolution models, multiple restart files and A runs were used to avoid large location shifts. CTM/CCMs usually have a locked in 24 h integration step starting at 00:00 UTC that is extremely difficult to modify in order to try to match the local solar time of observation, especially as it changes along aircraft flights. We tested the results with a recoded UCI CTM to start at 12:00 UTC but retain the same clouds fields over the day and found only percentage-level differences between a midnight or noon start.

These A runs averaged over cloud conditions by simulating 5 d in August at least 5 d apart. Assessment of the modeled photolysis rates and comparison with the ATom-measured $J$ values is presented in Hall et al. (2018, hence H2018). All models agreed that a small fraction of chemically hot air parcels in the synthetic data set controlled most of the total reactivity. Some models had difficulty in implementing the A runs because they overwrote the specified water vapor with the modeled value, but this problem is fixed here. In both P2017 and P2018, the GISS-E2 model stood out with the most unusual chemistry patterns and sometimes illogical correlations. Efforts by a co-author to clarify the GISS results or identify errors in the implementation have not been successful. GISS results are included here for completeness in the set of three papers but are not reconciled. Overall, three models showed remarkable inter-model agreement in the three Rs with less than half of the RMSD (root-mean-square difference) as compared with the other models. UCI also tested the effect of different model years (1997 and 2015 versus reference year 2016), which varies the cloud cover and photolysis rates, and found an inter-year RMSD about half of that of the core model's RMSD. Thus, there is a fundamental uncertainty in this approach due to the inability to specify the cloud/photolysis history seen by a parcel over 24 h, but it is less than the inter-model differences among the most similar models.

**2 Introduction**

The NASA Atmospheric Tomography (ATom) mission completed a four-season deployment, each deployment flying from the Arctic to Antarctic and back, traveling south through the middle of the Pacific Ocean, across the Southern Ocean and then north through the Atlantic Ocean, with near-constant profiling of the marine troposphere from 0.2 to 12 km altitude (see Fig. S1 in the Supplement). The DC8 was equipped with in situ instruments that documented the chemical composition and conditions at time intervals ranging from < 1 to about 100 s (Wofsy et al., 2018). ATom measured hundreds of gases and aerosols, providing information on the chemical patterns and reactivity in the vast remote ocean basins, where most of the destruction of tropospheric ozone ($O_3$) and methane ($CH_4$) occurs. Reactivity is defined here as in P2017 to include the production and loss of $O_3$ (P-O3 and L-O3, ppb/d) and loss of $CH_4$ (L-CH4, ppb/d). Here we report on this model-derived product that was proposed for ATom, the daily averaged reaction rates determining the production and loss of $O_3$ and the loss of $CH_4$ for 10 s averaged air parcels. We calculate these rates with 3D chemical models that include variations in clouds and photolysis and then assemble the statistical patterns describing the heterogeneity (i.e., high spatial variability) of these rates and the underlying patterns of reactive gases.

Tropospheric $O_3$ and $CH_4$ contribute to climate warming and global air pollution (Stocker et al., 2013). Their abundances in the troposphere are controlled largely by tropospheric chemical reactions. Thus, chemistry–climate assessments seeking to understand past global change and make future projections for these greenhouse gases have focused on the average tropospheric rates of production and loss and how these reactivities are distributed in large semi-hemispheric zones throughout the troposphere (Griffiths et al., 2021; Myhre et al., 2014; Naik et al., 2013; Prather et al., 2001; Stevenson et al., 2006, 2013, 2020; Voulgarakis et al., 2013; Young et al., 2013). The models used in these assessments disagree on these overall $CH_4$ and $O_3$ reactivities (a.k.a. the budgets), and resolving the cause of such differences is stymied because of the large number of processes involved and the resulting highly heterogeneous distribution of chemical species that drive the reactions. Simply put, the models use emissions, photochemistry and meteorological data to generate the distribution of key species such as nitrogen oxides ($NO_x = NO + NO_2$) and hydrogen peroxide (HOOH) (step 1) and then calculate the $CH_4$ and $O_3$ reactivities from these species (step 2). There is no single average measurement that can test the verisimilitude of the models. Stratospheric studies such as Douglass et al. (1999) have provided a quantitative basis for testing chemistry and transport and defining model errors, but few of these studies have tackled the problem of modeling the heterogeneity of tropospheric chemistry. The major model differences lie in the first step because when we specify the mix of key chemical species, most models agree on the $CH_4$ and $O_3$ chemical budgets (P2018). The intent of ATom was to collect an atmospheric sampling of all the key species and the statistics defining their spatial variability and thus that of the reactivities of $CH_4$ and $O_3$.

Many studies have explored the ability of chemistry–transport models (CTMs) to resolve finer scales such as pollution layers (Eastham and Jacob, 2017; Rastigejev et al., 2010; Tie et al., 2010; Young et al., 2018; Zhuang et al., 2018), but these have not had the chemical observations (statistics) to evaluate model performance. In a great use of chemical statistics, Yu et al. (2016) used 60 s data ($\sim 12$ km) from the SEAC$^4$RS aircraft mission to compare cumulative probability densities (PDs) of $NO_x$, $O_3$, HCHO and isoprene over the Southeast US with the GEOS-Chem CTM run at different resolutions. They identified clear biases at the high and low ends of the distribution, providing a new test of models based on the statistics rather than mean values. Heald et al. (2011) gathered high-resolution profiling of organic and sulfate aerosols from 17 aircraft missions and calculated statistics (mean, median, quartiles) but only compared with the modeled means. The HIAPER Pole-to-Pole Observations (HIPPO) aircraft mission (Wofsy, 2011) was a precursor to ATom with regular profiling of the mid-Pacific including high-frequency 10 s sampling that identified the small scales of variability throughout the troposphere. HIPPO measurements were limited in species, lacking $O_3$, $NO_x$ and many of the core species needed for reactivity calculations. ATom, with a full suite of reactive species and profiling through the Atlantic basin, provides a wealth of chemical statistics that challenge the global chemistry models.

Our task here is the assembly of the modeling data stream (MDS), which provides flight-wise continuous 10 s data (air parcels) for the key reactive species. The MDS is based on direct observations and interpolation methods to fill gaps as documented the Supplement. Using the MDS, we have six chemical models calculating the 24 h reactivities, producing a reactivity data stream (RDS) using protocols noted in the Prologue (P2017) and described further in Sect. 2. There, we describe the updated modeling protocol RDS* necessary to address measurement noise in key species that can be very short-lived. In Sect. 4, we examine the statistics of reactivity over the Atlantic and Pacific oceans, focusing on air parcels with high reactivity; for example, 10 % of the parcels produce 25 %–35 % of total reactivity over the oceans. We compare these ATom-1 statistics, species and reactivities with August climatologies from six global chemistry models. In one surprising result, ATom-1 shows a more reactive tropical troposphere than found in most models' climatologies associated with higher $NO_x$ levels than in the models. Section 5 concludes that the ATom PDs based on 10 s air parcels do provide a valid chemistry metric for global models with $1°$ resolution. It also presents some examples where ATom measurements and modeling can test the chemical relationships and may address the cause of differences in the $O_3$ and $CH_4$ budgets currently seen across the models. With this paper we release the full ATom MDS-2 from all four deployments, along with the updated RDS* reactivities from the UCI model.

**3 Models and data**

**3.1 The modeling data stream (MDS)**

The ATom mission was designed to collect a multi-species, detailed chemical climatology that documents the spatial patterns of chemical heterogeneity throughout the remote troposphere. Figure S1 in the Supplement maps the 48 research flights, and the Supplement has tables summarizing each flight. We required a complete set of key species in each air parcel to initialize the models that calculate the $CH_4$ and $O_3$ reactivities. We choose the key reactive species ($H_2O$, $O_3$, CO, $CH_4$, $NO_x$, $PSSNO_x$ (photostationary state $NO_x$), $HNO_3$, $HNO_4$, PAN, $CH_2O$, $H_2O_2$, $CH_3OOH$, acetone, acetaldehyde, $C_2H_6$, $C_3H_8$, $i$-$C_4H_{10}$, $n$-$C_4H_{10}$, alkanes, $C_2H_4$, alkenes, $C_2H_2$, $C_5H_8$, benzene, toluene, xylene, $CH_3ONO_2$, $C_2H_5ONO_2$, $RONO_2$, $CH_3OH$) directly from the ATom measurements and then add corollary species or other observational data indicative of industrial or biomass burning pollution or atmospheric processing (HCN, $CH_3CN$, $SF_6$, relative humidity, aerosol surface area (four modes) and cloud indicator). We choose 10 s averages for our air parcels as a compromise and because the 10 s merged data are a standard product (Wofsy et al., 2018). A few instruments measure at 1 s intervals, but the variability at this scale is not that different from 10 s averages (Fig. S2). Most of the key species are reported as 10 s values, with some being averaged or sampled at 30 s or longer such as $\sim 90$ s for some flask measurements.

Throughout ATom, gaps occur in individual species on a range of timescales due to calibration cycles, sampling rates or instrument malfunction. The generation of the MDS uses a range of methods to fill these gaps and assigns a flag index to each species and data point to allow users to identify primary measurements and methods used for gap-filling. Where two instruments measure the same species, the MDS selects a primary measurement and identifies which instrument was used with a flag. The methodology and species-specific information on how the current MDS version 2 (MDS-2) is constructed, plus statistics on the 48 research flights and the 146 494 10 s air parcels in MDS-2, are given in the Supplement.

Over the course of this study, several MDS versions were developed and tested, including model-derived RDSs from these versions, some of which are used in this paper. In early ATom science team meetings, there was concern about the accuracy of $NO_2$ direct measurements when at very low concentrations. A group prepared an estimate for $NO_x$ using the NO and $O_3$ measurements to calculate a photostationary value for $NO_2$ and thus $NO_x$. This PSS-$NO_x$ became the primary $NO_x$ source in version 0 (i.e., MDS-0). The numbering of versions initially followed the notation of revisions in the mission data archive (MDS_R0, MDS_R1, ...), but this was restrictive, and we adopted the simpler notation here but still beginning with version 0. With MDS-0, we chose to gap-fill using correlations with CO to estimate the variability

of the missing measurement over the gap. The science team then rejected PSS-NO$_x$ as a proxy, and we reverted to the observed NO + NO$_2$ for MDS-1, resulting in increased NO$_x$ and reactivities (RDS-1). MDS-1 NO$_x$ values are 25 % larger on average than MDS-0 values (unweighted mean of 66 vs. 52 ppt), and this affects P-O3 most and L-CH4 least. We then estimated errors in the gap-filling and found that CO had little skill as a proxy for most other species. With MDS-2, we optimized and tested the treatments of gap-filling and lower limit of detection, along with other quality controls. MDS-2 is fully documented in the Supplement.

**3.2 The reactivity data stream (RDS)**

The concept of using an MDS to initialize 3D global chemistry models and calculate an RDS was developed in the pre-ATom methodology papers (P2017; P2018). In this paper, we use the original six models for their August chemical statistics, and we use five of them plus a box model to calculate the reactivities; see Table 1. The RDS is really a protocol applied to the MDS. It is introduced in the Prologue, and the details can be found in P2018. A model grid cell is initialized with all the core reactive species needed for a regular chemistry simulation. The model is then integrated over 24 h without transport or mixing, without scavenging and without emissions. Each global model uses its own varying cloud fields for the period to calculate photolysis rates, but the F0AM box model simply takes the instant $J$ values as measured on the flight and applies a diurnal scaling. We can initialize with the core species and let the radicals (OH, HO$_2$, RO$_2$) come into photochemical balance. The 24 h integration is not overly sensitive to the start time of the integration, and thus models do not have to synchronize with the local time of observation (see P2018's Fig. S8 and Table S8).

The initial RDS came from MDS-0 and six of the models in Table 1. This paper was nearly complete when we identified the problem with PSS-NO$_x$. We had gathered enough information on how models agree, or disagree, with RDS-0 and thus chose to assess MDS-1 with two of the models that closely agreed (GMI and UCI). The two models were very close in RDS-0 and also in RDS-1. We then found the problems with the CO proxy and chose to use only the UCI model as a transfer standard for the change from MDS-1 to MDS-2 (i.e., RDS-1 to RDS-2). This path avoided much extra work by the modeling groups and generated the same information on cross-model differences and a robust estimate of changes from RDS-0 to RDS-2.

Statistics for the three reactivities for six models using MDS-0, 2 alternative UCI model years using MDS-0, the GMI model using MDS-1 and the UCI model using MDS-2 are given in Tables 2 and S8 for three domains: global (all points), Pacific (oceanic data from 54° S to 60° N) and Atlantic (same constraints as Pacific). UCI MDS-1 is similar to UCI MDS-2 and is not shown. The statistics try to achieve equal latitude-by-pressure sampling by weighting each ATom parcel inversely according to the number of parcels in each 10° latitude by 100 hPa bin. We calculate the means and medians plus the percent of total reactivity in the top 10 % of the weighted parcels (Table 2) and also the mean reactivity of the top 10 %, percent of total reactivity in the top 50 %, 10 % and 3 % plus the mean $J$ values (Table S8).

Unfortunately, while investigating sensitivities and uncertainties in the RDS for a future study, we found an inconsistency between the reported concentrations of both pernitric acid (HNO$_4$) and peroxyacetyl nitrate (PAN) with respect to the chemical kinetics used in the models. High concentrations (attributed to instrument noise) were reported under conditions where the thermal decomposition frequency was > 0.4 per hour in the lower troposphere (> 253 K for HNO$_4$ and > 291 K for PAN). Thus, these species instantly become NO$_x$. There is no easy fix for this, and we left the species data in the MDS as they were reported but developed a new protocol RDS* to deal with them. Both species are allowed to decay for 24 h using their thermal decomposition rate before being put into the model. This avoids most of the fast thermal release of NO$_x$ in the 24 h of the RDS calculation but does not affect the release of NO$_x$ from photolysis or OH reactions in the upper troposphere where thermal decomposition in inconsequential. It is possible that some of the high concentrations of HNO$_4$ and PAN in the lower troposphere are real and that we are missing this large source of NO$_x$ with the RDS* protocol, but there are no obvious sources of these species in the remote oceanic regions that would produce enough to match the thermal loss. Both this problem and its solution do not affect the initial NO$_x$. This revised protocol (UCI2*) is shown in Tables 2 and S8 next to the standard protocol (UCI2). The reactivities drop slightly (3 % for P-O3, 2 % for L-O3 and 0 % for L-CH4) as expected with less NO$_x$, but the sensitivity of the reactivities to these compounds ($\partial \ln R / \partial \ln X$) drops by a factor of 2 or more. We use the UCI2* results as our best estimate of the ATom reactivities for the figures in this paper.

**3.3 Inter-model differences**

Variations in reactivities due to clouds are an irreducible source of uncertainty: predicting the cloud-driven photolysis rates that a shearing air parcel will experience over 24 h is not possible here. The protocol uses 5 separated 24 h days to average over synoptically varying cloud conditions. The standard deviation ($\sigma$) of the 5 d, as a percentage of the 5 d mean, is averaged over all parcels and shown in Table S9 for the five global models. Three central models (GC, GMI, UCI) show 9 %–10 % $\sigma$(Js) values and similar $\sigma$(Rs) values as expected if the variation in $J$ values is driving the reactivities. Two models (GISS, NCAR) have 12 %–17 % $\sigma$(Js), which might be explained by more opaque clouds, but the amplified $\sigma(R)$ values (14 %–32 %) are inexplicable. This discrepancy needs to be resolved before using these two models for ATom RDS analysis.

**Table 1.** Chemistry models.

| Used for | ID | Model name | Model type | Meteorology | Model grid |
|---|---|---|---|---|---|
| clim | GFDL | GFDL-AM3 | CCM | NCEP (nudged) | $C180 \times L48$ |
| clim, MDS-0 | GISS | GISS-E2.1 | CCM | Daily SSTs, nudged to MERRA | $2° \times 2.5° \times 40L$ |
| clim, MDS-0/1 | GMI | GMI-CTM | CTM | MERRA | $1° \times 1.25° \times 72L$ |
| clim, MDS-0 | GC | GEOS-Chem | CTM | MERRA-2 | $2° \times 2.5° \times 72L$ |
| clim, MDS-0 | NCAR | CAM4-Chem | CCM | Nudged to MERRA | $0.47° \times 0.625° \times 52L$ |
| clim, MDS-0/1/2 | UCI | UCI-CTM | CTM | ECMWF IFS Cy38r1 | $T159N80 \times L60$ |
| MDS-0 | F0AM | F0AM | box | MDS + scaled ATom Js | n/a |

The descriptions of models used in the paper. The first column denotes if the model's August climatology is used ("clim") and also the MDS versions used. F0AM used chemical mechanism MCMv331 plus J-HNO$_4$ plus O$^1$D)+CH$_4$. For the global models, see P2017, P2017 and H2018. n/a – not applicable

**Table 2.** Reactivity statistics for the three large domains (global, Pacific, Atlantic).

| Value | Region | MDS-0 | | | | | | | | MDS-1 | MDS-2 | |
|---|---|---|---|---|---|---|---|---|---|---|---|---|
| | | F0AM | GC | GISS | GMI | NCAR | UCI | U15 | U97 | GMI1 | UCI2 | UCI2* |
| P-O3, mean, ppb/d | | | | | | | | | | | | |
| | Global | 1.94 | 1.91 | 2.31 | 1.86 | 1.97 | 2.15 | 2.13 | 2.13 | 2.07 | 2.18 | 2.11 |
| | Pacific | 1.91 | 1.95 | 1.94 | 1.92 | 1.92 | 2.13 | 2.08 | 2.10 | 2.06 | 2.33 | 2.26 |
| | Atlantic | 1.88 | 1.99 | 3.29 | 2.07 | 2.28 | 2.32 | 2.32 | 2.34 | 2.22 | 2.08 | 2.02 |
| L-O3, mean, ppb/d | | | | | | | | | | | | |
| | Global | 1.63 | 1.45 | 1.75 | 1.50 | 1.51 | 1.56 | 1.55 | 1.55 | 1.50 | 1.57 | 1.54 |
| | Pacific | 1.60 | 1.48 | 1.74 | 1.51 | 1.44 | 1.54 | 1.50 | 1.52 | 1.48 | 1.53 | 1.50 |
| | Atlantic | 2.06 | 1.90 | 2.23 | 2.04 | 2.28 | 2.14 | 2.14 | 2.16 | 2.04 | 2.15 | 2.11 |
| L-CH4, mean, ppb/d | | | | | | | | | | | | |
| | Global | 0.72 | 0.66 | 0.38 | 0.65 | 0.62 | 0.68 | 0.68 | 0.68 | 0.67 | 0.68 | 0.68 |
| | Pacific | 0.81 | 0.78 | 0.38 | 0.76 | 0.73 | 0.79 | 0.77 | 0.78 | 0.77 | 0.79 | 0.79 |
| | Atlantic | 0.77 | 0.74 | 0.49 | 0.77 | 0.80 | 0.80 | 0.80 | 0.81 | 0.79 | 0.79 | 0.79 |
| P-O3, % of total $R$ in top 10 % | | | | | | | | | | | | |
| | Global | 37 % | 34 % | 32 % | 34 % | 32 % | 36 % | 36 % | 36 % | 33 % | 34 % | 34 % |
| | Pacific | 34 % | 28 % | 28 % | 29 % | 29 % | 31 % | 30 % | 30 % | 28 % | 27 % | 27 % |
| | Atlantic | 25 % | 26 % | 24 % | 27 % | 25 % | 28 % | 28 % | 28 % | 25 % | 25 % | 25 % |
| L-O3, % of total $R$ in top 10 % | | | | | | | | | | | | |
| | Global | 37 % | 37 % | 35 % | 38 % | 38 % | 38 % | 39 % | 39 % | 39 % | 38 % | 38 % |
| | Pacific | 34 % | 32 % | 30 % | 32 % | 32 % | 32 % | 32 % | 32 % | 34 % | 31 % | 31 % |
| | Atlantic | 29 % | 31 % | 30 % | 31 % | 35 % | 31 % | 31 % | 31 % | 31 % | 31 % | 31 % |
| L-CH4, % of total $R$ in top 10 % | | | | | | | | | | | | |
| | Global | 36 % | 33 % | 29 % | 34 % | 34 % | 35 % | 35 % | 35 % | 35 % | 33 % | 33 % |
| | Pacific | 33 % | 29 % | 26 % | 30 % | 30 % | 30 % | 30 % | 30 % | 31 % | 27 % | 27 % |
| | Atlantic | 28 % | 26 % | 22 % | 27 % | 28 % | 28 % | 28 % | 28 % | 27 % | 28 % | 28 % |

Global includes all ATom-1 parcels, Pacific considers all measurements over the Pacific Ocean from 54° S to 60° N and Atlantic uses parcels from 54° S to 60° N over the Atlantic basin. All parcels are weighted inversely by the number of parcels in each 10° latitude by 100 hPa bin. Results from the different MDS versions (0, 1, 2) are shown. UCI2* uses the revised RDS* protocol that preprocesses the MDS-2 initializations with a 24 h decay of HNO$_4$ and PAN according to their local thermal decomposition frequencies; see text. See additional statistics in Table S8.

**Table 3.** Cross-model rms differences (RMSDs as a percentage of mean) for the three reactivities.

| P-O3 | F0AM | GC | GISS | GMI | NCAR | UCI |
|------|------|------|------|------|------|------|
| F0AM |      | 48 % | 95 % | 45 % | 55 % | 42 % |
| GC | 48 % |      | 78 % | **26 %** | 42 % | **32 %** |
| GISS | 95 % | 78 % |      | 81 % | 72 % | 75 % |
| GMI | 45 % | **26 %** | 81 % |      | 40 % | **35 %** |
| NCAR | 55 % | 42 % | 72 % | 40 % |      | 42 % |
| UCI | 42 % | **32 %** | 75 % | **35 %** | 42 % | *(10 %)* |
| **L-O3** |  |  |  |  |  |  |
| F0AM |      | 40 % | 44 % | 43 % | 76 % | 38 % |
| GC | 40 % |      | 33 % | **25 %** | 60 % | **24 %** |
| GISS | 44 % | 33 % |      | 36 % | 66 % | 30 % |
| GMI | 43 % | **25 %** | 36 % |      | 62 % | **28 %** |
| NCAR | 76 % | 60 % | 66 % | 62 % |      | 60 % |
| UCI | 38 % | **24 %** | 30 % | **28 %** | 60 % | *(11 %)* |
| **L-CH4** |  |  |  |  |  |  |
| F0AM |      | 47 % | 136 % | 48 % | 82 % | 45 % |
| GC | 47 % |      | 111 % | **20 %** | 60 % | 27 % |
| GISS | 136 % | 111 % |      | 114 % | 110 % | 121 % |
| GMI | 48 % | **20 %** | 114 % |      | 57 % | **30 %** |
| NCAR | 82 % | 60 % | 110 % | 57 % |      | 68 % |
| UCI | 45 % | **27 %** | 121 % | **30 %** | 68 % | *(14 %)* |

Matrices are symmetric. Calculated with the 31 376 MDS-0 unweighted parcels using the standard RDS protocol. F0AM lacks 5510 of these parcels because there are no reported $J$ values. UCI shows RMSD between years 2016 (default) and 1997 as the value in parentheses on diagonal. The unweighted mean $R$ values from three core models (GC, GMI, UCI) are P-O3 = 1.97, L-O3 = 1.50 and L-CH4 = 0.66; all are in units of ppb/d. The three core-model RMSDs are shown in bold.

Inter-model differences are shown in the parcel-by-parcel root-mean-square (rms) differences for RDS-0 in Table 3. Even when models adopt standard kinetic rates and cross sections (i.e., Burkholder et al., 2015), the number of species and chemical mechanisms included, as well as the treatment of families of similar species or intermediate short-lived reaction products, varies across models. For example, UCI considers about 32 reactive gases, whereas GC and GMI have over 100, and F0AM has more than 600. The other major difference across models is photolysis, with models having different cloud data and different methods for calculating photolysis rates in cloudy atmospheres (H2018). The three central models (GC, GMI, UCI) in terms of their 5 d variability (Table S9) are also most closely alike in these statistics, with rms = 20 %–30 % for L-CH4 up to 26 %–35 % for P-O3. These rms values appear to be about as close as any two models can get. The intra-model rms for different years (UCI 2016 versus 1997) is 10 %–13 % and shows that we are seeing basic differences in the chemical models across GC, GMI and UCI. F0AM is the closest to the central models, but it will inherently have a larger rms because it is a 1 d calculation and not a 5 d average. NCAR's rms is consistently higher and likely related to what is seen in the 5 d $\sigma$ values in Table S9. GISS is clearly different from all the others (L-CH4 MS > 100 %, while L-O3 rms < 66 %).

**4 Results**

Our analysis of the reactivities uses the six-model RDS-0 results to examine the consistency in calculating the Rs across models. Thereafter, we rely on the similar results from the three central models (GC, GMI, UCI) to justify use of UCI RDS*-2 as our best estimate for ATom reactivities. The uncertainty in this estimate can be approximated by the inter-model spread of the central models as discussed above. When evaluating the model's climatology for chemical species, we use MDS-2. A summary of the key data files used here, as well as their sources and contents, is given in Table 4.

**4.1 Probability densities of the reactivities**

The reactivities for three large domains (global, Pacific, Atlantic) from the six-model RDS-0 are summarized in Tables 2 and S8. Sorted PDs for the three Rs and Pacific and Atlantic Ocean basins are plotted in Fig. 1 and show the importance of the most reactive "hot" parcels with deeply convex curves and the sharp upturn in $R$ values above 0.9 cumulative weight (top 10 %). Both basins show a similar emphasis on the most reactive hot parcels: 80 %–90 % of total $R$ is in the top 50 % of the parcels, 25 %–35 % is in the top 10 % and about 10 %–14 % is in the top 3 %. The corollary is that the bottom 50 % parcels control only 10 %–20 % of the total reactivity, which is why the median is less than mean (except for P-O3 in the Atlantic). Each $R$ value and each ocean has a unique shape; for example L-O3 in the Atlantic is almost two straight lines breaking at the 50th percentile. In Fig. 1 the agreement across all models (except GISS) is clear, indicating that the conclusion in P2018 (i.e., that most global chemistry models agree on the $O_3$ and $CH_4$ budgets if given the chemical composition) also holds for the ATom-measured chemical composition. Comparing the dashed brown (UCI, RDS-0) and black (UCIP, RDS*-2) lines, we find that the shift to observed $NO_x$ and new $HNO_4$+PAN protocol has introduced noticeable changes only for P-O3: increasing reactivities overall in the Pacific while decreasing them slightly in the Atlantic. From Table 2, these changes primarily affected mean P-O3 and were due primarily to the shift from MDS-0 to MDS-2 and secondarily to the RDS* protocol, which reduced both P-O3 and L-O3 in both basins. We conclude that accurate modeling of chemical composition of the 80th and greater percentiles is important but that modest errors in the lowest 50th percentile are inconsequential; effectively, some parcels matter more than others (P2017).

How well does this ATom analysis work as a model intercomparison project? Overall, we find that most models give similar results when presented with the ATom-1 MDS. The broad agreement of the cumulative reactive PDs across a range of model formulations using differing levels of chemical complexity shows this approach is robust. The different protocols for calculating reactivities as well as the uncertainty in cloud fields appear to have a small impact on the

https://doi.org/10.5194/acp-21-13729-2021

**Table 4.** ATom data files used here.

| Primary aircraft data | Formatting and content | Comments |
|---|---|---|
| (a) Mor.all.at1234.2020-05-27.tbl
(b) Mor.WAS.all.at1234.2020-05-27.tbl
(c) Mor.TOGA.all.at1234.2020-05-27.tbl
All from Wofsy et al. (2018). | (a) 149 133 records × 675 csv columns, 10 s merges of flight data plus chemistry & environmental measurements
(b) 6991 records × 729 csv columns, 30–120 s intervals to fill flasks
(c) 12 168 records × 727 csv columns, 35 s intervals of instrument | Core source of ATom measurements. irregular and difficult formatting; extremely long asci records; large negative integers or "NA" for some non-data. |
| Modeling data stream (MDS-2) | | |
| (a) MDS_DC8_20160729_R3.ict
(b) MDS_DC8_ 20170126_ R4.ict
(c) MDS_DC8_20170928_R4.ict
(d) MDS_DC8_ 20180424_R4.ict
(e) ATom_MDS.nc
Derived here. | (a) ATom-1: 32 383 records × 87 csv columns, 10 s intervals of chemical & other data, plus flags to indicate gap-filling
(b) ATom-2: 33 424 records × 87 csv columns
(c) ATom-3: 40 176 records × 87 csv columns
(d) ATom-4: 40 511 records × 87 csv columns
(e) ATom MDS-2: all data in netcdf | Regular formatting; all data gap-filled; NaNs only for flight 46; for use in modeling of the chemistry and related statistics from the ATom 10 s data. |
| Reactivity data stream (RDS*-2) | | |
| (a) RDS_DC8_20160729_R1.ict
(b) RDS_DC8_ 20170126_ R1.ict
(c) RDS_DC8_20170928_R1.ict
(d) RDS_DC8_ 20180424_R1.ict
(e) ATom_RDS.nc
Derived here. | (a) ATom-1: 32 383 records × 16 csv columns, 10 s intervals of flight data, modeled reactivities & $J$ values plus 5 d SD
(b) ATom-2: 33 424 records × 16 csv columns
(c) ATom-3: 40 176 records × 16 csv columns
(d) ATom-4: 40 511 records × 16 csv columns
(e) ATom RDS*-2: all data in netcdf | Results from UCI CTM only, using RDS* protocol and MDS-2; NaNs only for flight 46; for use analyzing the reactivities from the ATom 10 s data. |

shape of the cumulative PDs but are informative regarding the minimum structural uncertainty in estimating the 24 h reactivity of a well-measured air parcel.

**4.2 Spatial heterogeneity of tropospheric chemistry**

A critical unknown for tropospheric chemistry modeling is what resolution is needed to correctly calculate the budgets of key gases. A similar question was addressed in Yu et al. (2016) for the isoprene oxidation pathways using a model with variable resolution (500, 250 and 30 km) compared to aircraft measurements; see also ship plume chemistry in Charlton-Perez et al. (2009). ATom's 10 s air parcels measure 2 km (horizontal) by 80 m (vertical) during most profiles. There are obviously some chemical structures below the 10 s air parcels we use here. Only some ATom measurements are archived at 1 Hz, and we examine a test case using 1 s data for $O_3$ and $H_2O$ for a mid-ocean descent between Anchorage and Kona in Fig. S2a in the Supplement. Some of the 1 s (200 m by 8 m) variability is clearly lost with 10 s averaging, but 10 s averaging preserves most of the variability. Lines in Fig. S2 demark 400 m in altitude, and most of the variability appears to occur on this larger, model-resolved scale. Figure S2b shows the 10 s reactivities during that descent and also indicates that much of the variability occurs at 400 m scales. A more quantitative example using all the tropical ATom reactivities is shown in comparisons with probability densities below (Fig. 5).

How important is it for the models to represent the extremes of reactivity? While the sorted reactivity curves

(Fig. 1, Tables 2 and S8) continue to steepen from the 90th to 97th percentile, the slope does not change that much. Thus we can estimate the 99th + percentile contributes $< 5\%$ of the total reactivity. Thus, if our model misses the top 1 % of reactive air parcels (e.g., due to the inability to simulate intensely reactive thin pollution layers), then we miss at most 5 % of the total reactivity. This finding is new and encouraging, and it needs to be verified with the ATom-2, 3 and 4 data.

The spatial structures and variability of reactivity as sampled by the ATom tropic transects (central Pacific, eastern Pacific and Atlantic) are presented as nine panels in Fig. 2. Here, the UCI RDS*-2 reactivities are averaged and plotted in 1° latitude by 200 m thick cells, comparable to some global models (e.g., GMI, NCAR, UCI). We separate the eastern Pacific (121° W, research flight (RF) 1) from the central Pacific (RFs 3, 4 and 5) because we are looking for contiguous latitude-by-pressure structures.

In the central Pacific (row 1), highly reactive (hot) P-O3 parcels (> 6 ppb/d) occur in larger, connected air masses at latitudes 20–22° N and pressure altitudes 2–3 km and in more scattered parcels (> 3 ppb/d) below 5 km down to 20° S. High L-O3 and L-CH4 coincide with this 20–22° N air mass and also with some high P-O3 at lower latitudes. This pattern of overlapping extremes in all three Rs is surprising because the models' mid-Pacific climatologies show a separation between regions of high L-O3 (lower-middle troposphere) and high P-O3 (upper troposphere, as seen in P2017's Fig. 3). The obvious explanation is that the models leave most of the lightning-produced $NO_x$ in the upper troposphere. The ATom

[Figure]

**Figure 1.** Sorted reactivities (P-O3, L-O3, L-CH4, ppb/d) for the Pacific and Atlantic domains of ATom-1. Each parcel is weighted; see text. The six modeled reactivities for MDS-0 using the standard RDS protocol are shown with colored lines, and the UCI calculation for MDS-2 using the new RDS* protocol (HNO4 and PAN damping, denoted UCIP) is shown as a dashed black line. The mean value for each model is shown with an open circle plotted at the 50th percentile. (Flipped about the axes, this is a cumulative probability density function.)

profiling seems to catch reactive regions in adjacent profiles separate by a few hundred kilometers, scales easily resolvable with 3D models.

In the eastern Pacific (row 2), the overlap of outbound and return profiles enhances the spatial sampling over the 10 h flight. The region of very large L-O3 (> 5 ppb/d) is extensive, beginning at 5–6 km at 10° N and broadening to 2–8 km at 28° N. The region of L-CH4 is similar, but loss at the upper altitudes of this air mass is attenuated because of the temperature dependence of L-CH4 and possibly because of differing OH : HO2 ratios with altitude. Large P-O3 (> 6 ppb/d) occurs in some but not all of these highly reactive L-O3 regions, suggesting that NOx is not as evenly distributed as HOx is. P-O3 also show regions of high reactivity above 8 km that are not in the high L-O3 and L-CH4 regions, probably evidence of convective sources of HOx and NOx but too cold and dry for the L-O3 and L-CH4 reactions. ATom-1 RF1 (29 July 2016) occurred during the North American Monsoon when there was easterly flow off Mexico; thus the high reactivity of this large air mass indicates that continental deep convection is a source of high reactivity for both O3 and CH4.

In the Atlantic (row 3), we also see similar air masses through successive profiles, particularly in the northern tropics. The Atlantic P-O3 shows high-altitude reactivity similar to the eastern Pacific. Likewise, the large values of L-O3 and L-CH4 match the eastern Pacific and not central Pacific. Unlike either Pacific transect, the Atlantic L-O3 and L-CH4 show some high reactivity below 1 km altitude. Overall, the ATom-1 profiling clearly identifies extended air masses of high L-O3 and L-CH4 extending over 2–5 km in altitude and

[Figure]

**Figure 2.** Curtain plots for P-O3 (0–6 ppb/d), L-O3 (0–6 ppb/d) and L-CH4 (0–3 ppb/d) showing the profiling of ATom-1 flights in the central Pacific (RF 3, 4 and 5), eastern Pacific (RF 1) and Atlantic (RF 7, 8 and 9). Reactivities are calculated with the UCI model using MDS-2 and the new RDS* protocol (UCI RDS*-2). The 10 s air parcels are averaged into 1° latitude and 200 m altitude bins.

10° of latitude. The high P-O3 regions tend to be much more heterogeneous with greatly reduced spatial extent, likely of recent convective origin as for the eastern Pacific.

Overall, the extensive ATom profiling identifies a heterogeneous mix of chemical composition in the tropical Atlantic and Pacific, with a large range of reactivities. What is important for those trying to model tropospheric chemistry is that the spatial scales of variability seen in Fig. 2 are within the capability of modern global models.

**4.3 Testing model climatologies**

The ATom data set provides a unique opportunity to test CTMs and CCMs in a climatological sense. In this section, we compare ATom-1 data and the six models' chemical statistics for mid-August used in P2017. The ATom profiles cannot be easily compared point by point with CCMs, and we use statistical measures of the three reactivities in the three tropical basins: mean profiles in Fig. 3 and PDs in Fig. 5.

**4.3.1 Profiles**

For P-O3 profiles (top row, Fig. 3), the discrepancy between models and measurements is stark. The models (less GISS) present a consistent picture of one world, while the ATom profiles describe an entirely different world. In the central Pacific at 8–12 km, the ATom-1 results tend to agree with models, showing ozone production of about 1 ppb/d. Below 8 km, ATom's P-O3 increases to a peak of 4 ppb/d at 2 km, while the models' P-O3 stays constant down to 4 km and then decrease to about 0.5 ppb/d below 2 km. This pattern indicates that in the middle of the Pacific, the $NO_x + HO_x$ combination that produces ozone is suppressed throughout the lower troposphere in all the models. In the eastern Pacific and Atlantic, both models' and ATom reactivities indicate that P-O3 is greatly enhanced above 6 km as compared to the central Pacific, but below 6 km ATom P-O3 is much larger than that of the models', by a factor of 2. In the upper troposphere, the agreement indicates that both models and ATom find the influence of deep continental convection bringing reactive $NO_x + HO_x$ air masses to the nearby oceanic regions but not to the central Pacific. The difference below 5 km in all three regions implies a consistent bias across the models in some combination of $HO_x$ sources and/or the vertical redistribution of lightning $NO_x$. This difference is unlikely to be a sampling bias in ATom-1, given it occurs in all three regions.

For L-O3 (middle row), the agreement in the central Pacific is very good throughout the 0–12 km range; i.e., ATom looks just like one of the models (except GISS). Moving to the eastern Pacific and Atlantic, both models and ATom show increased reactivity, consistent with continental convective outflow. The large ATom reactivity in the eastern Pa-

[Figure]

**Figure 3.** Mean profiles of reactivity (rows: P-O3, L-O3, L-CH4 in ppb/d) in three domains (columns: C. Pacific, 30° S–30° N by 180–210° E; E. Pacific, 0–30° N by 230–250° E; Atlantic, 30° S–30° N by 326–343° E). Air parcels are cosine(latitude)-weighted. ATom-1 (gray) results are from Fig. 2, while model results are taken from the August climatologies in Prather et al. (2017).

cific (3–8 km) is clear in Fig. 2 and likely due to easterly mid-tropospheric flow from convection over Mexico at that specific time (29 July 2016). Similarly, the ATom reactivity at a low level (1–3 km) in the Atlantic is associated with biomass burning in Africa and was measured in other trace species. Thus, in terms of L-O3, the ATom–model differences may be due to specific meteorological conditions, and this could be tested with CTMs using 2016 meteorology and wildfires.

For L-CH4 (bottom row), the ATom–model pattern is similar to L-O3, but higher ATom reactivity occurs at lower altitudes. Overall, the ATom L-CH4 is slightly greater than the modeled L-CH4. L-O3 is dominated by O(1D) and HO2 loss,

while L-CH4 is limited to OH loss. Overall, there is clear evidence that the Atlantic and Pacific have very different chemical mixtures controlling the reactivities and that convection over land (monsoon or biomass burning) creates air masses that are still highly reactive a day or so later.

**4.3.2 Key species**

The deficit in modeled P-O3 points to a NO$_x$ deficiency in the models, and this becomes obvious in the comparison of the PD histograms for NO$_x$ shown in Fig. 4. In the central Pacific over 0–12 km (first row), ATom has a reduced frequency of parcels with 2–20 ppt and corresponding increase

in parcels with 20–80 ppt. This discrepancy is amplified in the lower troposphere, 0–4 km (second row). In the middle of the Pacific, our chemistry models are missing a large source of lower tropospheric $NO_x$. The obvious source of oceanic $NO_x$ is lightning since oceanic sources of organonitrates or other nitrate species measured on ATom could not supply this amount. The ATom statistics indicate a lightning source must be vertically mixed. In the eastern Pacific, the ATom 0–4 km troposphere appears again to have large amounts of air with 20–50 ppt, while the full troposphere more closely matches the models, except for the large occurrence of air with 100–300 ppt $NO_x$. These high-$NO_x$ upper troposphere regions are probably direct outflow from very deep convection with lightning in the monsoon regions over Mexico at this time. In the Atlantic, the models' $NO_x$ shows too frequent occurrence of low $NO_x$ ($< 10$ ppt) and thus underestimates the 10–100 ppt levels at all altitudes. ATom has a strong peak occurrence about 80–120 ppt in the upper troposphere, and, like the eastern Pacific, this is probably due to lightning $NO_x$ from deep convection over land (Africa or South America). Overall, the models appear to be missing significant $NO_x$ sources throughout the tropics, especially below 4 km.

In Fig. 4, we also look at the histograms for the key $HO_x$-related species HOOH (third row) and HCHO (fourth row). For these species, the ATom–model agreement is generally good. If anything, the models tend to have too much HOOH. ATom shows systematically large occurrences of low HOOH (50–200 ppt, especially central Pacific), indicating, perhaps, that convective or cloud scavenging of HOOH is more effective than is modeled. HCHO shows reasonable agreement in the Atlantic, but in both central and eastern Pacific, the modeled low end ($< 40$ ppt) is simply not seen in the ATom data. Also, the models are missing a strong HCHO peak at 300 ppt in the eastern Pacific, probably convection-related. Thus, in terms of these $HO_x$ precursors, the model climatologies appear to be at least as reactive as the ATom data.

While the ATom-1 data in Fig. 4 are limited to single transects, the model $NO_x$ discrepancies apply across the three tropical regions, and the simple chemical statistics for these flights alone are probably enough to identify measurement-model discrepancies. For the $HO_x$-related species, the models match the first-order statistics from ATom. In terms of using ATom statistics as a model metric, it is encouraging that where individual models tend to deviate from their peers, they also deviate from the ATom-1 PDs.

**4.3.3 Probability densities**

Mean profiles do not reflect the heterogeneity seen in Fig. 2, and so we also examine the PDs of the tropical reactivities (Fig. 5). The model PDs (colored lines connecting open circles at the center of each bin) are calculated from the 1 d statistics for mid-August (P2017) using the model blocks shown in Fig. S1. The model grid cells are weighted by air mass and cosine(latitude) and limited to pressures greater than 200 HPa. The ATom PDs (black lines connecting black open circles) are calculated from the 10 s data weighted by (but not averaged over) the number of points in each 10° latitude by 200 hPa pressure bin and then also by cosine (latitude) to compare with the models. In addition, a PD was calculated from the 1° by 200 m average grid-cell values in Fig. 2 (black Xs), and this is also cosine(latitude)-weighted. To check if the high reactivities in the eastern Pacific affected the whole Pacific PD, a separate PD using only central Pacific 10 s data was calculated (gray lines connecting gray open circles). The mean reactivities (ppb/d) from the models and ATom are given in the legend; note that these values disagree with some table data that are not cosine-weighted. The PD binning is shown by the open circles, and occurrences of off-scale reactivities are included in the last point.

The obvious discrepancy is with P-O3 in both Pacific and Atlantic basins. ATom data have very low occurrence of P-O3 $< 1$ ppb/d and a broad, almost uniform frequency ($\sim 0.1$) extending out to 4 ppb/d. This result is consistent with the mean profile errors (Fig. 3). The match for L-CH4 is very good in both basins, although the models have a greater occurrence in the middle 0.5–1.5 ppb/d range and reduced occurrence in the higher 1.5–2.5 ppb/d range. For L-O3, the match is very good and similar to L-CH4, although the Atlantic has a high frequency of L-O3 $> 6$ ppb/d that is not seen in the models (except GISS). The extreme eastern Pacific reactivities are seen in the mean values displayed in the legend (e.g., CPac with 1.29 ppb/d L-O3 versus ATom (i.e., CPac + EPac) with 1.54 ppb/d), but the PDs (gray circles versus black circles) resemble each other more closely than any of the models.

The ability to test a model's reactivity statistics with the ATom 10 s data is not obvious, but the PDs based on 1° latitude by 200 m altitude cells (the black Xs) is remarkably close to the PDs based on 2 km (horizontal) by 80 m (vertical) 10 s parcels. With the coarser resolution, we see a slight shift of points from the ends of the PD to the middle as expected, but we find once again, that the loss in high-frequency, below-model grid-cell resolution is not great. Both ATom-derived PDs more closely resemble each other than any model PD. Thus, current global chemistry models with resolutions of about 100 km by 400 m should be able to capture much of the wide range of chemical heterogeneity in the atmosphere, which for the oceanic transects is, we believe, adequately resolved by the 10 s ATom measurements. Perhaps more surprising, given the different mean profiles in Fig. 3, is that the five model PDs in Fig. 5 look very much alike. This points to some significant underlying difference between our current global chemistry models and the ATom observations.

[Figure]

**Figure 4.** Histograms of probability densities (PDs) of $NO_x$ (0–12 km, row 1), $NO_x$ (0–4 km, row 2), HOOH (0–12 km, row 3) and HCHO (0–12 km, row 4) for the three tropical regions (central Pacific, eastern Pacific, Atlantic). The ATom-1 data are plotted on top of the six global chemistry models' results for a day in mid-August and sampled as described in Fig. 3.

**5  Discussion and path forward**

**5.1  Major findings**

This paper opens a door for what the community can do with the ATom measurements and the derived products. ATom's mix of key species allows us to calculate the reactivity of the air parcels and hopefully may become standard for tropospheric chemistry campaigns. We find that the reactivity of the troposphere with respect to $O_3$ and $CH_4$ is dominated by a fraction of the air parcels but not by so small and infrequent a fraction as to challenge the ability of current CTMs to simulate these observations and thus be used to study the oxidation budgets. In comparing ATom results with modeled climatologies, we find a clear model error – missing $O_3$ production over the tropical oceans' lower troposphere – and traced it to the lack of $NO_x$ below 4 km. The occurrence of the same error over the central and eastern Pacific as well as the Atlantic Ocean makes this a robust model–measurement discrepancy.

Building our chemical statistics (PDs) from the ATom 10 s air parcels on a scale of 2 km by 80 m, we can identify the fundamental scales of spatial heterogeneity in tropospheric chemistry. Although heterogeneity occurs at the finest scales (such as seen in some 1 s observations), the majority of variability in terms of the $O_3$ and $CH_4$ budgets occurs across scales larger than neighboring 2 km parcels. The PDs measured in ATom can be largely captured by global models'

[Figure]

**Figure 5.** Probability densities (PDs, frequency of occurrence) for the ATom-1 three reactivities (rows: P-O3, L-O3, L-CH4 in ppb/d) and for the Pacific and Atlantic from 54° S to 60° N (columns left and right). Each air parcel is weighted as described in the text for equal frequency in large latitude–pressure bins and also by cosine(latitude). The ATom statistics are from the UCI model, using MDS-2 and revised RDS* protocol (HNO$_4$ and PAN damping). The full Pacific results (solid black) also show just the central Pacific (dashed gray). The six models' values for a day in mid-August are averaged over longitude for the domains shown in Fig. S1 and then cosine(latitude)-weighted. Mean values (ppb/d) are shown in the legend but are different from some tables where the cosine weighting is not applied. The PD derived from the ATom 10 s parcels binned at 1° latitude and 200 m altitude (shown for the tropics only in Fig. 2) is typical of a high-resolution global model and denoted by black Xs.

100 km by 200 m grid cells in the lower troposphere. This surprising result is evident by comparing the ATom 1D PDs – both species and reactivities – with those from the models' climatologies (Fig. 5). These comparisons show that the modeled PDs are consistent with the innate chemical heterogeneity of the troposphere as measured by the 10 s parcels in ATom. A related conclusion for biomass burning smoke particles is found by Schill et al. (2020), where most of the smoke appears in the background rather than in pollution plumes, and therefore much of the variability occurs on synoptic scales resolved by global models (see their Fig. 1 compared with Fig. 2 here).

**5.2 Opportunities and lessons learned**

As a quick look at the opportunities provided by the ATom data, we present an example based on the Wolfe et al. (2019) study, which used the F0AM model and semi-analytical arguments to show that troposphere HCHO columns (measurable by satellite and ATom) are related to OH columns (measured by ATom) and thus to CH$_4$ loss. Figure 6 extends the Wolfe et al. study using the individual air parcels and plotting L-CH4 (ppb/d) versus HCHO (ppt) for the three tropical regions where most of the CH$_4$ loss occurs. The relationship is linear, with slopes ranging from 4 to 6 per day, but the

[Figure]

**Figure 6.** Scatterplot of L-CH4 (ppb/d) versus HCHO (ppt) for ATom 1 in the three tropical regions shown in Fig. 3. The air parcels are split into the lower troposphere (0–4 km pressure altitude, red dots), where most of the reactivity lies, and the middle and upper troposphere (4–12 km, blue). A simple linear fit to all data is shown (thin black line), and the slope is given in units of 1 per day.

[Figure]

**Figure 7.** 2D frequency of occurrence (PDs in log ppt mole fraction) of HOOH vs. NO$_x$ for the tropical central Pacific for all four ATom deployments. The cross marks the mean (in log space), and the ellipse is fitted to the rotated PD having the smallest semi-minor axis. The semi-minor and semi-major axes are 2 standard deviations of PD in that direction. The ellipses from ATom-2 (red), ATom-3 (blue) and ATom-4 (dark green) are also plotted in the ATom-1 quadrant.

largest reactivities (0–4 km, 1–3 ppb/d) are not so well correlated with HCHO.

As is usual with new model intercomparison projects, we have an opportunity to identify model "features" and identify errors. In the UCI model, an error in the lumped alkane formulation (averaging alkanes C$_3$H$_8$ and higher) did not show up in P2018, where UCI supplied all the species, but when the ATom data were used, the UCI model became an outlier. Once found, this problem was readily fixed. The divergence

of the NCAR RDS results is likely due to the implementation of the RDS protocol where CCM values overwrite the MDS values. We identified this problem in P2018 and thought it was solved, but perhaps it is not. Inclusion of the F0AM model with its extensive hydrocarbon oxidation mechanism provided an interesting contrast with the simpler chemistry in the global CCM/CTMs. For a better comparison of the chemical mechanisms, we should have F0AM use 5 d of photolysis fields from one of the CTMs. The anomalous GISS results have been examined by a co-author, but no clear causes have been identified as of this publication. The problem goes beyond just the implementation of the RDS protocol, as it shows up in the model climatology (Figs. 4 and 5, also in P2017).

Decadal-scale shifts in the budgets of $O_3$ and $CH_4$ are likely to be evident through the statistical patterns of the key species, rather than simply via average profiles. The underlying design of ATom was to collect enough data to develop such a multivariate chemical climatology. As a quick look across the four deployments, we show the joint 2D PDs on a logarithmic scale as in P2017 for HOOH versus $NO_x$ in Fig. 7. The patterns for the tropical central Pacific are quite similar for the four seasons of ATom deployments, and the fitted ellipses are almost identical for ATom 2, 3 and 4. Thus, for these species in the central Pacific, we believe that ATom provides a benchmark of the 2016–2018 chemical state, one that can be revisited with an aircraft mission in a decade to detect changes in not only chemical composition, but also reactivity.

ATom identifies which "highly reactive" spatial or chemical environments could be targeted in future campaigns for process studies or to provide a better link between satellite observations and photochemical reactivity (e.g., E. Pacific mid-troposphere in August, Fig. 2). The many corollary species measured by ATom (not directly involved in $CH_4$ and $O_3$ chemistry) can provide clues to the origin or chemical processing of these environments. We hope to engage a wider modeling community beyond the ATom science team, as in H2018, in the calculation of photochemical processes, budgets, and feedbacks based on all four ATom deployments.

*Data availability.* The MDS-2 and RDS*-2 data for ATom 1, 2, 3 and 4 are presented here as core ATom deliverables and are now posted on the NASA ESPO ATom website (https://espo.nasa.gov/atom/content/ATom, Science team of the NASA Atmospheric Tomography Mission, 2021). This publication marks the public release of the reactivity calculations for ATom 2, 3 and 4, but we have not yet analyzed these data, and thus users should be aware and report any anomalous features to the lead authors via haog2@uci.edu and mprather@uci.edu. Details of the ATom mission and data sets are found on the NASA mission website (https://espo.nasa.gov/atom/content/ATom, last access: 13 September 2021) and in the final archive at Oak Ridge National Laboratory (ORNL; https://daac.ornl.gov/ATOM/guides/ATom_merge.html, last access: 13 September 2021). The MATLAB codes and data sets used in the analysis here are posted on Dryad (https://doi.org/10.7280/D1Q699, Guo, 2021).

*Supplement.* The supplement related to this article is available online at: https://doi.org/10.5194/acp-21-13729-2021-supplement.

*Author contributions.* HG, CMF, SCW and MJP designed the research and performed the data analysis. SAS, SDS, LE, FL, JL, AMF, GC, LTM and GW contributed original atmospheric chemistry model results. GW, MK, JC, GD, JD, BCD, RC, KM, JP, TBR, CT, TFH, DB, NJB, ECA, RSH, JE, EH and FM contributed original atmospheric observations. HG, CMF and MJP wrote the paper.

*Competing interests.* The contact author has declared that neither they nor their co-authors have any competing interests.

*Acknowledgements.* The authors are indebted to the entire ATom Science Team including the managers, pilots and crew, who made this mission possible. Many other scientists not on the author list enabled the measurements and model results used here. Primary funding of the preparation of this paper at UC Irvine was through NASA grants NNX15AG57A and 80NSSC21K1454.

*Financial support.* The Atmospheric Tomography Mission (ATom) was supported by the National Aeronautics and Space Administration's Earth System Science Pathfinder Venture-Class Science Investigations: Earth Venture Suborbital-2. Primary funding of the preparation of this paper at UC Irvine was through NASA (grant nos. NNX15AG57A and 80NSSC21K1454).

*Review statement.* This paper was edited by Neil Harris and reviewed by two anonymous referees.

**References**

Burkholder, J. B., Sander, S. P., Abbatt, J. P. D., Barker, J. R., Huie, R. E., Kolb, C. E., Kurylo, M. J., Orkin, V. L., Wilmouth, D. M., and Wine, P. H.: Chemical kinetics and photochemical data for use in atmospheric studies: evaluation number 18, Pasadena, CA, Jet Propulsion Laboratory, National Aeronautics and Space Administration, available at: http://hdl.handle.net/2014/45510 (last access: 13 September 2021), 2015.

Charlton-Perez, C. L., Evans, M. J., Marsham, J. H., and Esler, J. G.: The impact of resolution on ship plume simulations with NOx chemistry, Atmos. Chem. Phys., 9, 7505–7518, https://doi.org/10.5194/acp-9-7505-2009, 2009.

Douglass, A. R., Prather, M. J., Hall, T. M., Strahan, S. E., Rasch, P. J., Sparling, L. C., Coy, L., and Rodriguez, J. M.: Choosing meteorological input for the global modeling initiative assessment of high-speed aircraft, J. Geophys. Res.-Atmos., 104, 27545–27564, https://doi.org/10.1029/1999JD900827, 1999.

Eastham, S. D. and Jacob, D. J.: Limits on the ability of global Eulerian models to resolve intercontinental transport

of chemical plumes, Atmos. Chem. Phys., 17, 2543–2553, https://doi.org/10.5194/acp-17-2543-2017, 2017.

Griffiths, P. T., Murray, L. T., Zeng, G., Shin, Y. M., Abraham, N. L., Archibald, A. T., Deushi, M., Emmons, L. K., Galbally, I. E., Hassler, B., Horowitz, L. W., Keeble, J., Liu, J., Moeini, O., Naik, V., O'Connor, F. M., Oshima, N., Tarasick, D., Tilmes, S., Turnock, S. T., Wild, O., Young, P. J., and Zanis, P.: Tropospheric ozone in CMIP6 simulations, Atmos. Chem. Phys., 21, 4187–4218, https://doi.org/10.5194/acp-21-4187-2021, 2021.

Guo, H.: Heterogeneity and chemical reactivity of the remote Troposphere defined by aircraft measurements, Dryad [data set], https://doi.org/10.7280/D1Q699, 2021.

Hall, S. R., Ullmann, K., Prather, M. J., Flynn, C. M., Murray, L. T., Fiore, A. M., Correa, G., Strode, S. A., Steenrod, S. D., Lamarque, J.-F., Guth, J., Josse, B., Flemming, J., Huijnen, V., Abraham, N. L., and Archibald, A. T.: Cloud impacts on photochemistry: building a climatology of photolysis rates from the Atmospheric Tomography mission, Atmos. Chem. Phys., 18, 16809–16828, https://doi.org/10.5194/acp-18-16809-2018, 2018.

Heald, C. L., Coe, H., Jimenez, J. L., Weber, R. J., Bahreini, R., Middlebrook, A. M., Russell, L. M., Jolleys, M., Fu, T.-M., Allan, J. D., Bower, K. N., Capes, G., Crosier, J., Morgan, W. T., Robinson, N. H., Williams, P. I., Cubison, M. J., DeCarlo, P. F., and Dunlea, E. J.: Exploring the vertical profile of atmospheric organic aerosol: comparing 17 aircraft field campaigns with a global model, Atmos. Chem. Phys., 11, 12673–12696, https://doi.org/10.5194/acp-11-12673-2011, 2011.

Myhre, G., Shindell, D., and Pongratz, J.: Anthropogenic and Natural Radiative Forcing, in Climate Change 2013: The Physical Science Basis, IPCC WGI Contribution to the Fifth Assessment Report, Cambridge University Press, 659–740, https://doi.org/10.1017/CBO9781107415324.018, 2014.

Naik, V., Voulgarakis, A., Fiore, A. M., Horowitz, L. W., Lamarque, J.-F., Lin, M., Prather, M. J., Young, P. J., Bergmann, D., Cameron-Smith, P. J., Cionni, I., Collins, W. J., Dalsøren, S. B., Doherty, R., Eyring, V., Faluvegi, G., Folberth, G. A., Josse, B., Lee, Y. H., MacKenzie, I. A., Nagashima, T., van Noije, T. P. C., Plummer, D. A., Righi, M., Rumbold, S. T., Skeie, R., Shindell, D. T., Stevenson, D. S., Strode, S., Sudo, K., Szopa, S., and Zeng, G.: Preindustrial to present-day changes in tropospheric hydroxyl radical and methane lifetime from the Atmospheric Chemistry and Climate Model Intercomparison Project (ACCMIP), Atmos. Chem. Phys., 13, 5277–5298, https://doi.org/10.5194/acp-13-5277-2013, 2013.

Prather, M. J., Ehhalt, D., Dentener, F., Derwent, R., Dlugokencky, E. J., Holland, E., Isaksen, I., Katima, J., Kirchhoff, V., Matson, P., and Midgley, P.: Chapter 4 – Atmospheric Chemistry and Greenhouse Gases, Climate Change 2001: The Scientific Basis, Third Assessment Report of the Intergovernmental Panel on Climate Change, 239–287, 2001.

Prather, M. J., Zhu, X., Flynn, C. M., Strode, S. A., Rodriguez, J. M., Steenrod, S. D., Liu, J., Lamarque, J.-F., Fiore, A. M., Horowitz, L. W., Mao, J., Murray, L. T., Shindell, D. T., and Wofsy, S. C.: Global atmospheric chemistry – which air matters, Atmos. Chem. Phys., 17, 9081–9102, https://doi.org/10.5194/acp-17-9081-2017, 2017.

Prather, M. J., Flynn, C. M., Zhu, X., Steenrod, S. D., Strode, S. A., Fiore, A. M., Correa, G., Murray, L. T., and Lamarque, J.-F.: How well can global chemistry models calculate the reactivity of

short-lived greenhouse gases in the remote troposphere, knowing the chemical composition, Atmos. Meas. Tech., 11, 2653–2668, https://doi.org/10.5194/amt-11-2653-2018, 2018.

Rastigejev, Y., Park, R., Brenner, M. P., and Jacob, D. J.: Resolving intercontinental pollution plumes in global models of atmospheric transport, J. Geophys. Res.-Atmos., 115, D012568, https://doi.org/10.1029/2009JD012568, 2010.

Schill, G. P., Froyd, K. D., Bian, H., Kupc, A., Williamson, C., Brock, C. A., Ray, E., Hornbrook, R. S., Hills, A. J., Apel, E. C., and Chin, M.: Widespread biomass burning smoke throughout the remote troposphere, Nat. Geosci., 13, 422–427, https://doi.org/10.1038/s41561-020-0586-1, 2020.

Science team of the NASA Atmospheric Tomography Mission: ATom [data set], available at: https://espo.nasa.gov/atom/content/ATom, last access: 13 September 2021.

Stevenson, D. S., Dentener, F. J., Schultz, M. G., Ellingsen, K., Van Noije, T. P. C., Wild, O., Zeng, G., Amann, M., Atherton, C. S., Bell, N., and Bergmann, D. J.: Multimodel ensemble simulations of present-day and near-future tropospheric ozone, J. Geophys. Res.-Atmos., 111, D006338, https://doi.org/10.1029/2005JD006338, 2006.

Stevenson, D. S., Young, P. J., Naik, V., Lamarque, J.-F., Shindell, D. T., Voulgarakis, A., Skeie, R. B., Dalsoren, S. B., Myhre, G., Berntsen, T. K., Folberth, G. A., Rumbold, S. T., Collins, W. J., MacKenzie, I. A., Doherty, R. M., Zeng, G., van Noije, T. P. C., Strunk, A., Bergmann, D., Cameron-Smith, P., Plummer, D. A., Strode, S. A., Horowitz, L., Lee, Y. H., Szopa, S., Sudo, K., Nagashima, T., Josse, B., Cionni, I., Righi, M., Eyring, V., Conley, A., Bowman, K. W., Wild, O., and Archibald, A.: Tropospheric ozone changes, radiative forcing and attribution to emissions in the Atmospheric Chemistry and Climate Model Intercomparison Project (ACCMIP), Atmos. Chem. Phys., 13, 3063–3085, https://doi.org/10.5194/acp-13-3063-2013, 2013.

Stevenson, D. S., Zhao, A., Naik, V., O'Connor, F. M., Tilmes, S., Zeng, G., Murray, L. T., Collins, W. J., Griffiths, P. T., Shim, S., Horowitz, L. W., Sentman, L. T., and Emmons, L.: Trends in global tropospheric hydroxyl radical and methane lifetime since 1850 from AerChemMIP, Atmos. Chem. Phys., 20, 12905–12920, https://doi.org/10.5194/acp-20-12905-2020, 2020.

Stocker, T. F., Qin, D., Plattner, G. K., Tignor, M., Allen, S. K., Boschung, J., Nauels, A., Xia, Y., Bex, V., and Midgley, P. M.: Contribution of working group I to the fifth assessment report of the intergovernmental panel on climate change. Cambridge University Press, 33–115, 2013.

Tie, X., Brasseur, G., and Ying, Z.: Impact of model resolution on chemical ozone formation in Mexico City: application of the WRF-Chem model, Atmos. Chem. Phys., 10, 8983–8995, https://doi.org/10.5194/acp-10-8983-2010, 2010.

Voulgarakis, A., Naik, V., Lamarque, J.-F., Shindell, D. T., Young, P. J., Prather, M. J., Wild, O., Field, R. D., Bergmann, D., Cameron-Smith, P., Cionni, I., Collins, W. J., Dalsøren, S. B., Doherty, R. M., Eyring, V., Faluvegi, G., Folberth, G. A., Horowitz, L. W., Josse, B., MacKenzie, I. A., Nagashima, T., Plummer, D. A., Righi, M., Rumbold, S. T., Stevenson, D. S., Strode, S. A., Sudo, K., Szopa, S., and Zeng, G.: Analysis of present day and future OH and methane lifetime in the ACCMIP simulations, Atmos. Chem. Phys., 13, 2563–2587, https://doi.org/10.5194/acp-13-2563-2013, 2013.

Wofsy, S. C.: HIAPER Pole-to-Pole Observations (HIPPO): fine-grained, global-scale measurements of climatically important atmospheric gases and aerosols, Philos. T. R. Soc. A, 369, 2073–2086, https://doi.org/10.1098/rsta.2010.0313, 2011.

Wofsy, S. C., Afshar, S., Allen, H. M., Apel, E. C., Asher, E. C., Barletta, B., Bent, J., Bian, H., Biggs, B. C., Blake, D. R., Blake, N., Bourgeois, I., Brock, C. A., Brune, W. H., Budney, J. W., Bui, T. P., Butler, A., Campuzano-Jost, P., Chang, C.S., Chin, M., Commane, R., Correa, G., Crounse, J. D., Cullis, P. D., Daube, B.C., Day, D. A., Dean-Day, J. M., Dibb, J. E., DiGangi, J. P., Diskin, G. S., Dollner, M., Elkins, J. W., Erdesz, F., Fiore, A. M., Flynn, C. M., Froyd, K. D., Gesler, D. W., Hall, S. R., Hanisco, T. F., Hannun, R. A., Hills, A. J., Hintsa, E. J., Hoffman, A., Hornbrook, R. S., Huey, L. G., Hughes, S., Jimenez, J. L., Johnson, B. J., Katich, J. M., Keeling, R. F., Kim, M. J., Kupc, A., Lait, L. R., Lamarque, J.-F., Liu, J., McKain, K., Mclaughlin, R. J., Meinardi, S., Miller, D. O., Montzka, S. A, Moore, F. L., Morgan, E. J., Murphy, D. M., Murray, L. T., Nault, B. A., Neuman, J. A., Newman, P. A., Nicely, J. M., Pan, X., Paplawsky, W., Peischl, J., Prather, M. J., Price, D. J., Ray, E. A., Reeves, J. M., Richardson, M., Rollins, A. W., Rosenlof, K. H., Ryerson, T. B., Scheuer, E., Schill, G. P., Schroder, J. C., Schwarz, J. P., St.Clair, J. M., Steenrod, S. D., Stephens, B. B., Strode, S. A., Sweeney, C., Tanner, D., Teng, A. P., Thames, A. B., Thompson, C. R., Ullmann, K., Veres, P. R., Vieznor, N., Wagner, N. L., Watt, A., Weber, R., Weinzierl, B., Wennberg, P. O., Williamson, C. J., Wilson, J. C., Wolfe, G. M., Woods, C. T., and Zeng L. H.: ATom: Merged Atmospheric Chemistry, Trace Gases, and Aerosols, ORNL DAAC [data set], Oak Ridge, Tennessee, USA, https://doi.org/10.3334/ORNLDAAC/1581, 2018.

Wolfe, G. M., Nicely, J. M., Clair, J. M. S., Hanisco, T. F., Liao, J., Oman, L. D., Brune, W. B., Miller, D., Thames, A., Abad, G. G., and Ryerson, T. B.: Mapping hydroxyl variability throughout the global remote troposphere via synthesis of airborne and satellite formaldehyde observations, P. Natl. Acad. Sci. USA, 116, 11171–11180, https://doi.org/10.1073/pnas.1821661116, 2019.

Young, P. J., Archibald, A. T., Bowman, K. W., Lamarque, J.-F., Naik, V., Stevenson, D. S., Tilmes, S., Voulgarakis, A., Wild, O., Bergmann, D., Cameron-Smith, P., Cionni, I., Collins, W. J., Dalsøren, S. B., Doherty, R. M., Eyring, V., Faluvegi, G., Horowitz, L. W., Josse, B., Lee, Y. H., MacKenzie, I. A., Nagashima, T., Plummer, D. A., Righi, M., Rumbold, S. T., Skeie, R. B., Shindell, D. T., Strode, S. A., Sudo, K., Szopa, S., and Zeng, G.: Pre-industrial to end 21st century projections of tropospheric ozone from the Atmospheric Chemistry and Climate Model Intercomparison Project (ACCMIP), Atmos. Chem. Phys., 13, 2063–2090, https://doi.org/10.5194/acp-13-2063-2013, 2013.

Young, P. J., Naik, V., Fiore, A. M., Gaudel, A., Guo, J., Lin, M. Y., Neu, J. L., Parrish, D. D., Rieder, H. E., Schnell, J. L., and Tilmes, S.: Tropospheric Ozone Assessment Report: Assessment of global-scale model performance for global and regional ozone distributions, variability, and trends, Elementa, 6, 10, https://doi.org/10.1525/elementa.265, 2018.

Yu, K., Jacob, D. J., Fisher, J. A., Kim, P. S., Marais, E. A., Miller, C. C., Travis, K. R., Zhu, L., Yantosca, R. M., Sulprizio, M. P., Cohen, R. C., Dibb, J. E., Fried, A., Mikoviny, T., Ryerson, T. B., Wennberg, P. O., and Wisthaler, A.: Sensitivity to grid resolution in the ability of a chemical transport model to simulate observed oxidant chemistry under high-isoprene conditions, Atmos. Chem. Phys., 16, 4369–4378, https://doi.org/10.5194/acp-16-4369-2016, 2016.

Zhuang, J., Jacob, D. J., and Eastham, S. D.: The importance of vertical resolution in the free troposphere for modeling intercontinental plumes, Atmos. Chem. Phys., 18, 6039–6055, https://doi.org/10.5194/acp-18-6039-2018, 2018.

---

## Author Response (AR2)

Dear editor

Please find the enclosed revision of our manuscript "Significant formation of sulfate aerosols contributed by the heterogeneous drivers of dust surface" (Manuscript Number: acp-2022-227). We thank the valuable comments from the reviewers and editor, which have greatly improved the manuscript. The point-by-point replies to the comments are attached on the following pages.

Thank you for your consideration.

Sincerely yours,

Liwu Zhang

Shanghai Key Laboratory of Atmospheric Particle Pollution and Prevention, Department of Environmental Science and Engineering, Fudan University

Shanghai, 200433, China

E: zhanglw@fudan.edu.cn

Referee #2

Reviewer comment: Is chemistries a real word?

Original author response: As a real word, it was widely utilized in the research papers of this field (e.g. doi.org/10.1021/cr500501m; doi.org/10.1021/cr5003485).

Editor comment: 'Chemistry' is defined as a branch of science that studies the composition and transformation of molecules and materials (cf e.g. Wikipedia). Therefore, in this context, it should not be used at all, but should be replaced by 'chemical processes' (or 'chemical composition' where appropriate) throughout the manuscript.

Revised author response:

Thanks for your comments.

  We have replaced "chemistries" by the more appropriate words including "chemical processes", "processes", "pathways", and "reactions" throughout the revised manuscript. The following sentences are listed as examples:

  Line 115-117: "*Hereby, upon understanding the driving factors and driving force of the airborne dust surface, this work compared dust heterogeneous pathways with the gas- and aqueous-phase ones with respect to the formation rate of sulfate and atmospheric lifetime of $SO_2$.*"

  Line 446-447: *"During nighttime, the gas-phase, aqueous-phase and heterogeneous processes explain 31.6, 39.8 and 28.6% of secondary sulfate, respectively (Fig. 5b)."*

Reviewer comment: Figure 3: why does the reactive uptake coefficient drastically increases after particle acidity reached ~4.5/5?

Original author response: Figure 3 displays the particle acidity-dependent reactive uptake coefficients (γ) for the dust-mediated heterogeneous chemistry. We have discussed the dramatically increased γ when pH exceeds 4.5 or 5.0, in the revised manuscript (line 367-369). "The γ of $NO_2$, $O_3$, HOCl, HOBr and $CH_3COOOH$ presents positive dependence toward pH, in accordance with the evolution of the effective Henry's law constant for the studied sulfur species, as theoretically illustrated by Eq. (5)."

Editor comment: I do not understand this response. A few lines above you state that "the dust mediated heterogeneous oxidation can be assumed to primarily proceed in the surface water layers" but yet you ascribe a significant impact on g by the term that describes the aqueous phase diffusion, i.e. the second term in the right-hand side in Eq. -5. What are the contributions to the total $\gamma$ of the mass accommodation term ($1/\alpha$) and the aqueous

phase diffusion and reaction term [v/(4HRT($D_a k_{chem}$)$^{-1}$)/$f_r$]? A figure should be included to demonstrate that indeed the second term increases significantly at increasing pH.

Revised author response:

We have explained the evaluation methodology for dust-mediated heterogeneous processes, and then discussed the absolute level and pH dependence of dust-mediated heterogeneous γ. The revisions are shown in detail as follows.

First, Eq. (5) quantitatively describes the heterogeneous sulfate formation mediated by dust surface. We have added its definition into the revised manuscript (line 246-248). *"The terms on the right-hand side (RHS) of Eq. (5) show the two contributions to the overall resistance to dust-mediated heterogeneous uptake: mass accommodation at surface, diffusion and reaction in surface liquid water layers."*

Second, in contrast to the Henry's law constant for $SO_2$ that quantifies its gas-liquid equilibrium, the H* in Eq. (5) refers to the effective Henry's law constant for the studied $S_{(IV)}$ specie(s). In other words, H* is considered to quantify the total dissolved sulfur species in the water layers of dust surface. We have emphasized the definition of H* in the revised manuscript (line 244-245, line 248). *"$H^*$ is the effective Henry's law constant for the studied $S_{(IV)}$ specie(s) (M atm$^{-1}$), and is jointly determined by the gas-liquid equilibrium of $SO_2$ and the ionization equilibriums of dissolved sulfur species" "The $S_{(IV)}$ species include $SO_2·H_2O$, $HSO_3^-$ and $SO_3^{2-}$."*

Last, the first term of the RHS of Eq. (5), mass accommodation (α), is several orders of magnitude higher than the calculated dust-mediated γ and no pH dependence has been reported for it. Thus, the absolute level and pH dependence of the dust-mediated heterogeneous γ is largely associated with the second term of the RHS of Eq. (5). We have rewritten the relevant paragraph in the revised manuscript (line 399-409). *"Figure 4 presents the pH-dependent γ for natural dust-mediated heterogeneous pathway, along with the experimental data of natural dust-driven heterogeneous pathway. The dust-mediated γ is orders of magnitude lower than the α of $SO_2$ (~ 0.14 under the experimental temperature of 296.8 K). By definition, α is the probability that a $SO_2$ molecule striking a liquid surface finally enters into the liquid phase, whereas γ is the sulfate formation rate normalized by the total surface collision rate of $SO_2$ (Jacob, 2000; Davidovits et al., 2006). That is, γ involves all the uptake processes, including the mass accommodation at surface (Seinfeld and Pandis, 2016). Moreover, unlike the pH-independent α, γ varies with pH, somewhat coinciding to the evolution of dissolved sulfur species that can be quantitatively described by the H* of $SO_2$. The γ values of $H_2O_2$, $CH_3OOH$ and $CH_3COOOH$ are pH-independent under specific conditions, in relation to the acid-catalyzed rate-limiting steps relevant to these peroxides (Lind et al., 1987; Liu et al., 2021a). Generally, the dust-mediated heterogeneous γ is largely determined by the diffusion and reaction processes in the water layers of dust surface, as characterized by the second item on the RHS of Eq. (5)."*

Reviewer comment: For daytime chemistry, did you consider the possibility of the formation of sulfate radicals on surfaces with high Ti content?

Original author response: We have considered the impact of sulfate radical on the heterogeneous formation of sulfate. Necessary discussions are added into the revised manuscript (line 266-267). "Sulfate radical ($SO_4^{\bullet-}$) is generated by the presence of abundant •OH and participates into the oxidation events (Antoniou et al., 2018; Kim et al., 2019; Li et al., 2020b)."

Editor comment: This response and revision is not sufficient to discuss the potential role of sulfate radicals. A discussion on possible chemical mechanisms of the observed processes should be added including the possibility of oxidation by the sulfate radical. You can either add this to section 3.3. or dedicate a full new section to this discussion.

Revised author response:

We have provided more discussions on reaction mechanism, including the possible effects of sulfate radical.

    In this manuscript, Sect. 3.1 identified the dominate dust surface drivers that influence the heterogeneous sulfate formation, whereas Sect. 3.3 performed comparison among diverse atmospheric oxidation pathways to figure out the predominate sulfate contributors. Relative to Sect. 3.3, Sect. 3.1 is believed to be more appropriate to display the reaction mechanism details. In order to make Sect. 3.1 more accessible to our readers, we enriched it by adding more knowledges on reaction mechanism. In addition, a scheme has been drawn to illustrate the possible chemical mechanism of dust-driven heterogeneous oxidation (Scheme 1 in manuscript, copied as follows).

    By the first paragraph of Sect. 3.1, we claimed the close association between correlation analysis and mechanism illustration (line 269-270). *"Correlation analysis is performed to identify the dust surface drivers (Fig. 1), based on which the reaction mechanism of dust heterogeneous pathway can be better understood (Scheme 1)."*

    The following contents, some of which are newly organized, are displayed in Sect. 3.1 to illustrate the relevant reaction mechanism details.

[revised manuscript text omitted]

Referee #3:

Reviewer comment: Line 373-374: Earlier authors assert that the sulfate formation is more during nighttime and less during the daytime, but this statement is opposite and confusing. Can the author clarify this?

Original author response: In the real atmosphere, the sulfate concentration during nighttime could be comparable to or even exceed that during daytime, as normally explained by the different meteorological conditions. We have explained the relevant details in the revised manuscript (line 396-401) to make our calculation results more reasonable. "It is worthwhile to mention that, in the real atmosphere, the observed sulfate concentration during nighttime may be comparable to or exceed that during daytime, which can be explained by the higher nocturnal humidity facilitating the liquid oxidations or the lower boundary layer causing the adverse diffusion conditions (Liu et al., 2017c; Tutsak and Koçak, 2019; Li et al., 2020c). The relevant meteorological factors were not considered by this comparison model, and the current results emphasized the different sulfate formation potentials through kinetic regime."

Editor comment: I still got confused what you compare in this section vs to the discussion earlier in the manuscript. Please reorganize Section 3.3 such that it becomes clear that these are estimates based on a model for which you assume ambient conditions whereas previously you discussed experimental lab data. It might help if you refer in the discussion of your experimental data to 'illuminated' and 'dark' conditions only and in the discussion of model results for atmospheric conditions to 'day' and 'night'.

Revised author response:

We have revised the relevant contents to emphasize the addition of experimental data into the newly developed comparison model. And the results from laboratory research and modeling simulation have been distinguished in a better way.

First, at the start of Sect. 3.3, we introduced the combination of laboratory and modeling works (line 430-433). *"Figure 5 compares the sulfate formation rates of diverse atmospheric oxidation pathways by a newly developed comparison model. Based on the parameterization scheme in model and the experimental results discussed above, the total sulfate formation rates are summed to be 0.795 μg m$^{-3}$ h$^{-1}$ during nighttime and 5.179 μg m$^{-3}$ h$^{-1}$ during daytime, under the acidity of natural dust (Fig. 5a and e)."* Moreover, we have reorganized Sect. 3.3 to make it clearer and more readable. More revisions are highlighted in blue in the revised manuscript.

Second, the results obtained from laboratory experiments and modeling simulations have been described by different characteristic words. We have checked through the manuscript, exampled as follows.

(1) Description of experimental results (line 376-377): *"The γ values for the natural dust-driven heterogeneous reactions are calculated to be 6.08 × 10$^{-6}$ under dark condition and 1.14 × 10$^{-5}$ under illuminated condition (Fig. 3a).*

(2) Description of modeling results (line 450-453): *"The diurnal sulfate formation rates of gas-phase, aqueous-phase and heterogeneous processes are respectively 9.4, 6.8 and 3.0 times greater than the nocturnal levels, indicating that the oxidations in gaseous and liquid media could be more kinetically susceptible to the occurrence of sunlight than those relevant to the humidified gas-solid interface."*

Last, according to the question raised by the reviewer, we noticed some documented atmospheric observations where the sulfate concentrations during nighttime approach or even exceed those during daytime, in contrast to the modeling data herein. In order to better explain such discrepancy and emphasize the feature of the current comparison model, we rewrote the relevant sentences in the revised manuscript (line 434-438). *"It is worthwhile to mention that, the nocturnal sulfate concentration was reported to approach or exceed the subsequent diurnal level, as explained by the higher nocturnal humidity facilitating the liquid oxidations or the lower boundary layer at night causing the adverse diffusion conditions (Liu et al., 2017c; Tutsak and Koçak, 2019; Li et al., 2020c). Meteorological factors like humidity are not considered by the current model, which emphasizes the comparison of diverse pathways through kinetic regime."*

Reviewer comment: Line 380-390: In addition to the temperature dependence, did the authors consider the change of relative humidity during the day and night can also contribute to the heterogeneous chemistry reaction with sulfate?

Original author response: We attempted to carry out the heterogeneous laboratory experiments under the higher relative humidity (RH) relative to the current setting. However, the presence of surface adsorbed water makes the Gaussian/Lorentzian deconvolution difficult to be performed, as reflected by the large uncertainties in determining the relative abundance of $SO_2 \cdot H_2O$ and $SO_3^{2-}$. Accordingly, the RH of 50% was utilized for the experiments. In the revised manuscript (line 654-656), we have emphasized the importance of various meteorological factors when discussing the atmospheric relevance of heterogeneous chemistry, as shown follows. "In addition, the influence of meteorological factors, like temperature, humidity and irradiance, on the atmospheric relevance of heterogeneous chemistry warrants further research. "

Editor comment: I understand that you cannot do any quantitative estimates on the influence of humidity. However, a statement on the extent to which relative humidity would change particle radius and aqueous volume and therefore sulfate formation should be added.

Revised author response:

A new paragraph has been added into "Conclusions and implications" section to discuss the possible effects of ambient humidity on the atmospheric relevance of dust heterogeneous oxidation (line 692-701). The new paragraph is shown as follows.

*"Meteorological factors impact the atmospheric relevance of dust heterogeneous pathway. Taking humidity as an example, on dust surface, the heterogeneous reaction of $SO_2$ is humidity-dependent, and the exact dependence varies with the type of dust and the condition of reaction (Huang et al., 2015; Park et al., 2017; Urupina et al., 2022). The uptake of gas-phase oxidants over dust surface is also influenced by humidity (Kumar et al., 2014). For aerosol droplet, increased humidity elevates its liquid volume and radius (Wu et al., 2018; Ding et al., 2019). Aerosol liquid water serves as an efficient medium for multiphase reactions (Wu et al., 2018; Yue et al., 2019), whereas radius is negatively associated with the sulfate formation at droplet interface (Hung et al., 2018; Wang et al., 2021a; Chen et al., 2022). Furthermore, droplet acidity decreases as the liquid volume increases, thereby increasing the ionization of dissolved sulfur species, followed by the increased sulfate formation rate (Yue et al., 2019; Jin et al., 2020; Gao et al., 2022). Overall, how meteorological factors influence the relative importance of gas-phase, aqueous-phase and heterogeneous processes warrants further research."*

Additional editor comments

l. 32: What do you mean by 'in mainstream'?

Response:

We have revised this description based on the recent publications (line 32-35), as shown follows.

*"In addition, the reported oxidation channels were compared with each other by aerosol observations or modeling investigations (Berglen et al., 2004; Sarwar et al., 2013; He et al., 2018; Ye et al., 2018; Fan et al., 2020; Tao et al., 2020; Zheng et al., 2020; Song et al., 2021; Tilgner et al., 2021; Liu et al., 2021b; Gao et al., 2022; Wang et al., 2022a; Ye et al., 2022)."*

l. 35: What do you mean by 'aimed liquid reaction' as opposed to 'aqueous phase reaction'? Do you imply reactions in liquid non-aqueous phases?

Response:

The phrase "aimed liquid reaction" referred to the newly discovered aqueous-phase reaction that was ready for the comparison research. To avoid misconception, we have rewritten this sentence in the revised manuscript (line 35-36), as shown follows.

*"In general, these studies emphasized the importance of certain newly discovered aqueous-phase process or compared the contributions from the documented gas- and aqueous-phase pathways."*

l. 37: What do you mean by 'gas-solid interactions'? What solid phase are you referring to? In the lines above, you refer to reactions on or in liquid phases.

Response:

The purpose of this introduction paragraph is to emphasize the limited knowledges on the sulfate formation relevant to aerosol surface, including both gas-liquid and gas-solid (mostly humidified) interfaces. We have revised this sentence in the revised manuscript (line 36-38), as shown follows.

*"Nevertheless, heterogeneous reaction was scarcely involved in discussion, thus hindering the deeper understanding of the atmospheric relevance of aerosol surfaces."*

l. 64-66: This sentence is too convoluted and mixes multiple processes and effects. The study by Clegg and Abbatt was performed on ice, Wu et al explored uptake on calcium carbonate and Wang et al. reactivity on hematite. Either split this sentence into three describing the essential information from each study or remove.

Response:

In the revised manuscript (line 65-67), we attempted to explain the temperature effects by several sentences, as shown follows.

*"The reversible adsorption of $SO_2$ is believed to be exothermic (Clegg and Abbatt, 2001a). In contrast, there were positive temperature dependences observed for $CaCO_3$ below 250 K over the entire reaction (Wu et al., 2011) and for $Fe_2O_3$ within 284-318 K during the initial reaction stage (Wang et al., 2018a)."*

l. 68: What do you mean by 'negative light effect'?

Response:

The phrase "negative light effect" referred to the decreased heterogeneous reactivity of iron oxide by the presence of stimulated solar irradiation. We have performed revision to make the sentence clearer, as shown follows (line 67-70).

*"Light irradiation normally accelerates the transformation of (bi)sulfites to (bi)sulfates (Li et al., 2010; Nanayakkara et al., 2012; Han et al., 2021), while iron oxides may undergo photoreactive dissolution and thus exhibit weaker reactivity under sunlight (Fu et al., 2009)."*

l. 73-74: 'In terms of...' – what does this sentence say?

Response:

The original sentence attempted to introduce the particle physical properties that influence the heterogeneous reaction of $SO_2$. We have rewritten this sentence to make the contents more readable (line 76-78), as shown follows.

*"The particle physical properties, including size (Baltrusaitis et al., 2010; Zhang et al., 2016), morphology (Li et al., 2019) and crystal structure (Yang et al., 2017), display varied impacts on heterogeneous reaction."*

l. 75: 'is more efficient' for what?

Response:

We have revised this sentence to highlight the higher heterogeneous reactivity of $Fe_2O_3$ than $Al_2O_3$ and $SiO_2$ (line 78-80), as shown follows.

*"When considering the particle chemical properties, $Fe_2O_3$ is more active than $Al_2O_3$ and $SiO_2$ in the heterogeneous uptake of $SO_2$ and formation of sulfate (Chughtai et al., 1993; Zhang et al., 2006; He et al., 2014)."*

l. 77: What do you mean by 'dust community'?

Response:

We have rewritten the sentence to explain that the addition of one certain atmospherically relevant constituent onto dust surface would accelerate the heterogeneous uptake of $SO_2$. More recent publications have been added into the content (line 80-82), as shown follows.

*"Furthermore, the presence of moderate nitrate (Kong et al., 2014; Du et al., 2019), or $Al_2O_3$ (Wang et al., 2018b), or surfactant (Zhanzakova et al., 2019), or oxalate (Li et al., 2021b), or (bi)carbonate (Liu et al., 2022) on dust surfaces could favor the heterogeneous kinetics under specific conditions."*

l. 108: correct 'derisble'

Response:

We have corrected the word to "desirable" (line 113).

l. 119: replace 'Technical Route' by 'Experimental methodology' or something similar.

Response:

The section title has been replaced by "Methodology overview".

l. 125: In the sentence starting 'In the dust-mediated...' is a verb missing.

Response:

We have revised this sentence in the manuscript (line 130-131), as shown follows.

*"In dust-mediated mode, dust surface functions as a reaction medium that supports the interaction between adsorbed oxidants and $SO_2$."*

l. 140: Please clarify this sentence. Why don't you state simply the relevant temperature range as being representative for the warm season?

Response:

We have provided an approximate temperature range concluded from the various temperature levels determined by the atmospheric observations (line 145-146), as shown follows.

"The measurements conducted in warm seasons were considered in priority to correspond the experimental temperature. *The relevant temperature range is 293-303 K in representative of warm season.*"

l. 178/179: Which 'gas and aqueous phase parameters' are you referring here to?

Response:

We have rewritten this sentence to make it more readable (line 184-186), as shown follows.

*"The kinetic parameters involved in the gas- and aqueous-phase oxidation pathways (listed by Sect. 2.2) were corrected by the experimental temperature as much as possible."*

l. 195: The reference to Hennigan et al. 2015 would be much better placed in this context here than in l. 190 as it was shown in their study that $HSO_4^-$ and $SO_4^{2-}$ are not sufficient to calculate acidity.

Response:

We have read the publication by Hennigan and performed more discussions on the particle acidity determination, as shown follows.

First, considering that the proxy methods reviewed by Hennigan et al. (2015) may not accurately estimate the pH of clay mineral, we cited this publication at the beginning of paragraph.

Furthermore, we noted the contents illustrating the limitation of pH calculation by ion balance. Such disadvantage would not affect the results from this work because the relative abundance of $S_{(IV)}$ species were derived from the infrared measurements. We have added discussions into the revised manuscript (line 204-208).

*"As noted by Keene et al. (2004) and Hennigan et al. (2015), estimating the pH of airborne aerosol by ion balance may be difficult because it is unable to distinguish between free and undissociated $H^+$ (e.g. protons associated with $HSO_4^-$ and $HSO_3^-$). Herein, the relative abundance of $S_{(IV)}$ species is derived from the different infrared absorption*

*signals relevant to atomic sulfur. Thus, the current method is recommended for the pH determination of humidified gas-solid interface."*

l. 212: what do you mean by 'standard item'? Is it the 'standard mixture' or 'standard solution'?

Response:

The "standard item" referred to the $Na_2SO_4$ powders that were mixed with the prepared clay mineral particles as sulfate standard. We have revised the sentence in the manuscript (line 223-224), as shown follows.

*"Because $Na_2SO_4$ was thoroughly mixed with the particles, $S_{BET}$ was used to calculate γ."*

l. 255: replace 'corrects' with 'correlates'

Response:

We have revised the spelling error.

l. 259: Why only 'nocturnal heterogeneous reaction'? Why does this not happen during the day? Could 'nocturnal' be omitted in this sentence?

Response:

The Al-bearing mineral constituents, such as $Al_2O_3$, would block the active sites of other constituents under both dark and illuminated conditions. We have deleted "nocturnal" and rewritten the sentence (line 275-277), as shown follows.

*"Conversely, $Al_2O_3$ presents weaker heterogeneous reactivity than $Fe_2O_3$ (Zhang et al., 2006; Yang et al., 2018b; Xu et al., 2021), and may hinder heterogeneous reaction by blocking the active sites of other mineral constituents (Wang et al., 2018b)."*

l. 263 – 291: This text should not be in the result section but belongs in the introduction where it is partially already cited. Please condense this paragraph and either move relevant parts into the discussion of your results (which are not presented at this place of the paper yet) or to the introduction.

Response:

We have reorganized the relevant contents as suggested.

In Sect. 3.1, in the description contents relevant to Fig. 1, each paragraph introduces the results of correlation analysis at first, followed by the discussion on the possible reaction mechanisms. We combined the result and discussion contents due to their close associations in illustrating the dominate dust surface drivers. We have emphasized this attempt by the first paragraph of Sect. 3.1 in the revised manuscript (line 269-270). *"Correlation analysis is performed to identify the dust surface drivers (Fig. 1), based on which the reaction mechanism of dust heterogeneous processes can be better understood (Scheme 1)."*

Moreover, we have rewritten or shorten the sentences that read like those in introduction section. For instance, each paragraph is now started by the correlation analysis result rather than the cited knowledge, as shown follows. We believe that the current text is appropriate for better understanding the primary elements that influence the heterogeneous sulfate formation over dust surfaces.

Start of the second paragraph of Sect. 3.1: *"The mineral drivers are investigated at first (Fig. 1a). Under dark condition, the sulfate production rate correlates positively with Fe, while presents negative dependence against Al."*

Start of the third paragraph of Sect. 3.1: *"The sulfate yield enhanced by solar irradiation associates positively with the abundance of Ti or Al."*

Start of the fourth paragraph of Sect. 3.1: *"No correlation can be observed between the abundance of element and the sulfate production rate of photoreaction."*

Start of the fifth paragraph of Sect. 3.1: *"The driving effects of water-soluble ions are further studied (Fig. 1b). Herein, $Na^+$ and $Cl^-$ are observed to present positive impacts on the sulfate formation under dark condition."*, followed by the illustration on the negative effects of water-soluble ions: *"Moreover, there are negative associations between the photoinduced sulfate enhancement and the abundances of $Na^+$, $K^+$, $Ca^{2+}$."*

Start of the sixth paragraph of Sect. 3.1 *"Apart from the dust surface drivers and inhibitors, sulfate radical ($SO_4^{\cdot-}$) may also participate into the heterogeneous event."*

l. 350: This sentence is not clear. Please clarify what you want to say here.

Response:

We have rewritten the sentence in the revised manuscript (line 285).

*"Particle acidity is further calculated to discuss the driving force of dust surface for sulfate formation."*

l. 350/351: 'no significant correlations can be found between γ and pH' contradicts your later discussion and findings in Figure 3. Please clarify.

Response:

The γ in Fig. 3 (previously denoted as Fig. 2) characterizes the kinetics of dust-driven heterogeneous pathway. Statistical analysis indicates that no significant correlation can be found between the dust-driven heterogeneous γ and particle acidity (pH). Thus, the absolute acidity of reacted dust may be primarily dependent on the dust's basic nature. To avoid misunderstanding, we have rewritten this sentence by emphasizing the dust-driven process (line 387-389), as shown follows.

*"Because no significant correlation can be found between the γ for dust-driven sulfate formation and the pH of reacted clay mineral, the absolute acidity level depends largely on the basic nature of dust."*

By contrast, the γ in Fig. 4 (previously denoted as Fig. 3) evaluates the kinetics of dust-mediated heterogeneous pathway. The γ for dust-mediated heterogeneous process is pH-dependent, which relates to the effective Henry's law constant for the studied $S_{(IV)}$ specie(s), as discussed above. We have emphasized this point in the revised manuscript (line 399), as shown follows.

*"Figure 4 presents the pH-dependent γ for natural dust-mediated heterogeneous pathway"*

l. 359/360: Are three digits of the pH value significant figures? What is the variability in this value?

Response:

The pH level was previously calculated with three significant digits. For consistency, two significant digits are shown for all pH values in the revised manuscript. Moreover, standard errors have been added into Fig. 3 to make it more accurate. The revised figure is shown as follows.

[Figure]

**Figure 3.** Analysis results of the *in-situ* infrared spectra recorded for the heterogeneous reaction of $SO_2$ on clay minerals and natural dust.

(a) Reactive uptake coefficients ($\gamma$) for the heterogeneous formation of sulfate. (b) Particle acidity (pH) of the reacted particle samples. The dark (grey square) and light (yellow circle) conditions were both considered. Dots represent the results of clay minerals, and those of natural dust are showed by the lines. All error bars represent 1 SD.

l. 430: replace 'great heterogeneous performances' by 'significant contributions by heterogeneous sulfate formation' (or similar)

Response:

We have revised the sentence (line 465-467), as shown follows.

   *"The dust-driven heterogeneous sulfate formation is mainly attributed to IMt that owns the largest proportion in dust community, followed by NAu and SWy with relatively significant contributions by heterogeneous sulfate formation (Fig. 5d and h)."*

l. 446 – 448: The sentence starting 'As proved by ...' seems out of place here. Either remove or clarify its connection to the text in this paragraph.

Response:

We have rewritten this sentence in the revised manuscript (line 486-489), as shown follows.

*"The important role of dust in sulfate formation was confirmed by atmospheric observation research. For example, secondary sulfate was observed to accumulate on the dust-dominant super-micron particles collected in the North China Plain, and the mass fraction of coarse-mode sulfate dramatically increased during the evolutionary stages of haze episode (Xu et al., 2020)."*

l. 454-461: 1) Refer to Figure 6 at the beginning of this text , 2) Be more specific in the text, rather than just saying longest, second longest etc lifetime.

Response:

First, we have referred to Fig. 7 (previously numbered as Fig. 6) at the beginning of this paragraph (line 491), as shown follows.

*"Figure 7 compares the lifetimes of $SO_2$ influenced by the diverse atmospheric oxidation pathways."*

Furthermore, we have specified the discussions, and the revised content is shown as follows.

*"Figure 7 compares the lifetimes of $SO_2$ influenced by the diverse atmospheric oxidation pathways. Calculations of lifetimes can be useful in estimating how long the $SO_2$ is likely to remain airborne before it is removed from the atmosphere (Seinfeld and Pandis, 2016). Theoretically, the lifetime of $SO_2$ determined by heterogeneous reaction is negatively correlated with $\gamma$ and $S_p$, and $S_p$ is positively associated with the concentration and $S_{BET}$ of dust, as respectively described by Eq. (12) and (10). As a result, the dust with greater heterogeneous reactivity, or higher atmospheric loading, or larger $S_{BET}$ is prone to cause the shorter lifetime. During nighttime, IMt causes the shortest $SO_2$ lifetime (46.90 days) due to its highest concentration relative to the other clay minerals. The relatively large heterogeneous uptake capacity of NAu and SWy link to the second and third shortest $SO_2$ lifetimes (221.30 and 260.12 days, respectively). On the other hand, the weakest heterogeneous reactivity of KGa leads to the second longest lifetime (1758.65 days), and the longest result caused by CCa (2256.25 days) can be interpreted by its lowest $S_{BET}$ that causes the lowest $S_p$. The presence of solar irradiation alters the lifetime ranking: IMt (20.39 days) < SWy (256.96 days) < NAu (437.85 days) < KGa (597.54 days) < CCa (1488.46 days), as influenced by the different photoactivities of the clay minerals."*

Figure 4: 1) Make labels on panels b, c, d, f, g, h larger. 2) Explain the colors in panels a and e. Are they same as in panels b and f?

Response:

We have revised this figure and its caption, as copied follows.

[Figure]

**Figure 5.** Contributions of diverse atmospheric oxidation pathways to secondary sulfate aerosols.

Gas-phase oxidation can be induced by hydroxyl radical (OH), stabilized Criegee intermediates (CIs), as well as the nitrate radical ($NO_3$) only for nighttime. Aqueous-phase oxidation can be induced by hydrogen peroxide ($H_2O_2$), nitrogen oxide ($NO_2$), ozone ($O_3$), hypochlorous acid (HOCl), hypobromous acid (HOBr), methyl hydroperoxide ($CH_3OOH$), peroxyacetic acid ($CH_3COOOH$), dissolved nitrous acid (HONO), transition-metal ion-catalyzed oxygen (TMI-$O_2$), and the photosensitization (T*) and nitrate photolysis ($P_{NO_3^-}$) only for daytime. Dust-mediated heterogeneous oxidation can be initiated by the surface oxidants ($H_2O_2$, $NO_2$, $O_3$, HOCl, HOBr, $CH_3OOH$, $CH_3COOOH$, HONO) co-adsorbed with $SO_2$. Dust-driven heterogeneous oxidation can be ascribed to the heterogeneous drivers (transition-metal-bearing components and water-soluble ions) on the surfaces of natural dust and clay minerals [Nontronite (NAu), Chlorite (CCa), Montmorillonite (SWy), Kaolin (KGa), Illite (IMt)]. The (a-d) nighttime and (e-h) daytime conditions were distinguished by the different parameterizations. (a, e) Particle acidity-dependent sulfate formation rates of the diverse atmospheric oxidation pathways. The elements' colors and shapes are characterized by the legends in solid boxes. (b, f) Quantified sulfate contribution proportions of the studied gas-phase (red), aqueous-phase (blue), and heterogeneous (yellow) reaction pathways. Sulfate formation rates of the dust-mediated and dust-driven pathways during (c-d) nighttime and (g-h) daytime. The effects of ionic strength on the aqueous-phase oxidation were not taken into account. The dust concentration was set to be 55 µg m⁻³, in representative of the common atmospheric condition of North China (Zhang et al., 2012). The panels in dashed boxes right to the legends illustrate the primary physical-chemical processes of atmospheric sulfate formation. More parameterization and methodology details can be found in the Texts S2-S4 of Supporting Information and Sect. 2.1-2.5 of the main content.

l. 526: should 'merely' be 'only' here?

Response:

We have replaced "merely" by "only".

l. 547: These dust concentrations seem only representative for some regions of the globe (e.g. https://agupubs.onlinelibrary.wiley.com/doi/epdf/10.1029/2021MS002845). Please tone down your statement and state that your conclusions are only relevant for specific regions.

Response:

We have considered the uncertainties relevant to the concentration of dust (line 584-587). The revised sentences are shown as follows.

*"Such dust concentrations are sometimes common in the troposphere, especially during the dust storm periods (Li et al., 2021a; Yin et al., 2021; Filonchyk, 2022) or near the dust source regions (Ke et al., 2022). Therefore, the heterogeneous loss of $SO_2$ by airborne dust surface may have a similar magnitude as the main gas-phase loss process and can be taken as an important sink for $SO_2$."*

l. 587: What do you mean by gas-solid interface? Aerosol particles are never completely dry and thus, gases are always exposed to aqueous surfaces.

Response:

We have replaced the "gas-solid interface" by "humidified gas-phase interface" to emphasize the surface of dust particles exposed to the humid atmospheric environments. The revision in manuscript (line 625-626) is shown as follows.

*"Besides the humidified gas-solid interface, gas-liquid interface of microdroplet is another type of medium that supports the heterogeneous formation of sulfate."*

l. 648: I do not understand the expression 'plausible influencer'. I suggest removing it.

Response:

The expression of "plausible influencer" has been deleted.

l. 674: Please add all data into a suitable repository that can be accessed freely by the readers, corresponding to the journals' data policy https://www.atmospheric-chemistry-and physics.net/policies/data_policy.html.

Response:

We have added all data into a repository and introduced the data availability in the revised manuscript (line 719), as shown follows.

*"Data availability. A dataset for this paper can be accessed at https://data.mendeley.com/datasets/hyvdz7khs6/1."*

---

## Editor Decision (ED2)

Previous referee comments are displayed in green and preceded by 'R', previous author responses are displayed in blue and start by 'A'. They are followed by editor comments.
At the end, there are numerous additional editor comments that need to be addressed before the paper can be considered for acceptance.

**Referee #2**

R4: Is chemistries a real word?
A: As a real word, it was widely utilized in the research papers of this field (e.g. doi.org/10.1021/cr500501m; doi.org/10.1021/cr5003485).

**Editor comment:** 'Chemistry' is defined as a branch of science that studies the composition and transformation of molecules and materials (cf e.g. Wikipedia). Therefore, in this context, it should not be used at all, but should be replaced by 'chemical processes' (or 'chemical composition' where appropriate) throughout the manuscript.

R5. Figure 3: why does the reactive uptake coefficient drastically increases after particle acidity reached ~4.5/ 5?
A: Figure 3 displays the particle-acidity-dependent reactive uptake coefficients (γ) for the dust-mediated heterogeneous chemistry. We have discussed the dramatically increased γ when pH exceeds 4.5 or 5.0, in the revised manuscript (line 367-369).
*"The γ of NO2, O3, HOCl, HOBr and CH3COOOH presents positive dependence toward pH, in accordance with the evolution of the effective Henry's law constant for the studied sulfur species, as theoretically illustrated by Eq. (5)."*

**Editor comment:** I do not understand this response.
A few lines above you state that " the dust mediated heterogeneous oxidation can be assumed to primarily proceed in the surface water layers" but yet you ascribe a significant impact on g by the term that describes the aqueous phase diffusion, i.e. the second term in the right-hand side in Eq. -5.
What are the contributions to the total $\gamma$ of the mass accommodation term ($1/\alpha$) and the aqueous phase diffusion and reaction term $[v/(4\ H\ RT\ (D_a\ k_{chem})^{-1})\ x\ 1/f_r]$?
A figure should be included to demonstrate that indeed the second term increases significantly at increasing pH.

R6. For daytime chemistry, did you consider the possibility of the formation of sulfate radicals on surfaces with high Ti content?
A: We have considered the impact of sulfate radical on the heterogeneous formation of sulfate. Necessary discussions are added into the revised manuscript (line 266-267).
*"Sulfate radical (SO4•−) is generated by the presence of abundant •OH and participates into the oxidation events (Antoniou et al., 2018; Kim et al., 2019; Li et al., 2020b)."*

**Editor comment:** This response and revision is not sufficient to discuss the potential role of sulfate radicals. A discussion on possible chemical mechanisms of the observed processes should be added including the possibility of oxidation by the sulfate radical. You can either add this to section 3.3. or dedicate a full new section to this discussion.

R: Line 373-374: Earlier authors assert that the sulfate formation is more during nighttime and less during the daytime, but this statement is opposite and confusing. Can the author clarify this?

A: In the real atmosphere, the sulfate concentration during nighttime could be comparable to or even exceed that during daytime, as normally explained by the different meteorological conditions. We have explained the relevant details in the revised manuscript (line 396-401) to make our calculation results more reasonable.

 *"It is worthwhile to mention that, in the real atmosphere, the observed sulfate concentration during nighttime may be comparable to or exceed that during daytime, which can be explained by the higher nocturnal humidity facilitating the liquid oxidations or the lower boundary layer causing the adverse diffusion conditions (Liu et al., 2017c; Tutsak and Koçak, 2019; Li et al., 2020c). The relevant meteorological factors were not considered by this comparison model, and the current results emphasized the different sulfate formation potentials through kinetic regime."*

**Editor comment:** I still got confused what you compare in this section vs to the discussion earlier in the manuscript. Please re-organize Section 3.3. such that it becomes clear that these are estimates based on a model for which you assume ambient conditions whereas previously you discussed experimental lab data.
It might help if you refer in the discussion of your experimental data to 'illuminated' and 'dark' conditions only and in the discussion of model results for atmospheric conditions to 'day' and 'night'.

R: Line 380-390: In addition to the temperature dependence, did the authors consider the change of relative humidity during the day and night can also contribute to the heterogeneous chemistry reaction with sulfate?

A: We attempted to carry out the heterogeneous laboratory experiments under the higher relative humidity (RH) relative to the current setting. However, the presence of surface adsorbed water makes the Gaussian/Lorentzian deconvolution difficult to be performed, as reflected by the large uncertainties in determining the relative abundance of $SO_2 \cdot H_2O$ and $SO_3^{2-}$ . Accordingly, the RH of 50% was utilized for the experiments. In the revised manuscript (line 654-656), we have emphasized the importance of various meteorological factors when discussing the atmospheric relevance of heterogeneous chemistry, as shown follows.

*"In addition, the influence of meteorological factors, like temperature, humidity and irradiance, on the atmospheric relevance of heterogeneous chemistry warrants further research. "*

**Editor comment:** I understand that you cannot do any quantitative estimates on the influence of humidity. However, a statement on the extent to which relative humidity would change particle radius and aqueous volume and therefore sulfate formation should be added.

**Additional editor comments**

l. 32: What do you mean by 'in mainstream'?

l. 35: What do you mean by 'aimed liquid reaction' as opposed to 'aqueous phase reaction'? Do you imply reactions in liquid non-aqueous phases?

l. 37: What do you mean by 'gas-solid interactions'? What solid phase are you referring to? In the lines above, you refer to reactions on or in liquid phases.

l. 64 – 66: This sentence is too convoluted and mixes multiple processes and effects. The study by Clegg and Abbatt was performed on ice, Wu et al explored uptake on calcium carbonate and Wang et al. reactivity on hematite. Either split this sentence into three describing the essential information from each study or remove.

l. 68: What do you mean by 'negative light effect'?

l. 73-74: 'In terms of...' – what does this sentence say?

l. 75: 'is more efficient' for what?

l. 77: What do you mean by 'dust community'?

l. 108: correct 'derisble'

l. 119: replace 'Technical Route' by 'Experimental methodology' or something similar.

l. 125: In the sentence starting 'In the dust-mediated...' is a verb missing.

l. 140: Please clarify this sentence. Why don't you state simply the relevant temperature range as being representative for the warm season?

l. 178/179: Which 'gas and aqueous phase parameters' are you referring here to?

l. 195: The reference to Hennigan et al. 2015 would be much better placed in this context here than in l. 190  as it was shown in their study that HSO4- and SO42- are not sufficient to calculate acidity.

l. 212: what do you mean by 'standard item'? Is it the 'standard mixture' or 'standard solution'?

l. 255: replace 'corrects' with 'correlates'

l. 259: Why only 'nocturnal heterogeneous reaction'? Why does this not happen during the day? Could 'nocturnal'  be omitted in this sentence?

l. 263 – 291: This text should not be in the result section but belongs in the introduction where it is partially already cited. Please condense this paragraph and either move relevant parts into the discussion of your results (which are not presented at this place of the paper yet) or to the introduction.

l. 350: This sentence is not clear. Please clarify what you want to say here.

l. 350/351:' no significant correlations can be found between γ and pH' contradicts your later discussion and findings in Figure 3. Please clarify.

l. 359/360: Are three digits of the pH value significant figures? What is the variability in this value?

l. 430: replace 'great heterogeneous performances' by 'significant contributions by heterogeneous sulfate formation' (or similar)

l. 446 – 448: The sentence starting 'As proved by ...' seems out of place here. Either remove or clarify its connection to the text in this paragraph.

l. 454-461: 1) Refer to Figure 6 at the beginning of this text , 2) Be more specific in the text, rather than just saying longest, second longest etc lifetime.

Figure 4: 1) Make labels on panels b, c, d, f, g, h larger. 2) Explain the colors in panels a and e. Are they same as in panels b and f?

l. 526: should 'merely' be 'only' here?

l. 547: These dust concentrations seem only representative for some regions of the globe (e.g. https://agupubs.onlinelibrary.wiley.com/doi/epdf/10.1029/2021MS002845). Please tone down your statement and state that your conclusions are only relevant for specific regions.

l. 587: What do you mean by gas-solid interface? Aerosol particles are never completely dry and thus, gases are always exposed to aqueous surfaces.

l. 648: I do not understand the expression 'plausible influencer'. I suggest removing it.

l. 674: Please add all data into a suitable repository that can be accessed freely by the readers, corresponding to the journals' data policy https://www.atmospheric-chemistry-and-physics.net/policies/data_policy.html .